# SCVO: Addressing Sparse But Critical Variable Overwhelm In VLMs For Advertising Image Preference Prediction Across Multi-Country Markets

## Abstract

Vision-language models (VLMs) have demonstrated remarkable capabilities in multimodal tasks, yet their sensitivity to sparse and critical variables that are often overwhelmed remains unexplored. The image preference prediction across multi-country markets task serves as a representative case in this regard. Specifically, VLMs (e.g., Qwen-VL) are tasked with judging between two images (A and B) for the same product across diverse markets (e.g., Korea, France), and the model's predictions often collapse to a single output (e.g., always "A") despite ground-truth preferences varying by country. This failure is attributed to Sparse Critical Variable Overwhelm (SCVO): the model is overwhelmed by dominant high-volume variables (e.g., product attributes, image patches consuming hundreds of tokens), while the critical low-volume variables (e.g., country names consuming only a few tokens) are statistically drowned out. To study this, we first collect a dataset of real-world advertising image click-through preferences across multi-country markets, and then present a novel training framework that strategically mitigates SCVO, and we use it to train on the dataset, yielding CountryReward, a reward model for advertising image preference prediction across multi-country markets. Our framework involves three tailored modules: (1) a cross-country retrieval-augmented generation module that injects click-through preferences aligned with target markets into the training process, enhancing localized relevance prediction. (2) a country adapter module that dynamically modulates image representations based on textual country embeddings, enabling precise visual preference adaptation for diverse markets. (3) a focus-driven penalty loss function that penalizes mispredictions related to the overlooked variable more heavily. Finally, we apply the CountryReward as the reward model to finetune VLMs through Reinforcement Learning (RL), enabling the model to output background designs fed to the text-to-image model (e.g., SDXL) and generate effective e-commerce images for a targeted country. Experiments on the proposed dataset show that our approach significantly mitigates the SCVO effect and improves the preference prediction accuracy. This work highlights the need for robust handling of sparse critical variables in VLMs and offers a scalable solution for real-world applications where subtle contextual shifts drive decision-making.

## 1 Introduction

Vision-language models (VLMs) (Wang et al., 2024; Chen et al., 2024) have emerged as a cornerstone of modern artificial intelligence, demonstrating remarkable proficiency across a broad spectrum of multimodal tasks, from visual question answering and image captioning to complex reasoning about visual scenes. Their ability to learn powerful multimodal representations has been used as multimodal reward models (RMs) (He et al., 2024; Liu et al., 2025a; Xu et al., 2025), which can provide crucial reward signals to guide model training (Ouyang et al., 2022; Rafailov et al., 2024; Schulman et al., 2017) and inference (Gulcehre et al., 2023; Snell et al., 2024). However, despite their impressive performance, a critical aspect of their real-world applicability remains underex-

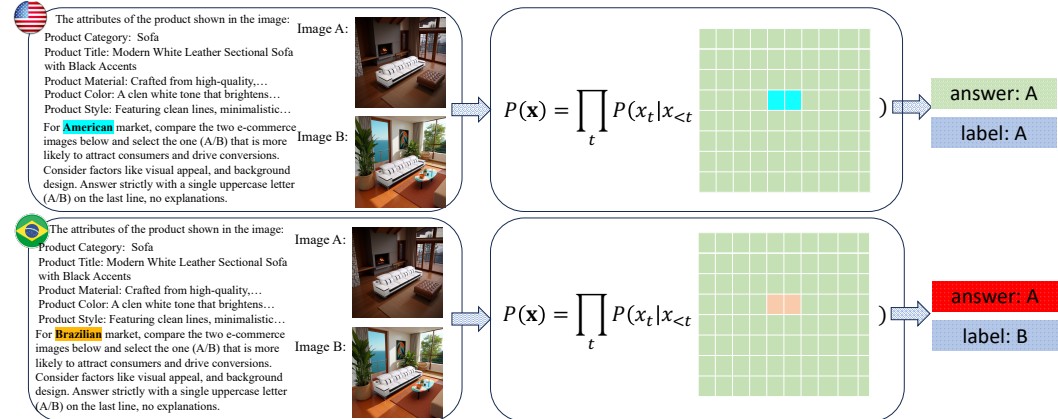

Figure 1: Sparse Critical Variable Overwhelm (SCVO): The only differing variable (blue or orange squares) shown in the two examples is the country names, which are sparse but critical. Other high-volume variables (green squares shown in the image) are dominated, including the same product attributes and the same two images. The model's autoregressive decision-making process collapses, producing a market-invariant prediction.

plored: their robustness and sensitivity to specific, often underrepresented, variables in instruction-following scenarios. In particular, the sensitivity of VLMs to the sparse but critical variables that are overwhelmed by excessive or complex input instructions poses a significant challenge to their reliability and trustworthiness.

The image preference prediction across multi-country markets task is a representative case of the issue described above. As shown in Figure 1, in this setting, models such as Qwen-VL (Bai et al., 2023) are required to discern nuanced preferences between two images (A and B) depicting the same product across distinct markets, for instance, comparing consumer choices between the American market and the Brazilian market. Despite clear empirical evidence that human preferences vary significantly by region, VLMs frequently exhibit a collapse in their decision-making process, defaulting to a single output choice (e.g., persistently selecting "A") irrespective of the target market. This failure is attributed to Sparse Critical Variable Overwhelm (SCVO): a VLM's autoregressive probability chain, $P(\mathbf{x}) = \prod_t P(x_t|x_{<t})$, is dominated by high-volume variables (e.g., product attributes, image patches consuming hundreds of tokens), causing the influence of the sparse critical variable (e.g., country names consuming only a few tokens) to be statistically drowned out during attention-weighted feature fusion. The green squares in Figure 1 represent the same variables, drowning out the influence of the blue or orange squares, resulting in the model failing to allocate sufficient sensitivity to the critical variable, breaking the dependence implied by the chain rule.

To study this, we first collect a dataset, called Multi-Country Ad Click Preference (MACP), a real-world dataset of advertising image click-through preferences across multi-country markets. Our dataset contains 823K training samples and 180K test samples involving 10 countries. The number of samples from different countries is balanced. Each sample includes two different images of the same product, their Click-Through Rate (CTR), and the detailed product information, including titles, categories, and other relevant attributes. These samples are collected from the same e-commerce platform, ensuring consistency in the data source and characteristics.

In the domain of advertising image preference prediction across different country markets, the presence of SCVO poses significant challenges to model accuracy. To address this issue, we introduce a novel training framework specifically designed to mitigate SCVO-related limitations. The proposed approach leverages a comprehensive dataset to train CountryReward, a specialized reward model for advertising image preference prediction across different country markets. Our framework incorporates three strategically designed components: (1) A cross-country retrieval-augmented generation module that integrates click-through preference signals aligned with target markets, thereby enhancing the model's capacity for localized relevance assessment. (2) A country adapter module that dynamically adjusts visual representations using textual country embeddings, enabling fine-grained

adaptation to diverse market-specific characteristics. (3) A focus-driven penalty loss function that assigns weighted penalties to prediction errors associated with previously overlooked variables. Through these innovations, our framework significantly mitigates the SCVO effect and improves prediction accuracy while maintaining robustness across varied market environments.

In the domain of cross-border e-commerce visual content generation, accurately aligning generated imagery with market-specific preferences remains a challenging task. To address this limitation, we integrate CountryReward as a reward model to finetune VLMs through Reinforcement Learning (RL). The optimized VLM subsequently generates detailed background designs tailored to specific markets, which serve as inputs to text-to-image models such as SDXL (Podell et al., 2023). The final output consists of highly targeted e-commerce images designed to resonate with consumers in particular countries, thereby enhancing visual relevance and commercial effectiveness.

**RFrLS W1:** We summarize our contributions as follows:

- Identifying and formalizing a novel research problem: This work is the first to systematically identify a critical deficiency in VLMs: Sparse Critical Variable Overwhelm (SCVO).

- We collect the *Multi-Country Ad Click Preference (MACP)* dataset, a novel large real-world e-commerce advertising image click-through preference dataset comprising data from 10 countries.

- We propose *CountryReward*, a reward model that can accurately predict image preference across multi-country markets. This model integrates three tailored modules:
  - A *Cross-Country Retrieval-Augmented Generation* module that enhances the model's understanding of localized relevance by leveraging click-through data aligned with target markets.
  - A *Country Adapter Module* that enables fine-grained adaptation to diverse market-specific features by dynamically adjusting visual representations using textual country embeddings.
  - A *Focus-Driven Penalty Loss* that can adaptively apply varying penalties based on the focus of features such as country, image, and product when a prediction error occurs.

- We further use *CountryReward* as a reward model to finetune VLMs via Reinforcement Learning (RL), enabling the model to generate market-adapted background designs for each country and helping text-to-image models (e.g., SDXL) generate images for targeted country markets.

## 2 RELATED WORK

### 2.1 MULTIMODAL REWARD MODELS

Multimodal reward models play an increasingly critical role in aligning vision, understanding, and generation systems with human preferences. A widely adopted strategy involves fine-tuning vision-language models (VLMs) (Li et al., 2024; Bai et al., 2022), capitalizing on their strong multimodal alignment capacities to acquire reward functions reflective of human judgments. Previous research has investigated reward modeling in the context of visual generation (Liu et al., 2025a; Xu et al., 2025; He et al., 2024; Wang et al., 2025b) and visual understanding tasks (Zang et al., 2025; Xiong et al., 2025). For example, Ziegler et al. (2020) devises an efficient pipeline for building multimodal preference datasets and utilizes existing high-quality data to train IXC-2.5-Reward, a model capable of accurately assessing outputs from visual understanding tasks. Similarly, Wang et al. (2025b) gathers human feedback to create a dataset of human-rated videos used to train LiFTCritic, a reward model designed to evaluate how closely generated videos match human expectations. Wang et al. (2025c) proposes UnifiedReward, a unified reward model that can evaluate image and video generation as well as understanding tasks, showing that collaborative learning across various visual domains leads to significant synergistic improvements. Wang et al. (2025a) present UnifiedReward-THINK, a unified multimodal reward model based on lengthy chain-of-thought (CoT) reasoning, facilitating multi-dimensional long-chain reasoning for visual understanding and generation tasks. Despite their promising performance, existing reward models do not consider the setting where instructions contain sparse but critical variables, frequently leading to imprecise or untrustworthy

reward signals. To this end, we propose CountryReward, a reward model that can adaptively consider different semantic clues (e.g., country name, product attributes, and image features), especially the sparse but critical cues that are overwhelmed by dominant features in the input.

## 2.2 LEARNING FROM HUMAN FEEDBACK

Reinforcement Learning from Human Feedback (RLHF) (Bai et al., 2022; Luong et al., 2024; Ziegler et al., 2020; Ouyang et al., 2022; Jiao et al., 2025; Zhang et al., 2025; Ying et al., 2024; Yang et al., 2024; OpenAI et al., 2024; Shao et al., 2024; Hui et al., 2024) collects human feedback regarding model outputs. The feedback is then used to optimize the generative model via reinforcement learning methods such as PPO (Schulman et al., 2017), DPO (Rafailov et al., 2024), and GRPO (DeepSeek-AI et al., 2025). The RL applications for VLMs include visual quality assessment (Li et al., 2025), visual perception and reasoning (Liu et al., 2025b), mitigating hallucinations (Sun et al., 2023; Yu et al., 2024a), and aligning models with human preferences (Yu et al., 2024b; Zhou et al., 2024). To bridge VLMs and T2I models, a classifier (Wu et al., 2023) is trained on human-curated image choices and outputs a human preference score used to adapt the T2I model. Parrot (Lee et al., 2024) jointly optimizes the prompt expansion and T2I model network together via a multi-reward RL approach for improving image quality. CAIG (Chen et al., 2025) first explores the utilization of VLMs for generating advertising images by optimizing for CTR as the objective. Through RL, the CTR reward model is used to finetune VLMs. The finetuned VLMs can generate background designs, which are fed into T2I models to generate an image better aligned with user preferences. However, they noted that a limitation is that their reward model overlooks the preferences of niche market segments, and this lack of personalization can lead to suboptimal experiences for diverse user groups. Moreover, our work can better integrate user preferences across different country markets. We use our CountryReward, trained to overcome SCVO, as a preference reward model for fine-tuning a VLM. This allows the generative model to produce background designs optimized for specific country markets that cater to the needs and behaviors of global users.

## 3 METHOD

### 3.1 COUNTRYREWARD

As shown in Figure 2, our proposed model, named CountryReward, is built upon the Qwen2-VL framework, incorporating several key innovations to better handle SCVO and improve performance. The overall architecture consists of a vision transformer for image feature extraction, a language model for text understanding, and a country adapter mechanism. Additionally, we introduce a focus-driven regularization technique to guide the model's focus toward critical tokens (e.g., country, product, and image tokens). In addition, before training, there is a retrieval-augmented generation process to create an augmented choice based on the experiential knowledge.

#### 3.1.1 CROSS-COUNTRY RETRIEVAL-AUGMENTED GENERATION

To enhance the model's capacity for localized relevance prediction, we propose a Cross-Country Retrieval-Augmented Generation (CC-RAG) that incorporates click-through preferences aligned with target markets. This module enables the model to leverage domain-specific behavioral patterns from regional users, thereby improving the model's sensitivity to the country variable, a sparse but critical variable. For efficiency, CC-RAG is applied before training CountryReward to obtain the augmented choice $\hat{y}_{\text{aug}}$.

**RZ86t W2&Q2&S1:** We employ a two-stage hierarchical retrieval process with strict retrieval scope control to avoid label leakage. Specifically, the retrieval index is constructed solely from the training set, and a maximum similarity threshold (set to 0.7 in our experiments) is applied. The process is as follows:

**Index Construction:** For each country $c \in \{\text{US}, \text{FR}, \text{KR}, ...\}$, we construct a text embedding matrix $\mathbf{T}_c \in \mathbb{R}^{N \times d}$ and an image embedding matrix $\mathbf{I}_c \in \mathbb{R}^{N \times d}$ from the training set, where $N$ is the number of training items.

**Stage 1: Text-based Retrieval**

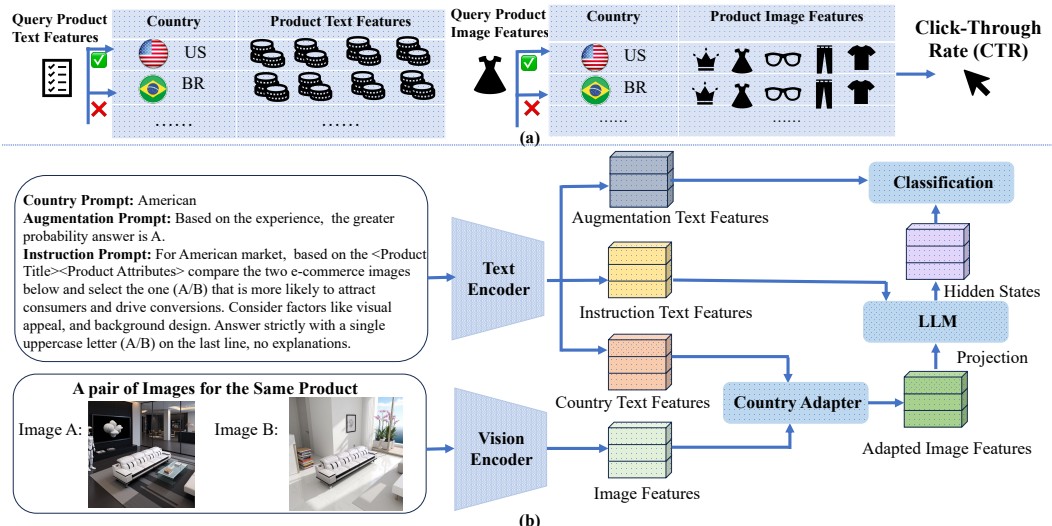

Figure 2: CountryReward: Figure (a) illustrates that based on the American knowledge base, the query product text features first retrieve items which have similar product text features, and then the query product image features retrieve the CTR values among the items retrieved in the first stage. Figure (b) depicts the training framework of CountryReward.

Given the query text embedding $\mathbf{q}_t \in \mathbb{R}^d$, we compute its similarity with all training text embeddings. We then apply a maximum similarity threshold $\tau = 0.7$ (the query text is implicitly excluded because its self-similarity would typically exceed the threshold), and finally we select the top-k highest similarity indices from $\mathcal{C}_t$ to form the text-retrieved neighbor set:

$$\mathbf{s}_t = \mathbf{q}_t \cdot \mathbf{T}_c^T \in \mathbb{R}^N, \mathcal{C}_t = \{j \mid \mathbf{s}_t[j] < \tau\} \tag{1}$$

Finally, we select the top-k highest similarity indices from $\mathcal{C}_t$ to form the text-retrieved neighbor set:

$$\mathcal{N}_t = \text{Top-}k\left(\{\mathbf{s}_t[j] \mid j \in \mathcal{C}_t\}\right) \tag{2}$$

**Stage 2: Image-based Retrieval**

Within the subset of image embeddings $\mathbf{I}_{\mathcal{N}_t}$ corresponding to $\mathcal{N}_t$ (i.e., the images of those training items), we perform retrieval for each candidate image embedding $\mathbf{q}_i^A$ and $\mathbf{q}_i^B$. We apply the same threshold $\tau = 0.7$ to filter out high-similarity items (the query image is implicitly excluded because its self-similarity would typically exceed the threshold), and we then select the top-m highest similarity indices. Indices for candidate A are computed as follows:

$$\mathcal{N}_i^A = \text{Top-}m\left(\{\mathbf{s}_i^A[j] \mid j \in \mathcal{C}_i^A\}\right), \quad \text{where } \mathbf{s}_i^A = \mathbf{q}_i^A \cdot \mathbf{I}_{\mathcal{N}_t}^T, \quad \mathcal{C}_i^A = \{j \in \mathcal{N}_t \mid \mathbf{s}_i^A[j] < \tau\} \tag{3}$$

Similarly, for candidate B:

$$\mathcal{N}_i^B = \text{Top-}m\left(\{\mathbf{s}_i^B[j] \mid j \in \mathcal{C}_i^B\}\right), \quad \text{where } \mathbf{s}_i^B = \mathbf{q}_i^B \cdot \mathbf{I}_{\mathcal{N}_t}^T, \quad \mathcal{C}_i^B = \{j \in \mathcal{N}_t \mid \mathbf{s}_i^B[j] < \tau\} \tag{4}$$

**RFrLS Q2: Preference Aggregation:** A position-aware weighting scheme is used, with weight $w_i = m - i$ for the i-th retrieved neighbor. Rather than relying on a single best match, this method maintains robustness by aggregating multiple neighbors. The preference scores are computed as:

$$S_A = \frac{\sum_{i=1}^{m} w_i \cdot \mathbb{I}\left[\text{CTR}\left(\mathbf{I}_{\mathcal{N}_i^A}\right) \geq \text{CTR}\left(\mathbf{I}_{\mathcal{N}_i^B}\right)\right]}{\sum_{i=1}^{m} w_i}, \quad S_B = \frac{\sum_{i=1}^{m} w_i \cdot \mathbb{I}\left[\text{CTR}\left(\mathbf{I}_{\mathcal{N}_i^B}\right) > \text{CTR}\left(\mathbf{I}_{\mathcal{N}_i^A}\right)\right]}{\sum_{i=1}^{m} w_i} \tag{5}$$

where $\mathbb{I}[\cdot]$ is the indicator function and $\text{CTR}(\cdot)$ denotes the click-through rate knowledge. The final augmented prediction is:

$$\hat{y}_{\text{aug}} = \begin{cases} A & \text{if } S_A > S_B \\ B & \text{otherwise} \end{cases} \tag{6}$$

This process ensures retrieval is confined to the training set, avoids the query item itself, near-duplicates, or identical products through the similarity threshold and the position-weighted voting system, thereby limiting the scope to semantically similar but distinct items and preventing label leakage.

### 3.1.2 Country Adapter Module

Effectively adapting large vision-language (VL) models to diverse global markets requires sensitivity to country-specific visual content preferences. To this end, we introduce a Country Adapter Module (CAM). Inspired by FiLM (Perez et al., 2017), this module dynamically modulates the visual features extracted by the vision encoder based on textual embeddings derived from country-specific information, allowing the model to adjust its perceptual processing for different country markets.

The core mechanism involves generating a set of affine transformation parameters (scale and shift) from a learned country embedding. Let $\mathbf{c}_i \in \mathbb{R}^d$ denote the mean-pooled embedding vector of the tokenized country name for the $i$-th sample in a batch, where $d$ is the hidden dimension size.

The adaptation parameters are generated by a small feed-forward network, the Country Adapter:

$$\gamma_i, \beta_i = \text{Split}(\text{CountryAdapter}(\mathbf{c}_i)) \tag{7}$$

where CountryAdapter: $\mathbb{R}^d \to \mathbb{R}^{2d}$ is implemented as:

$$\text{CountryAdapter}(\mathbf{c}_i) = \mathbf{W}_2(\text{ReLU}(\mathbf{W}_1\mathbf{c}_i + \mathbf{b}_1)) + \mathbf{b}_2 \tag{8}$$

Here, $\mathbf{W}_1 \in \mathbb{R}^{d/2 \times d}$, $\mathbf{b}_1 \in \mathbb{R}^{d/2}$, $\mathbf{W}_2 \in \mathbb{R}^{2d \times d/2}$, $\mathbf{b}_2 \in \mathbb{R}^{2d}$ are learnable parameters. The output is split into two vectors $\gamma_i \in \mathbb{R}^d$ (scale) and $\beta_i \in \mathbb{R}^d$ (shift).

Let $\mathbf{V}_i \in \mathbb{R}^{N \times d}$ represent the sequence of visual features (e.g., N image patch embeddings) corresponding to the i-th sample before integration into the language model's input embedding space. The adapted visual features $\tilde{\mathbf{V}}_i$ are computed via an element-wise affine transformation:

$$\tilde{\mathbf{V}}_i = \gamma_i \odot \mathbf{V}_i + \beta_i \tag{9}$$

where $\odot$ denotes the Hadamard (element-wise) product. This transformation is applied to the entire set of visual features $\mathbf{V}_i$ associated with the specific country embedding $\mathbf{c}_i$. This allows the model to selectively emphasize or suppress certain visual patterns based on learned country-specific cues, effectively tailoring the visual representation to relevant country markets.

### 3.1.3 Focus-Driven Penalty Loss

To enhance the model's ability to leverage multimodal inputs effectively and improve country-specific adaptation, we propose a novel focus-driven penalty loss (FDPL), which is designed to penalize the model when it fails to adequately attend to input components (e.g., country tokens, product descriptors, or image features) during erroneous predictions, while imposing no additional penalty for correct predictions. This is achieved by introducing an auxiliary penalty term that is dynamically scaled based on the relative focus allocated to each key component. Let $\mathbf{H} \in \mathbb{R}^{T \times d}$ denote the hidden states of the final transformer layer, where T is the sequence length and d is the hidden dimension. The hidden states $\mathbf{H}$ are obtained by feeding the adapted visual features $\tilde{\mathbf{V}}$ (stated in 3.1.2) and instruction text features $\tilde{\mathbf{T}}$ into the VLM. Next, for each sample in a batch, we identify the token positions of key input components: country token $t_c$, product token $t_p$, and image token $t_i$. The focus intensity toward each component is approximated using the L2-norm of their corresponding hidden states:

$$\text{Focus}_c = \frac{\|\mathbf{H}[t_c]\|_2}{\sum_{j=1}^{T} \|\mathbf{H}[j]\|_2}, \quad \text{Focus}_p = \frac{\|\mathbf{H}[t_p]\|_2}{\sum_{j=1}^{T} \|\mathbf{H}[j]\|_2}, \quad \text{Focus}_i = \frac{\|\mathbf{H}[t_i]\|_2}{\sum_{j=1}^{T} \|\mathbf{H}[j]\|_2}, \tag{10}$$

where $\|\cdot\|_2$ is the L2-norm. The penalty terms for country, product, and image are defined as:

$$\mathcal{P}_c = 1 - \text{Focus}_c, \quad \mathcal{P}_p = 1 - \text{Focus}_p, \quad \mathcal{P}_i = 1 - \text{Focus}_i. \tag{11}$$

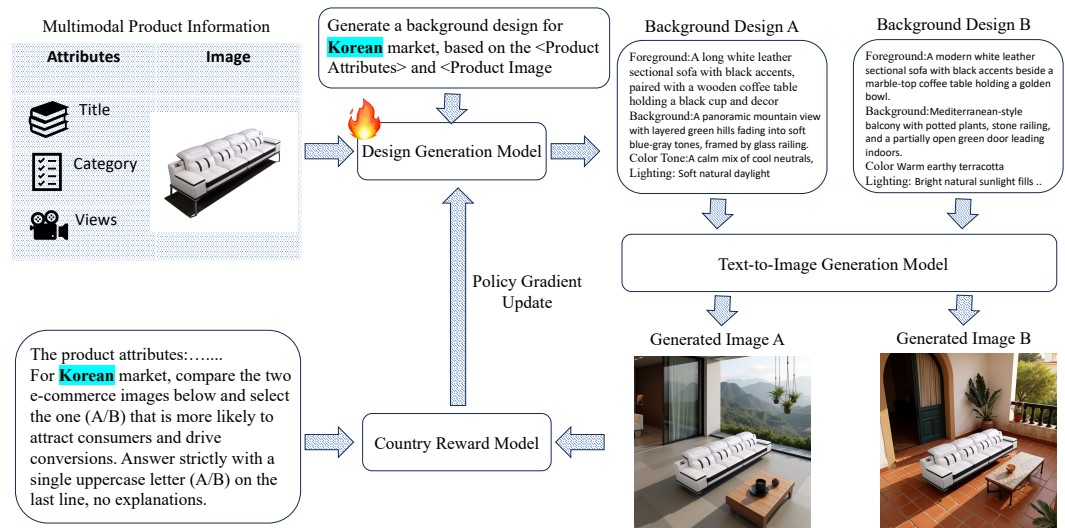

Figure 3: Country-Adapted Background Design Generation Framework: The Design Generation Model generates background designs for the target country, the T2I Generation Model creates product images according to the designs, the Country Reward Model then predicts the image preference according to the target country, giving feedback to optimize the Design Generation Model.

These penalties are activated only when the model makes an incorrect prediction. For a batch of size $B$, let $\hat{y}_i$ and $y_i$ be the predicted probability and ground-truth label for the i-th sample, respectively. The indicator function for incorrect prediction is:

$$\mathbb{I}_i = \begin{cases} 1 & \text{if } (\hat{y}_i \geq 0.5) \neq (y_i = 1), \\ 0 & \text{otherwise.} \end{cases} \tag{12}$$

The total penalty loss for the batch is computed as:

$$\mathcal{L}_{\text{penalty}} = \frac{1}{B} \sum_{i=1}^{B} \mathbb{I}_i \cdot \left( \mathcal{P}_c^{(i)} + \mathcal{P}_p^{(i)} + \mathcal{P}_i^{(i)} \right). \tag{13}$$

The overall training objective combines the binary cross-entropy loss $\mathcal{L}_{\text{BCE}}$ with the penalty loss:

$$\mathcal{L} = \mathcal{L}_{\text{BCE}}(\sigma(\mathbf{W}_{\text{classifier}} \cdot (\mathbf{e}_{\text{aug}} + \mathbf{h}_{\text{last}})), y) + \lambda \mathcal{L}_{\text{penalty}}, \tag{14}$$

Following common practice in sequence classification with LLMs (Touvron et al., 2023), we use $\mathbf{h}_{\text{last}}$, the last token of the hidden state as the discriminative representation, since it summarizes the contextual information of the entire input sequence, $\mathbf{e}_{\text{aug}}$ is the text embedding derived from the augmented choice $\hat{y}_{\text{aug}}$ (stated in 3.1.1), $\mathbf{W}_{\text{classifier}}$ refers to the weight matrix in the classifier component of our model, $\sigma$ is the sigmoid activation function, $\lambda$ is a scaling hyperparameter (set to 0.1). This design encourages the model to strengthen its focus on under-attended components when errors occur, thereby improving feature utilization and country-specific decision-making.

## 3.2 COUNTRY-ADAPTED BACKGROUND DESIGN GENERATION

To address the challenge of generating market-adapted visual content for cross-border e-commerce, we propose a reinforcement learning-based framework that leverages a CountryReward model to optimize the generation of country-specific background designs. The framework consists of three stages (shown in Figure 3):

Firstly, a Design Generation Model (DGM), implemented as a finetuned Qwen2-VL (Wang et al., 2024), a Vision-Language Model (VLM) trained on the proposed dataset, which generates textual

Table 1: Comparison of accuracy performance across different reward models on MACP. Higher is better for both accuracy and sensitivity, and their units are percentage (%).

| Model | Accuracy | Sensitivity | BR | CL | ES | FR | KR | JP | US | MX | AU | SA |
|---|---|---|---|---|---|---|---|---|---|---|---|---|
| SAIL2-8B | 49.26 | 26.32 | 48.96 | 48.85 | 49.35 | 49.38 | 49.05 | 49.12 | 49.15 | 49.95 | 49.53 | 49.28 |
| InternVL3-8B | 49.15 | 27.10 | 49.86 | 48.67 | 50.04 | 49.04 | 49.99 | 48.25 | 48.94 | 49.48 | 48.42 | 48.54 |
| Qwen2-VL-7B | 49.76 | 31.55 | 49.77 | 50.26 | 48.85 | 49.16 | 48.20 | 50.58 | 51.17 | 49.18 | 49.75 | 50.70 |
| Qwen2-VL-7B (finetuned) | 44.61 | 20.82 | 46.96 | 43.71 | 49.76 | 46.15 | 47.78 | 40.99 | 42.99 | 46.66 | 40.92 | 40.16 |
| Qwen2-VL-7B (with FC Head) | 55.60 | 36.73 | 53.69 | 56.57 | 50.86 | 54.41 | 53.07 | 59.10 | 56.94 | 53.75 | 58.47 | 59.21 |
| CountryReward (w/o CC-RAG) | 56.81 | 37.40 | 54.93 | 57.05 | 53.80 | 55.39 | 54.48 | 59.57 | 57.65 | 55.64 | 59.52 | 60.11 |
| CountryReward (w/o CAM) | 57.98 | 38.95 | 55.50 | 57.85 | 53.78 | 56.75 | 55.19 | 61.47 | 59.39 | 55.98 | 61.71 | 62.21 |
| CountryReward (w/o FDPL) | 56.95 | 37.47 | 54.79 | 57.09 | 52.80 | 56.12 | 54.44 | 61.01 | 58.79 | 55.33 | 58.83 | 60.33 |
| CountryReward | **60.37** | **40.84** | **58.70** | **61.12** | **56.33** | **59.38** | **57.88** | **63.54** | **61.82** | **58.72** | **63.30** | **62.93** |

background designs for the target country. The process of training and inference of DGM is:

$$d = \text{DGM}(\text{country}, \text{pro}, I_{ori}) \tag{15}$$

where country, pro, $I_{ori}$ represent the target country, the product attributes, and the original product image, respectively. Next, the pair of generated background designs $d_A$ and $d_B$ will be obtained via $d_A = \text{DGM}(\text{country}, \text{pro}, I_{ori})$ and $d_B = \text{DGM}(\text{country}, \text{pro}, I_{ori})$.

Secondly, a controlled Text-to-Image (T2I) Generation Model, implemented by integrating a Stable Diffusion Model (Podell et al., 2023) with a ControlNet adapter (Zhang et al., 2023), that allows us to condition the generation process on a control map based on the canny edge of the product image. The component can enable the generated image to not only align with target market preferences but also adhere to the original product layout. The T2I$(p, I_{ori})$ function can be represented as follows:

$$
\begin{aligned}
z_{t-1} &= \frac{1}{\sqrt{\alpha_t}}\left(z_t - \frac{1 - \alpha_t}{\sqrt{1 - \bar{\alpha}_t}} \epsilon_\theta(z_t, t, \tau p, \text{Canny}(I_{ori}))\right) + \sigma_t \epsilon, \\
I &= \text{Decoder}(z_0),
\end{aligned}
\tag{16}
$$

where $z_t$ is the latent representation at timestep $t$, $I_{ori}$ is the input image, $p$ is the text prompt, $\tau p$ is the text encoder, $\alpha t$, $\bar{\alpha}_t$, $\sigma_t$ are the noise scheduling parameters, Canny$(\cdot)$ is the canny edge extraction function, and Decoder$(\cdot)$ can decode the final latent $z_0$ to the generated image $I$. We will obtain the pair of generated images $I_A$ and $I_B$ via $I_A = \text{T2I}(d_A, I_{ori})$ and $I_B = \text{T2I}(d_B, I_{ori})$.

Thirdly, a Country Reward Model (CRM) that predicts the country-specific preference for the pair of generated images ($I_A$ and $I_B$). According to the obtained preference choice, the design of a more attractive image is denoted as $d^+$, and the design of a less attractive image is represented as $d^-$. In order to finetune the DGM to choose a more attractive design $d^+$ and reject less attractive ones $d^-$. The feedback signals provided by CRM are used to refine the DGM via Direct Preference Optimization (DPO) (Rafailov et al., 2024). Specifically, given an optimization policy model DGM$_\theta$ and a reference model DGM$_{ref}$, the optimization objective is:

$$\mathcal{L}_{dpo} = -\log \sigma\left(\beta \log \frac{\text{DGM}_\theta(d^+|\text{country}, \text{pro}, I_{ori})}{\text{DGM}_{ref}(d^+|\text{country}, \text{pro}, I_{ori})} - \beta \log \frac{\text{DGM}_\theta(d^-|\text{country}, \text{pro}, I_{ori})}{\text{DGM}_{ref}(d^-|\text{country}, \text{pro}, I_{ori})}\right), \tag{17}$$

where $\sigma$ and $\beta$ are the sigmoid activation function and a regularization parameter, respectively. DGM$_\theta$ and DGM$_{ref}$ are policy and reference models, respectively, where the policy one is optimized while the reference one is frozen. In addition, we utilize the finetuned DGM to generate background designs for products. These designs are then fed into the T2I Generation Model to create product advertising images, ensuring that the generated background designs are tailored to the target country's preferences.

## 4 EXPERIMENT

### 4.1 EXPERIMENTAL SETUP

**RZ86t W2: Dataset.** We evaluate our proposed method on the collected Multi-Country Ad Click Preference (MACP) dataset. We rigorously filtered near-duplicates and identical samples between

training and test sets. Specifically, we used product IDs, image perceptual hashes, and textual embeddings to identify and remove any overlapping samples. This prevents the model from leveraging the same-product information across splits. Finally, the dataset comprises 823K training samples and 180K test samples, uniformly distributed across 10 distinct country markets, including "BR", "CL", "ES", "FR", "KR", "JP", "US", "MX", "AU", and "SA". Each sample contains detailed product information, including titles, categories, tags, and other relevant attributes, two different advertising images (A and B) for the same product, and the Click-Through Rate (CTR) indicating user preference in the specific market. To ensure the confidence level of the click-through rate (CTR), the CTR data is obtained by dividing 30-day cumulative clicks by the corresponding cumulative impressions. The dataset is sourced from a major cross-border e-commerce platform, containing 67K product samples with 250K unique advertising images, and ensuring consistency in data source and characteristics.

## 4.2 ANALYSIS ON COUNTRYREWARD

**Evaluation Metric.** To evaluate the performance of our CountryReward, we introduce the accuracy and sensitivity metrics. **RZ86t W9:** From a business perspective, a model that achieves higher accuracy in predicting which image historically garnered a higher CTR has learned a reward function that directly mirrors past user engagement, and a high sensitivity score ensures that the model's preference judgments are consistent and reliable across country pairs for the same product, which is a prerequisite for trustworthy global deployment. Accuracy measures the proportion of correct predictions, and sensitivity measures the proportion of simultaneous correct predictions across different country combinations, reflecting the cross-country sensitivity of the model's predictions, which are defined as:

$$\text{Accuracy} = \frac{1}{N}\sum_{i=1}^{N}\mathbb{I}[\hat{y}_i = y_i], \quad \text{Sensitivity} = \frac{\sum_{i=1}^{M}\sum_{(c_j,c_k)\in\mathcal{C}_2(S_i)}\mathbb{I}[\hat{y}_{i,c_j} = y_{i,c_j} \land \hat{y}_{i,c_k} = y_{i,c_k}]}{\sum_{i=1}^{N}|\mathcal{C}_2(S_i)|} \tag{18}$$

where $N$ represents the total number of samples, $\hat{y}_i$ denotes the predicted class label for the i-th sample, obtained by thresholding the sigmoid normalized logits at $0.5$ and mapping to class labels A, B, $y_i$ corresponds to the ground-truth label, $M$ is represented as total number of unique items, $S_i$ is the set of countries for item $i$, $\mathcal{C}_2(S_i)$ is the set of all 2-combinations of countries in $S_i$, $\hat{y}_{i,c}$ is the predicted answer for item $i$ in country $c$, $y_{i,c}$ is the ground-truth answer for item $i$ in country $c$, and $\mathbb{I}[\cdot]$ is the indicator function (1 if condition true, 0 otherwise).

**RZ86t Q1&W1:** The ground-truth labels $y_i$ (A or B) are derived from the CTR data as follows: We have two images (A and B) with their respective CTRs: $\text{CTR}_A$ and $\text{CTR}_B$. We use a threshold-based approach to account for statistical uncertainty: Compute the relative CTR difference, $\Delta = \frac{|\text{CTR}_A - \text{CTR}_B|}{\max(\text{CTR}_A, \text{CTR}_B)}$. We only include pairs where $\Delta \geq \theta$ (threshold, set to 0.1) to ensure meaningful preference distinctions. To mitigate label ambiguity, all pairs with $\Delta < \theta$ are excluded from both the training and evaluation sets. The label $y_i$ is assigned as A if $\text{CTR}_A > \text{CTR}_B$, and B otherwise, for pairs passing the threshold. The labels $y_{i,c}$ are obtained in the same way.

**Quantitative Results.** As shown in Table 1, experimental results on our MACP benchmark demonstrate that SAIL2-8B (Yin et al., 2025), InternVL3-8B (Zhu et al., 2025), and Qwen2-VL-7B (Wang et al., 2024) exhibit a significant performance gap, with accuracy approximately $11.11\%$, $11.23\%$, and $10.61\%$ lower than our proposed method ($60.37\%$) respectively, alongside notably poorer sensitivity, and when the original Qwen2-VL model is finetuned on the MACP dataset using a standard approach, the model exhibited a complete prediction collapse. These performance degradations primarily stem from their vulnerability to the SCVO effect.

**Ablation Study.** To dissect the contribution of each proposed component, we conduct ablation studies on the test set. The results are summarized in Table 1. **RZ86t W2:** Removing the Cross-Country Retrieval-Augmented Generation (w/o CC-RAG) leads to a $3.56\%$ drop in overall accuracy and $3.44\%$ drop in sensitivity. This highlights the importance of injecting market-specific preference knowledge to guide the model. We also analyze the correlation between the augmented choice $\hat{y}_{aug}$ and ground-truth labels. The Pearson correlation coefficient was measured at $0.67$, indicating a moderate relationship, sufficient to provide auxiliary guidance but not dominant enough to overshadow the country token's influence. Removing the Country Adapter Module (w/o CAM) causes a more substantial drop of $2.39\%$ in accuracy and drop of $1.89\%$. This underscores the critical role

Table 2: Comparison of performance across different DGMs on MACP. The unit of CountryReward is percentage (%).

| Model | Metric | Accuracy | BR | CL | ES | FR | KR | JP | US | MX | AU | SA |
|---|---|---|---|---|---|---|---|---|---|---|---|---|
| DGM (w/o RL) | CountryReward | 56.04 | 54.03 | 57.05 | 51.67 | 55.40 | 53.59 | 60.40 | 57.64 | 54.36 | 56.38 | 59.89 |
| DGM | CountryReward | **59.60** | **58.36** | **60.10** | **54.66** | **57.31** | **56.22** | **61.76** | **60.28** | **57.33** | **68.26** | **61.71** |

of dynamically modulating visual features based on country embeddings for adapting to local visual preferences. Removing the Focus-Driven Penalty Loss (w/o FDPL) results in a 3.42% accuracy decrease and a 3.37% sensitivity decrease. This demonstrates that explicitly penalizing the model for under-attending to critical tokens during errors is an effective regularization strategy. The cumulative effect of all three modules is clear, as their removal (CountryReward-w/o-Modules) results in a significantly lower accuracy (55.60%) and lower sensitivity (36.73%).

**Performance per Country.** Table 1 shows the accuracy breakdown for each country. CountryReward achieves more balanced and higher performance across all countries compared to baselines. The variances of Qwen2-VL-7B with FC Head and CountryReward are 7.12% and 8.18% respectively. CountryReward's specialized components obtain more robust adaptation to diverse markets.

## 4.3 ANALYSIS ON COUNTRY-ADAPTED BACKGROUND DESIGN GENERATION

**RFrLS Q3: Evaluation Metric.** We use CountryReward to evaluate the performance of our Country-Adapted Background Design Generation. We use accuracy as the evaluation metric. It measures the proportion of correct predictions. Specifically, each test sample in the MACP dataset contains a product, its attributes, a target country, and a ground-truth preference image ($I_{gt}$). We use the Design Generation Model (DGM) to produce a predicted background ($d_{pred}$). The design is then fed into the controlled T2I pipeline to generate the corresponding advertising image $I_{pred}$. The fixed, pre-trained CountryReward model is then invoked to evaluate the pair ($I_{gt}$, $I_{pred}$) conditioned on the same product and country context from the original sample. CountryReward outputs a binary prediction $\hat{y} \in \{\text{gt}, \text{pred}\}$ indicating its inferred preference. The accuracy is calculated as: Accuracy $= \frac{1}{N} \sum_{i=1}^{N} \mathbb{I}[\hat{y}_i = y_i]$, where $N$ is the number of test samples, $y_i$ denotes the target label pred, and $\mathbb{I}[\cdot]$ is the indicator function. A prediction is counted as correct only when the model's prediction favors the predicted image ($I_{pred}$), implying that the model is considered effective solely when the image it generates surpasses the corresponding ground-truth (human-preferred) image.

**Quantitative Results.** Table 2 shows images generated using our method achieve a substantially higher CountryReward score across all tested countries. This indicates that the optimized DGM produces background designs that lead to images better aligned with country-specific preferences.

**Case Study.** Figure 4 in the appendix presents a case study for two products for five targeted country markets. This qualitative analysis demonstrates our method's capability to produce highly customized visual content that aligns with the preferences of diverse global markets. These results demonstrate our model's ability to capture nuanced, country-specific visual preferences, validating its effectiveness in mitigating SCVO and enabling tailored content generation for global markets.

## 5 CONCLUSION

This work identifies and addresses the Sparse Critical Variable Overwhelm (SCVO) problem in VLMs, where models fail to respond to instruction-critical variables that are sparse in the input space. We propose a novel training framework that effectively mitigates SCVO through integrated components, including retrieval augmentation, a country adapter module, and a focus-driven penalty loss. Evaluated on the newly introduced MACP dataset, our resulting CountryReward model demonstrates significant improvements in cross-country preference prediction accuracy. Furthermore, we showcase its practical utility by employing it as a reward signal to optimize background design generation for targeted markets. This study provides a foundation for enhancing sensitivity to critical but sparse variables in multimodal reward models.

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

## A ETHICS STATEMENT

We confirm that this work adheres to the ethical guidelines and principles outlined in the ICLR 2026 Code of Conduct. We have conducted a careful review of the potential societal impacts of our work. To the best of our knowledge, we do not foresee our research being directly used for malicious purposes or contributing to significant negative societal consequences. All data used in this study comply with applicable legal and ethical standards. We are committed to conducting research in a responsible and ethical manner and will continue to monitor the implications of our work.

## B REPRODUCIBILITY STATEMENT

We confirm that the methodology presented in this paper is fully reproducible. To support transparency and facilitate further research, we will publicly release all data and source code used in our experiments upon acceptance of the paper. The code repository includes detailed instructions for environment setup, training, and evaluation to ensure easy replication of our results.

## C LLM DISCLAIMER

We acknowledge the use of Large Language Models (LLMs) in the preparation of this manuscript. Specifically, DeepSeek DeepSeek-AI et al. (2025) was used solely for two purposes: (1) to assist in literature review by summarizing existing research and identifying relevant papers, and (2) to polish the text for improved fluency and readability. All ideation, theoretical development, experimental design, data analysis, and result interpretation were conducted solely by the authors. The authors take full responsibility for the content, accuracy, and originality of the work presented herein.

## D CASE STUDY

This case study investigates product advertising adaptation across multi-country markets by generating location-specific marketing imagery for two products. A compact blue car and a pair of white sneakers are shown in the Figure, across five distinct countries: France (FR), Korea (KR), Brazil (BR), Spain (ES), and the United States (US). This qualitative analysis demonstrates our framework's capability to produce highly customized visual content that aligns with the nuanced aesthetic preferences of diverse global markets, such as Parisian architecture for FR, traditional wooden interiors for KR, tropical coastal vistas for BR, Mediterranean urban textures for ES, and iconic desert or coastal landscapes for US. The study highlights the role of multimodal generative AI in scalable, location-aware marketing design, paving the way for automated, globally distributed visual campaigns that remain sensitive to regional identity and consumer expectations.

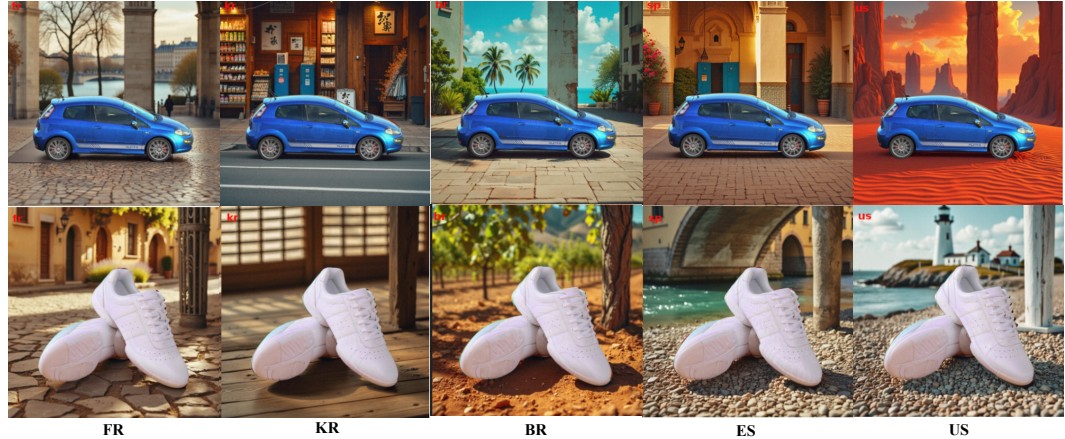

Figure 4: Case Study for two products across five distinct countries.

**Rhwo5 W3:** Most errors of CountryReward are attributed to extreme visual similarity. When the two candidate images (A and B) are visually nearly identical, the model struggles to discern a preference. As shown in Figure 5, the third and fourth columns in the first row have very similar backgrounds. The images in the first and second columns of the second row also share very similar backgrounds. These findings are valuable for future work, suggesting avenues like incorporating finer-grained visual difference detection.

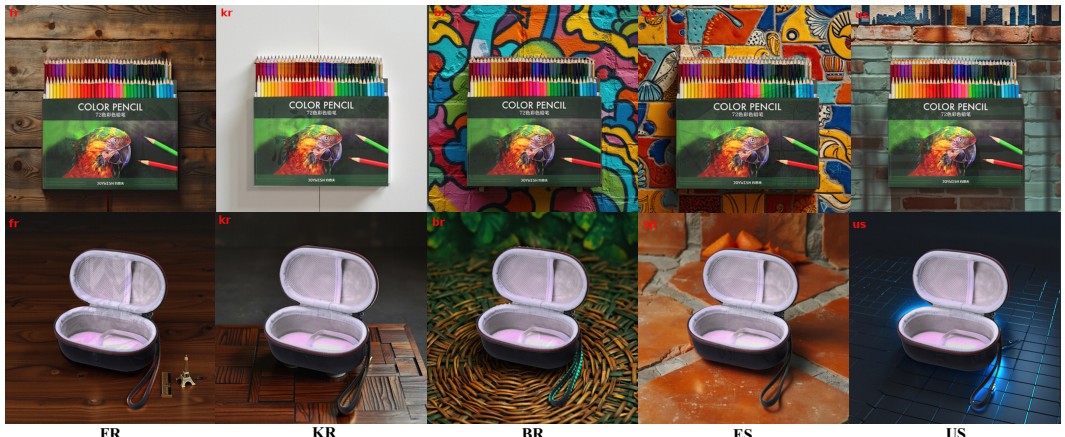

Figure 5: Bad Case Study for two products across five distinct countries.

# E    DETERMINATION OF THE OPTIMAL K-VALUE IN CC-RAG

To determine the optimal k value for top-k retrieval in our CC-RAG system, we employ a decay analysis method based on the cumulative attenuation contribution rate. Specifically, we compute the average cumulative decay of similarity scores across ranking positions from a large-scale retrieval experiment. The k value is set at the point where the cumulative decay contribution rate exceeds a threshold of 80%, indicating that including more results beyond this point yields diminishing returns. This data-driven approach ensures that we capture the majority of relevant information while maintaining efficiency.

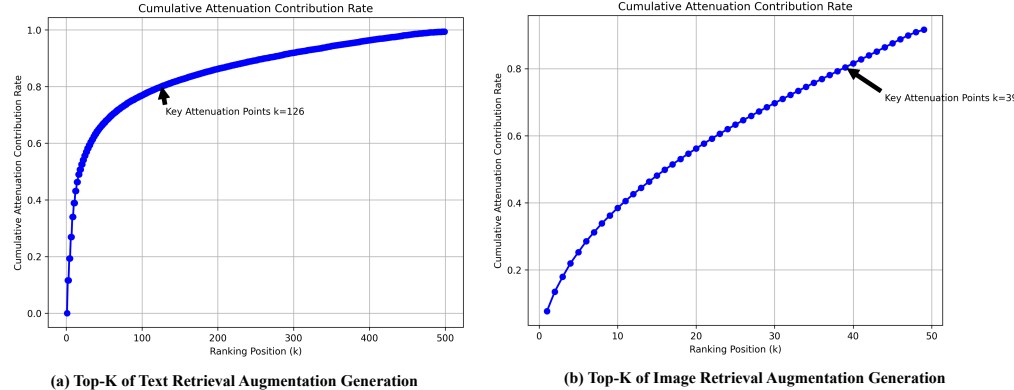

Figure 6: The determination of the optimal K-value.

## F    IMPLEMENTATION DETAILS

For CountryReward, we employ the Qwen2-VL-7B (Wang et al., 2024) as our foundation model. In the CC-RAG process, the top-k values are 127 and 39 in the text and image retrieval stages, respectively. This training phase takes about 20 hours to complete. All experiments are conducted on a machine equipped with 8 NVIDIA A100 GPUs. To optimize training performance, DeepSpeed and FlashAttention-2 are adopted. We use a per-device batch size of 8, gradient accumulation steps of 2, learning rates of $1e-5, 5e-6, 1e-4, 1e-4$ for the projector, LLM, country adapter, and classification head, respectively, a cosine learning rate schedule, and 3 epochs with BF16 mixed-precision enabled. The $\lambda$ is set to $0.1$ in FDPL. For our T2I generation model, we use Stable Diffusion XL (Podell et al., 2023), enhanced with ControlNet (Zhang et al., 2023).

**RZ86t W6&Q3:** For our RL training, we used Direct Preference Optimization (DPO) to finetune the VLM (Qwen2-VL-7B) using CountryReward as the preference signal. For each product, the DGM generates two background designs $d_A$ and $d_B$. Using the T2I model, we create images $I_A$ and $I_B$, which are scored by CountryReward. The design with the higher reward is treated as the preferred ($d^+$) and the other as rejected ($d^-$). The DPO loss uses a $\beta$ (regularization parameter) value of $0.05$. The policy model (DGM$_\theta$) was trained for 1 epoch using the AdamW optimizer with a learning rate of $5e-6$ for the LLM components and $1e-5$ for the multimodal projector. Training used a per-device batch size of 2 and gradient accumulation over 16 steps, resulting in an effective batch size of 32. Max sequence length is 8192 tokens. LR scheduler is cosine with $10\%$ warmup. Training was conducted on 8 NVIDIA A100 GPUs using DeepSpeed ZeRO-3. We ran 1 RL epoch, which involves prompt generation $->$ image generation $\rightarrow$ reward scoring $\rightarrow$ DPO fine-tuning. We will release all training scripts, configuration files, and the final finetuned model weights upon acceptance to ensure full reproducibility.

## G    AUGMENTATION STRATEGY IN CC-RAG

Table 3: Comparison of accuracy performance across different augmentation strategies on MACP. Higher is better for both accuracy and sensitivity, and their units are percentage (%).

| Model | Accuracy | Sensitivity | BR | CL | ES | FR | KR | JP | US | MX | AU | SA |
|---|---|---|---|---|---|---|---|---|---|---|---|---|
| Qwen2-VL-7B (with FC Head) | 55.60 | 36.73 | 53.69 | 56.57 | 50.86 | 54.41 | 53.07 | 59.10 | 56.94 | 53.75 | 58.47 | 59.21 |
| Qwen2-VL-7B (with Instruct RAG) | 54.58 | 36.49 | 52.41 | 54.10 | 50.34 | 53.76 | 51.86 | 57.81 | 55.74 | 52.51 | 58.36 | 58.94 |
| Qwen2-VL-7B (with Embedding RAG) | 55.69 | 37.23 | 53.54 | 56.70 | 51.45 | 54.37 | 52.98 | 58.68 | 57.20 | 53.97 | **58.99** | 58.98 |
| Qwen2-VL-7B (with Scaled Embedding RAG) | **56.97** | **38.87** | **53.79** | **57.28** | **52.58** | **55.22** | **54.52** | **60.13** | **57.73** | 54.39 | 56.05 | **59.95** |

Our investigation focuses on the effective incorporation of augmented answers (from Figure 2(a)) into CountryReward. We evaluate two paradigms (Table 3): instruction-based injection ("Qwen2-VL-7B with Instruct RAG") and embedding-based addition ("Qwen2-VL-7B with Embedding

RAG") of the answer features to the discriminative features. Since both methods yielded inferior results to the baseline, we subsequently scaled the augmented text features to mitigate potential magnitude mismatches with the discriminative features.

$$\mathbf{E}_{\text{cls}} = \mathbf{h}_{\text{dis}} + \left( \frac{\|\mathbf{h}_{\text{dis}}\|_2}{\|\mathbf{e}_{\text{aug}}\|_2} \right) \cdot \mathbf{e}_{\text{aug}} \tag{19}$$

where $\mathbf{E}_{\text{cls}}$ is used to feed into classification head, $\mathbf{e}_{\text{aug}}$ is the extracted text embedding of augmented answers, and $\mathbf{h}_{\text{dis}}$ is the hidden states of the last token from the VLM.

## H  REAL-WORLD VALIDATION THROUGH ONLINE A/B TESTING

**RZ86t W9& Rhwo5 W2:** It is crucial to validate the practical impact of our method using real-world user behavior data. To address this and eliminate potential concerns of circular evaluation, we conducted a large-scale online A/B test on a major cross-border e-commerce platform. The test was designed to compare the performance of product images generated using our optimized Design Generation Model (DGM) against those generated by the baseline DGM (without RL fine-tuning). For a consistent product set spanning the 10 countries in our MACP dataset, two advertising image variants were created: one using the background design from our RL-optimized DGM and the other from the baseline DGM. These image pairs were then served randomly to users in their respective target countries. The test ran for one week, accumulating over 6 million impressions in total.

The primary evaluation metric was the Click-Through Rate (CTR), defined as the number of clicks divided by the number of impressions for each image variant. A higher CTR indicates that an image is more effective at attracting user engagement in a real-world scenario. The results, summarized in Table 4, demonstrate consistent and statistically significant improvements in CTR for images generated by our method across all countries. The average CTR improvement was 1.1%, with gains ranging from 0.7% to 1.4% depending on the market.

These results provide strong external validation that our CountryReward-optimized framework generates advertising images that are significantly better aligned with country-specific user preferences, leading to higher engagement. This real-world evidence substantiates the practical utility of our approach in mitigating the SCVO effect and enhancing cross-market visual content generation.

Table 4: CTR improvements of generated images using our optimized DGM vs. the baseline DGM across countries.

| Country | BR | CL | ES | FR | KR | JP | US | MX | AU | SA |
|---|---|---|---|---|---|---|---|---|---|---|
| CTR Improvement (%) | 1.3 | 1.2 | 1.0 | 0.8 | 1.4 | 1.3 | 1.1 | 0.8 | 1.1 | 0.7 |

# I DISCUSSION ON MODEL DEPLOYMENT

**RFrLS Q1:** A crucial aspect of the proposed Cross-Country Retrieval-Augmented Generation (CC-RAG) module is its deployment strategy and associated computational cost, particularly concerning whether it operates online during inference. To clarify, CC-RAG is primarily employed as an offline, pre-computation component. During the training phase, the augmented choice $\hat{y}_{\text{aug}}$ for every sample in the training set is pre-computed offline prior to the commencement of model training. These static, pre-computed augmentations are then integrated into the training process of CountryReward, enriching the model's exposure to market-aligned preferences without incurring retrieval latency during gradient updates.

For the inference or deployment phase, our framework adopts a dual-strategy approach centered on offline-first efficiency. In the standard and recommended deployment setting, CC-RAG is also employed as an offline, pre-computation component. The pre-computed augmentations are then integrated into the inference process of CountryReward. However, the architecture retains the flexibility for online retrieval. When utilizing CC-RAG for online retrieval during inference, processing each sample takes 76 ms, while the offline version delivers faster performance. This makes CC-RAG well-suited for large-scale production environments.

# J GENERALIZATION EXPERIMENTS

**RgJK1 W1:** The generalizability of the Sparse Critical Variable Overwhelm (SCVO) problem is crucial. Below, we address these concerns by: (1) clarifying the fundamental nature of SCVO as a general architectural issue in VLMs, (2) presenting new experimental evidence on a different domain and attributes, and (3) explaining how our framework inherently generalizes.

## J.1 THE THEORETICAL BASIS FOR SCVO AS A GENERAL CHALLENGE

The core of the SCVO problem is not an artifact of the advertising domain but stems from a fundamental characteristic of autoregressive VLMs. Their probability chain rule, $P(\mathbf{x}) = \prod_t P(x_t|x_{<t})$, coupled with attention-based feature fusion, creates a statistical bias towards high-volume, frequently co-occurring tokens (e.g., detailed product descriptions, hundreds of image-patch tokens). In contrast, instruction-critical variables that are sparse in the input sequence (e.g., "country," "target user age," "target user profession") are easily overwhelmed during this fusion process. The advertising preference task, where "country" is a perfectly sparse yet decisive cue, serves as a clear and controlled manifestation of this underlying general architectural bias. Consequently, the mitigation principles embedded in our framework, including reinforcing sparse variables with external knowledge (CC-RAG), dynamically modulating representations based on them (CAM), and penalizing their neglect (FDPL), are designed to be general mechanisms applicable beyond the specific case of "country."

## J.2 EXPERIMENTAL SETUP ON IMAGE QUALITY ASSESSMENT

We leverage the ImageGen-CoT-Reward-5K dataset (Wang et al., 2025a) from the UnifiedReward-THINK. This dataset involves evaluating pairs of AI-generated images across multiple quality dimensions (Semantic Consistency, Aesthetics, Authenticity) using a chain-of-thought (CoT) reasoning process. To create an SCVO-style evaluation, we systematically reformulate this dataset:

Specifically, we transform the ImageGen-CoT-Reward-5K dataset from the UnifiedReward study into an SCVO-style task. In this transformed dataset, the sparse critical variable is the evaluation dimension (e.g., "Semantic Consistency", "Aesthetics", or "Authenticity"), while all other inputs (images, captions, instruction structure) remain identical. The instruction is framed as: "For [Dimension], the caption of the image is [caption]. Given two images, Image A and Image B, which one is better?" This setup mimics the SCVO scenario where a sparse textual cue dictates the preference judgment.

The final processed dataset contains 12,811 training and 5,517 test samples, spanning the three evaluation dimensions. We compare our adapted CountryReward framework against a strong baseline:

a UnifiedReward model finetuned directly on this transformed dataset. This baseline represents a state-of-the-art general-purpose multimodal reward model adapted to the same data.

## J.3 RESULTS AND ANALYSIS

As shown in Table 5, our adapted CountryReward model achieves an accuracy of 67.9% and a sensitivity of 24.0%, significantly outperforming the finetuned UnifiedReward baseline (accuracy: 62.7%, sensitivity: 1.9%). This represents an improvement of over 5.2% in absolute accuracy and a 22.1% increase in sensitivity. This demonstrates that the SCVO problem is not domain-specific and that our framework generalizes well to other tasks, such as image quality assessment. The key adaptation lies in identifying the sparse critical variable for the new task and configuring the retrieval augmentation (CC-RAG) and feature modulation (CAM) components accordingly. The Focus-Driven Penalty Loss (FDPL) remains a universally applicable regularization technique.

## J.4 SCALABILITY TO MULTIPLE SPARSE VARIABLES

Given the potential complexity of instructions that contain multiple sparse critical variables (e.g., "For a [young person] in [France], which image is better?"), our framework is inherently scalable to such scenarios:

- The Focus-Driven Penalty Loss (FDPL) can naturally be extended to calculate separate penalties for inattention to multiple specified critical tokens (e.g., "young person" and "France"), encouraging the model to attend to all simultaneously.
- For highly complex, multi-factorial decisions, an ensemble approach could be employed. Specialized reward models (e.g., AgeReward, CountryReward), each finetuned to be sensitive to a specific type of sparse variable using our framework, could provide intermediate rewards. These signals could then be aggregated for a final decision via an ensemble strategy.

More effective models for generalizing to multiple sparse variables could be proposed in the future.

We acknowledge that comprehensively benchmarking SCVO across numerous domains and variables is a substantial endeavor. To our knowledge, this is the first work to identify and formalize this problem; our primary contribution is to establish its existence and propose a foundational solution framework. We are committed to expanding this line of research and are actively planning to develop a comprehensive benchmark to systematically evaluate SCVO across diverse tasks in the future. We will open-source the code and datasets to foster research on this critical issue.

In conclusion, while our current empirical focus is on the "country" variable in advertising, the SCVO problem is rooted in a fundamental architectural characteristic of VLMs. Our solution framework provides general, adaptable mechanisms to address it, supported by preliminary cross-attribute validation. We believe this work opens a crucial research direction toward building more robust and context-sensitive multimodal models.

Table 5: SCVO Generalization to Image Quality Assessment Task (%)

| Model | Sensitivity | Accuracy | Semantic Consistency | Aesthetics | Authenticity |
|---|---|---|---|---|---|
| Baseline | 1.9 | 62.7 | 63.5 | 61.4 | 63.2 |
| Our Model | 24.0 | 67.9 | 65.6 | 67.8 | 70.4 |

# K  BASELINE COMPARISONS

**Rhwo5 W1: Domain-Adaptive SFT:** We directly compared CountryReward with supervised fine-tuning (SFT) reward models on our MACP dataset. Table 6 shows that CountryReward (accuracy: 60.4%, sensitivity: 40.8%) substantially outperforms the Domain-Adaptive SFT baseline (accuracy: 44.6%, sensitivity: 20.8%). UnifiedReward is essentially a model finetuned on many existing general-purpose reward model datasets, without any changes to the model architecture. To reduce the undesirable interactions between these datasets and MACP, we reproduce UnifiedReward (Wang et al., 2025c) by performing supervised fine-tuning on Qwen2-VL-7B using the MACP dataset. In other words, UnifiedReward is equivalent to Domain-Adaptive SFT in Table 6. This indicates that specialized mechanisms in CountryReward, such as the Cross-Country Retrieval-Augmented Generation, Country Adapter Module, and Focus-Driven Penalty Loss, are crucial for handling SCVO, whereas general-purpose models lack such capabilities.

**Rhwo5 W1& RZ86t W5&S3: Prompt Engineering:** We crafted a strong instructional prompt that explicitly rearranges information to foreground the country variable: "For the [Country] market, for the product: [Attributes], compare the two e-commerce images below for the [Country] market and select the one (A/B) that is more likely to attract consumers in the [Country] market?" As shown in Table 6, although higher than the variant without the rearrangement and repetition prompt method (Qwen2-VL-7B (with SFT)), this baseline achieved only 49.77% accuracy (Qwen2-VL-7B (with SFT & Prompt Engineering)), indicating that sophisticated prompting alone cannot resolve the underlying representational imbalance causing SCVO.

**RFrLS W1: FiLM-style Conditioning Baseline:** This baseline corresponds directly to applying only the Country Adapter Module (CAM) from our framework. It modulates the visual features extracted by the vision encoder using affine transformation parameters generated from textual country embeddings, following the FiLM paradigm (Perez et al., 2017). This tests the sufficiency of dynamic visual feature conditioning based on sparse textual cues. The FiLM-style baseline (CountryReward with only CAM) achieves an accuracy of 55.92% and a sensitivity of 37.18%. While this represents a slight improvement over the base Qwen2-VL-7B model with a standard FC head (55.60% accuracy), the gain is marginal. This indicates that simply modulating visual features based on country, while beneficial, is insufficient to robustly overcome the overwhelming influence of dominant, high-volume variables in the input sequence.

**RFrLS W1: Attention-imbalance Baseline:** This baseline implements a standard attention-balancing regularization. It utilizes only the Focus-Driven Penalty Loss (FDPL) component with the Qwen2-VL-7B backbone. The loss penalizes the model based on the relative attention (approximated by the L2 norm of hidden states) allocated to critical tokens (country, product, image) when a prediction error occurs, aiming to mitigate attention overshadowing. The attention-imbalance baseline (CountryReward with only FDPL) reaches a modest improvement to 56.04% accuracy and 37.12% sensitivity. Explicitly penalizing low attention on critical tokens provides a modest boost but fails to fully resolve the prediction collapse across countries.

**RZ86t W5&S3: Token-Importance Re-Weighting:** We implement a baseline that re-weights the loss only based on country token importance (approximated by gradient norms). While it showed a slight improvement (55.87% accuracy) over the standard FC Head baseline (55.60%), its performance gain was limited and less stable across countries compared to our integrated approach. This suggests that simply re-weighting the loss is insufficient to address the core representational imbalance in SCVO.

**RZ86t W5&S3: Focal-Loss Discriminator:** This baseline adds a focal loss discriminator based on the Qwen2-VL-7B (with FC Head). Our experiments show that while focal loss slightly improves accuracy (58.10%) over the standard FC head (55.60%), it still falls short of CountryReward (60.37%) and exhibits significantly lower sensitivity (12.02% vs. 40.84%). This indicates that simply re-weighting the loss is insufficient to address the core representational imbalance in SCVO, as it fails to enhance cross-country consistency.

**RZ86t W5&S3: Logistic Regression:** This baseline is a classic CTR-prior logistic regression model with comprehensive feature engineering, including: Text features (TF-IDF vectors from product titles), Categorical features (One-hot encodings for country and category), Interaction features (Explicit country $\times$ category interactions to capture market-specific preferences), and Image fea-

tures (Differences between image embeddings of candidates A and B). This baseline now serves as a robust non-neural benchmark, directly modeling the problem of CTR prediction. Results show it achieves $52.28\%$ accuracy, significantly below our method ($60.37\%$), confirming that naive feature-based approaches fail to resolve SCVO without structured multimodal reasoning.

**RZ86t W5&S3: Gradient Boosted Decision Tree:** This baseline is a classic CTR-prior GBDT baseline with country×category interaction features, representing a strong traditional approach for preference prediction. This baseline incorporates: Text features (TF-IDF of item titles), Categorical features (one-hot encoded country and category), Country×category interaction features (both one-hot and statistical features), and Image embedding differences between candidate images. The results demonstrate that our CountryReward model substantially outperforms this strong non-neural baseline ($60.37\%$ vs $55.39\%$ accuracy), validating that our approach captures complex multimodal interactions beyond what traditional methods can achieve.

**RZ86t W5&S3: Country Token Mask:** This baseline is a causal intervention experiment where the country token is explicitly masked (replaced with [MASK]) in the input instruction. This tests the model's dependence on the sparse critical variable (country name). Masking the country token led to a significant accuracy drop (from $60.37\%$ to $57.25\%$) and a drastic sensitivity reduction (from $40.84\%$ to $12.26\%$). This demonstrates that CountryReward successfully grounds its predictions in the country variable, and its performance is causally tied to this sparse token. This ablation directly validates SCVO as a failure mode and shows that our framework restores sensitivity to the critical variable.

**RZ86t W5&S3:** We performed paired t-tests between CountryReward and all baselines. The improvements of CountryReward are statistically significant ($p < 0.01$) in all cases, confirming that our method consistently outperforms alternatives. One of the examples is shown in Section O. These additions reinforce the validity of our conclusions and demonstrate that CountryReward effectively mitigates SCVO with high statistical reliability.

Table 6: Comparison of accuracy performance across different baselines on MACP. Higher is better for both accuracy and sensitivity, and their units are percentage (%).

| Model | Accuracy | Sensitivity | BR | CL | ES | FR | KR | JP | US | MX | AU | SA |
|---|---|---|---|---|---|---|---|---|---|---|---|---|
| Logistic Regression | 52.28 ± 0.23 | 14.69 | 51.30 ± 0.73 | 52.46 ± 0.73 | 51.17 ± 0.73 | 52.87 ± 0.73 | 51.42 ± 0.73 | 53.99 ± 0.73 | 53.03 ± 0.73 | 52.02 ± 0.73 | 54.35 ± 0.73 | 53.17 ± 0.73 |
| Gradient Boosted Decision Tree | 55.39 ± 0.23 | 15.37 | 54.87 ± 0.73 | 55.39 ± 0.73 | 55.05 ± 0.73 | 55.56 ± 0.72 | 55.24 ± 0.73 | 56.07 ± 0.72 | 55.42 ± 0.73 | 55.05 ± 0.73 | 55.75 ± 0.72 | 55.51 ± 0.72 |
| Qwen2-VL-7B (with Domain-Adaptive SFT) | 44.61 ± 0.23 | 20.82 | 46.96 ± 0.73 | 43.71 ± 0.73 | 49.76 ± 0.72 | 46.15 ± 0.72 | 47.78 ± 0.70 | 40.99 ± 0.71 | 42.99 ± 0.73 | 46.66 ± 0.71 | 40.92 ± 0.73 | 40.16 ± 0.70 |
| Qwen2-VL-7B (with SFT & Prompt Engineering) | 49.77 ± 0.23 | 14.39 | 49.75 ± 0.72 | 48.34 ± 0.72 | 50.43 ± 0.72 | 49.17 ± 0.72 | 49.02 ± 0.72 | 50.01 ± 0.72 | 52.10 ± 0.71 | 49.94 ± 0.72 | 49.74 ± 0.72 | 49.23 ± 0.72 |
| Qwen2-VL-7B (with FC Head) | 55.60 ± 0.23 | 36.73 | 53.69 ± 0.73 | 56.58 ± 0.72 | 50.86 ± 0.73 | 54.41 ± 0.73 | 53.07 ± 0.73 | 59.10 ± 0.72 | 56.94 ± 0.72 | 53.75 ± 0.73 | 58.47 ± 0.72 | 59.21 ± 0.71 |
| Qwen2-VL-7B (with Focal Loss) | 58.10 ± 0.22 | 12.02 | 58.34 ± 0.71 | 59.84 ± 0.70 | 54.97 ± 0.71 | 58.55 ± 0.70 | 58.00 ± 0.71 | 61.88 ± 0.69 | 59.86 ± 0.70 | 56.68 ± 0.71 | 61.58 ± 0.69 | 55.24 ± 0.71 |
| CountryReward with only CAM (FiLM-style) | 55.92 ± 0.23 | 37.18 | 53.71 ± 0.73 | 56.27 ± 0.72 | 52.38 ± 0.73 | 54.89 ± 0.73 | 53.39 ± 0.73 | 58.67 ± 0.72 | 56.88 ± 0.72 | 53.56 ± 0.73 | 59.10 ± 0.72 | 58.64 ± 0.72 |
| CountryReward with only FDPL (Attention-imbalance) | 56.04 ± 0.23 | 37.12 | 53.76 ± 0.73 | 56.06 ± 0.72 | 52.48 ± 0.73 | 54.47 ± 0.73 | 53.36 ± 0.73 | 57.78 ± 0.72 | 56.32 ± 0.72 | 53.68 ± 0.73 | 58.65 ± 0.72 | 59.48 ± 0.72 |
| CountryReward (with Re-Weighting) | 55.87 ± 0.23 | 36.91 | 58.73 ± 0.72 | 58.93 ± 0.71 | 59.61 ± 0.73 | 53.52 ± 0.72 | 53.95 ± 0.70 | 57.08 ± 0.74 | 51.37 ± 0.73 | 54.28 ± 0.71 | 57.01 ± 0.72 | 54.19 ± 0.73 |
| CountryReward (with Country Token Mask) | 57.95 ± 0.23 | 12.26 | 55.62 ± 0.72 | 59.33 ± 0.72 | 53.63 ± 0.73 | 56.78 ± 0.72 | 55.08 ± 0.73 | 60.75 ± 0.71 | 59.86 ± 0.71 | 56.19 ± 0.72 | 61.22 ± 0.71 | 61.06 ± 0.71 |
| **CountryReward (Ours)** | **60.37 ± 0.23** | **40.84** | **58.70 ± 0.72** | **61.12 ± 0.71** | **56.33 ± 0.72** | **59.38 ± 0.72** | **57.88 ± 0.72** | **63.54 ± 0.70** | **61.82 ± 0.71** | **58.72 ± 0.72** | **63.30 ± 0.70** | **62.93 ± 0.72** |

## L  MECHANISM ANALYSIS

**RZ86t W3&Q5&S2:** To move beyond a phenomenological description and establish a rigorous mechanistic foundation for the Sparse Critical Variable Overwhelm (SCVO) concept, we conducted a series of analytical experiments. This section delineates how SCVO is distinct from related challenges and provides empirical evidence that traces the failure mode from its root cause to our proposed solution through attention analysis, gradient attribution, and causal interventions.

### L.1  DISTINCTION FROM RELATED CONCEPTS

**Compared with "Lost-in-the-Middle":** This primarily concerns the positional degradation of information retrieval in long documents. SCVO is agnostic to position; even if the country name is placed at the very beginning, its sparse token count relative to the visual and product descriptive tokens leads to its influence being overwhelmed in the fused representation used for the final prediction.

**Compared with "Modality-Volume Imbalance":** While related, prior work often focuses on general alignment or representation fusion between modalities (e.g., text vs. vision). SCVO pinpoints a more specific failure mode within a multimodal decision-making task: a specific, low-volume textual cue (the country) is overwhelmed by the combined volume of other textual cues (product at-

tributes) and a high-volume modality (image patches), rendering the model insensitive to a variable that critically determines the ground-truth outcome.

## L.2 Phenomenon: Sparse Critical Variable Overwhelm (SCVO)

SCVO manifests as a systematic failure in VLMs where sparse but critical instructional variables (e.g., country names in multi-country preference prediction) are overwhelmed by dominant high-volume variables (e.g., image patches or product attributes). In our MACP dataset, baseline models like Qwen2-VL exhibit prediction collapse (e.g., consistently outputting "A" regardless of country), resulting in low accuracy and sensitivity. This occurs despite clear ground-truth preference variations across countries, indicating that the model ignores critical sparse cues.

## L.3 Mechanism: Breaking the Chain Rule via Attention and Gradient Analysis:

We performed three core analyses comparing a baseline VLM (Qwen2-VL-7B with an FC Head) against our CountryReward model to uncover the mechanistic underpinnings of SCVO.

### L.3.1 Attention Map Analysis Reveals Neglect of Sparse Critical Tokens

We computed the average attention weight assigned to the sparse country token (e.g., "US") in the final transformer layer.

- Baseline VLM: The country token received a negligible mean attention weight of $0.05 \pm 0.01$.
- CountryReward: The attention weight increased significantly to $0.08 \pm 0.03$.

The baseline model fails to allocate sufficient computational focus to the critical sparse variable. The Country Adapter Module (CAM) and Focus-Driven Penalty Loss (FDPL) in CountryReward successfully recalibrate the attention mechanism, directly mitigating the "overwhelm" by strengthening the chain-rule link $P(x_t|x_{<t})$ for the country token.

### L.3.2 Gradient Attribution Analysis Confirms Underlying Insensitivity

Using Integrated Gradients, we quantified the contribution of the country token to the final prediction output.

- Baseline VLM: The country token contributed merely $6\%$ to the total gradient norm.
- CountryReward: Its contribution rose to $25\%$.

This demonstrates that in standard VLMs, the gradient signal for the sparse critical variable is vanishingly weak during training, preventing effective learning of its importance. Our framework amplifies this signal, ensuring the model's parameters become sensitive to the country variable.

### L.3.3 Intervention Experiments Establish Causal Link

We performed a causal intervention by ablating (masking) the country token from the input. Table 7 shows the results of intervention experiments.

- Baseline VLM: Ablation caused a negligible performance drop (sensitivity: from $36.73\%$ to $31.48\%$), confirming its predictions were already invariant to the country.
- CountryReward: Ablation led to a substantial decrease (sensitivity: from $40.84\%$ to $12.26\%$).

This experiment establishes a causal link. The country token becomes a decisive factor for CountryReward's decision-making, whereas it is functionally ignored by the baseline. The sharp drop in sensitivity (which measures cross-country consistency) is particularly telling, proving that CountryReward has learned to meaningfully condition its output on this previously overlooked variable.

### L.3.4 CONTROLLED SYNTHETIC EXPERIMENTS ISOLATE THE SCVO EFFECT:

To isolate the SCVO effect, we created a synthetic variant of our task where we systematically increased the "weight" of the critical variable by repeating the country token in the instruction. The baseline model's performance showed only marginal improvement (accuracy from 44.61% to 49.77%) even as the token count (one country token to five country tokens) for the critical variable increased. This confirms that SCVO is not merely a problem of token count but a fundamental failure in the model's sensitivity and representational integration of sparse critical information.

## L.4 SOLUTION: COUNTRYREWARD FRAMEWORK

Our framework directly counteracts SCVO by reinforcing the model's sensitivity to sparse critical variables through three novel components:

**Cross-Country Retrieval-Augmented Generation (CC-RAG):** Injects market-specific preferences into training, augmenting the sparse country signal with similar product knowledge. This explicitly conditions the chain rule on country-aware data, raising accuracy by 3.56% in ablations.

**Country Adapter Module (CAM):** Dynamically modulates visual features using country embeddings, ensuring that image representations are transformed based on sparse textual cues. This restores the chain rule's dependence on country tokens by aligning visual and textual pathways, contributing 2.39% to accuracy gains.

**Focus-Driven Penalty Loss (FDPL):** Penalizes under-attention to critical tokens during errors, directly optimizing attention distributions via a regularization term. This improves sensitivity by 3.37%.

## L.5 CONCLUSION OF MECHANISM ANALYSIS

These analyses collectively validate the SCVO phenomenon at a mechanistic level through spanning attention distribution, gradient flow, causal intervention, and controlled synthesis. Our CountryReward framework directly counteracts these mechanisms, which completes the "phenomenon → mechanism → solution" loop, providing a solid foundation for the SCVO concept and the efficacy of our approach.

Table 7: Performance Comparison Before and After Country Token Ablation.

| Model | Accuracy | Sensitivity | BR | CL | ES | FR | KR | JP | US | MX | AU | SA |
|---|---|---|---|---|---|---|---|---|---|---|---|---|
| Baseline (Ablated) | 49.81 | 31.48 | 49.82 | 50.31 | 48.79 | 49.21 | 48.25 | 50.63 | 51.22 | 49.23 | 49.80 | 50.75 |
| Baseline (Full) | 55.60 | 36.73 | 53.69 | 56.57 | 50.86 | 54.41 | 53.07 | 59.10 | 56.94 | 53.75 | 58.47 | 59.21 |
| CountryReward (Ablated) | 57.25 | 12.26 | 55.08 | 58.26 | 53.46 | 56.58 | 54.64 | 59.90 | 58.78 | 55.75 | 59.73 | 60.35 |
| CountryReward (Full) | **60.37** | **40.84** | **58.70** | **61.12** | **56.33** | **59.38** | **57.88** | **63.54** | **61.82** | **58.72** | **63.30** | **62.93** |

# M    SYNERGISTIC DESIGN FOR SCVO MITIGATION

**Rhwo5 W1:**The core novelty of our work lies not in the invention of entirely new components, but in their principled and synergistic integration to address the newly identified Sparse Critical Variable Overwhelm (SCVO) problem. The three core modules, Cross-Country Retrieval-Augmented Generation (CC-RAG), Country Adapter Module (CAM), and Focus-Driven Penalty Loss (FDPL), combat SCVO at distinct but complementary levels of the model pipeline, forming a holistic solution.

- Data-level amplification (CC-RAG): By retrieving and injecting click-through preferences aligned with the target market, CC-RAG directly amplifies the signal of the critical country variable. It provides an external, market-aware prior that forces the model to prioritize localized relevance, counteracting the dominance of generic high-volume features.

- Feature-level adaptation (CAM): Operating inside the vision encoder, CAM dynamically modulates visual representations using learned country embeddings. This ensures that the sparse critical variable actively conditions perceptual processing, preventing visual features from being treated uniformly and enabling fine-grained adaptation to market-specific visual preferences.

- Optimization-level regularization (FDPL): During training, FDPL penalizes the model when it makes errors while under-attending to the tokens corresponding to critical variables (country, product, image). It acts as a focus-driven regularizer, explicitly encouraging balanced attention allocation and reinforcing the importance of sparse variables in the decision process.

The synergy is closed-loop: as shown in our experiments (Table 8), CC-RAG provides external country-aware signals, CAM internally aligns visual features with those signals, and FDPL ensures robust attention to all critical components during learning. Ablating any module breaks this loop and leads to a measurable performance drop. We train models using only one of our proposed modules at a time (e.g., CountryReward with only CC-RAG, only CAM, or only FDPL). Their performances (56.01%, 55.92%, and 56.04% accuracy, respectively) are superior to the plain finetuned baseline but substantially worse than the full CountryReward (60.37%). This quantitatively demonstrates that each component contributes, but their combined effect is synergistic and necessary for peak performance.

Table 8: Synergistic Design Study. Higher is better for both accuracy and sensitivity, and their units are percentage (%).

| Model | Accuracy | Sensitivity | BR | CL | ES | FR | KR | JP | US | MX | AU | SA |
|---|---|---|---|---|---|---|---|---|---|---|---|---|
| Qwen2-VL-7B (with FC Head) | 55.60 | 36.73 | 53.69 | 56.57 | 50.86 | 54.41 | 53.07 | 59.10 | 56.94 | 53.75 | 58.47 | 59.21 |
| CountryReward (with only CC-RAG) | 56.01 | 37.10 | 53.88 | 56.93 | 52.01 | 54.78 | 53.79 | 59.16 | 57.49 | 53.82 | 58.96 | 59.22 |
| CountryReward (with only CAM) | 55.92 | 37.18 | 53.71 | 56.27 | 52.38 | 54.89 | 53.39 | 58.67 | 56.88 | 53.56 | 59.10 | 58.64 |
| CountryReward (with only FDPL) | 56.04 | 37.12 | 53.76 | 56.06 | 52.48 | 54.47 | 53.36 | 57.78 | 56.32 | 53.68 | 58.65 | 59.48 |
| CountryReward (w/o CC-RAG) | 56.81 | 37.40 | 54.93 | 57.05 | 53.80 | 55.39 | 54.48 | 59.57 | 57.65 | 55.64 | 59.52 | 60.11 |
| CountryReward (w/o CAM) | 57.98 | 38.95 | 55.50 | 57.85 | 53.78 | 56.75 | 55.19 | 61.47 | 59.39 | 55.98 | 61.71 | 62.21 |
| CountryReward (w/o FDPL) | 56.95 | 37.47 | 54.79 | 57.09 | 52.80 | 56.12 | 54.44 | 61.01 | 58.79 | 55.33 | 58.83 | 60.33 |
| **CountryReward** | **60.37** | **40.84** | **58.70** | **61.12** | **56.33** | **59.38** | **57.88** | **63.54** | **61.82** | **58.72** | **63.30** | **62.93** |

## N  MACP DATASET

### N.1  DATA BALANCE

**Rhwo5 W3 & RZ86t W8&Q4:** Our MACP dataset demonstrates exceptional statistical balance.

**Sample Balance Across Countries:** We constructed the MACP dataset with exceptional statistical balance across countries (see Appendix for detailed tables). As shown in Table 10, the test set contains exactly 18,055 samples per country (10.00% each). The training set distribution ranges from 9.34% to 10.57% per country, with minimal variance (0.18%) from perfect equality. This controlled, balanced environment was designed to validate our core hypothesis: that our method can effectively solve the SCVO problem when the critical country signal is present in the data.

**Country × Product Category Distribution:** We provide detailed cross-tabulation tables showing the distribution of product categories across countries for both training (Table 11) and test (Table 12) splits. The training set maintains consistent category representation across all countries. The test set shows perfect balance with identical product category counts per country, ensuring our evaluation is not biased by category-country interactions.

### N.2  DIVERSITY AND GENERALITY OF THE MACP DATASET

**Rhwo5 W3:** The MACP dataset, while sourced from a single major cross-border e-commerce platform, exhibits considerable diversity that supports its use for developing generally applicable models.

**Product Category Diversity:** As shown in Table 11-12, the dataset spans 9 major product categories (e.g., Beauty & Apparel, Consumer Electronics, Home & Garden, Sports & Entertainment). The distribution of these categories is carefully balanced across all 10 countries (variance $< 0.2\%$ from perfect balance per category), preventing confounding effects between country and product type.

**Geographical and Cultural Coverage:** The 10 countries were strategically selected to cover diverse cultural and economic regions: the Americas (US, BR, MX, CL), Europe (FR, ES), East Asia (JP, KR), the Middle East (SA), and Oceania (AU). No single country dominates the dataset.

**User Base:** While fine-grained user demographics cannot be shared due to privacy policies, the platform serves a global user base numbering in the hundreds of millions. Internal analytics indicate that the user demographics for each country (in terms of age groups, gender distribution, and shopping interests) are representative of national e-commerce trends, suggesting the captured preferences reflect broader consumer behavior.

This combination of product, geographical, and implied user diversity mitigates concerns that our findings are artifacts of a narrow data source and supports the generality of the SCVO problem and our proposed solution.

### N.3  ROBUSTNESS OF CC-RAG

**Rhwo5 W3 & RZ86t W8&Q4:** Our approach, particularly the CC-RAG component, benefits from the availability of click data knowledge. A legitimate concern is its effectiveness in markets with sparse or outdated data. Fundamentally, this highlights a broader principle: model performance is intrinsically linked to data quality and availability. While sophisticated algorithms can extract patterns from data, they cannot create a signal from its absence. For real-world scenarios with severe geographical imbalance, we employ standard mitigation strategies during the training data preparation for CountryReward. These include targeted oversampling of minority countries or categories and strategic undersampling of majority ones to create a more balanced learning foundation. For emerging markets with very limited initial data, we bootstrap knowledge by leveraging data from culturally or economically similar regions. As more market-specific data accumulates, the CC-RAG component becomes increasingly effective and specialized.

Regarding data "freshness," for the MACP dataset, the training and test sets were collected concurrently, eliminating concerns about data being outdated for the purpose of our experiments. We ensure reliability by calculating the Click-Through Rate (CTR) over a fixed 30-day sliding window

prior to the evaluation date. This captures recent user preferences and reduces temporal noise. In a production setting, the knowledge base for CC-RAG is updated periodically (e.g., every 30 days).

### N.4    DATA ETHICS AND AVAILABILITY

**Z86t W4&S4: Data Legality & Privacy Compliance:** The MACP dataset was collected and processed in full compliance with the e-commerce platform's data governance framework and applicable privacy regulations. All data usage has been formally authorized through the platform's research partnership program.

**Anonymization Process:** The dataset has undergone rigorous anonymization: (i) All user identifiers have been removed; (ii) CTR data is aggregated at product-country level; (iii) Personal browsing histories or individual user behaviors are excluded; (iv) Product metadata is limited to publicly available information.

**Geographic Bias Mitigation:** We implemented proactive bias control through stratified sampling across 10 diverse markets and balanced product category representation, as shown in Table 10, as evidenced by our detailed distribution tables (9.34%-10.57% per country in training, exact 10% in testing).

**Data Licensing & Access:** The complete MACP dataset will be publicly released under CC BY-NC 4.0 license upon paper acceptance. The release package includes: (i) Aggregated CTR preferences; (ii) Product images and metadata; (iii) Country-market labels.

**Controlled Access Assurance:** We will release the entire dataset, source code, model configurations, and training scripts to ensure full reproducibility of both CountryReward training and the background generation framework.

### N.5    COUNTRY NAME ABBREVIATIONS

**RZ86t W8:** As shown in the Table 9, a comprehensive country abbreviation table is clearly defined, listing all country codes used in our study.

Table 9: Country name abbreviations

| Abbreviation | Full Name |
| --- | --- |
| BR | Brazil |
| CL | Chile |
| ES | Spain |
| FR | France |
| KR | Korea (Republic of Korea) |
| JP | Japan |
| US | United States |
| MX | Mexico |
| AU | Australia |
| SA | Saudi Arabia |

Table 10: Per-country sample counts for Train and Test splits

| Country | Train Samples | Train % | Test Samples | Test % |
|---|---|---|---|---|
| Australia (AU) | 76934 | 9.34% | 18055 | 10.00% |
| Brazil (BR) | 86832 | 10.55% | 18055 | 10.00% |
| Chile (CL) | 82074 | 9.97% | 18055 | 10.00% |
| Spain (ES) | 85975 | 10.44% | 18055 | 10.00% |
| France (FR) | 80118 | 9.73% | 18055 | 10.00% |
| Japan (JP) | 78289 | 9.51% | 18055 | 10.00% |
| Korea (KR) | 87066 | 10.57% | 18055 | 10.00% |
| Mexico (MX) | 82351 | 10.00% | 18055 | 10.00% |
| Saudi Arabia (SA) | 82478 | 10.02% | 18055 | 10.00% |
| United States (US) | 81274 | 9.87% | 18055 | 10.00% |
| **Total** | 823391 | 100% | 180550 | 100.00% |

Table 11: Country × Product Category distribution for Train split

| Country | AU | BR | CL | ES | FR | JP | KR | MX | SA | US |
|---|---|---|---|---|---|---|---|---|---|---|
| **Watches & Luggage Bags** | 5567 | 6321 | 4547 | 6010 | 5714 | 5809 | 7185 | 5138 | 6814 | 6269 |
| **Beauty & Apparel & Shoes** | 11519 | 13623 | 13320 | 14407 | 13679 | 12107 | 11538 | 15620 | 13049 | 15232 |
| **Consumer Electronics** | 6100 | 7864 | 6613 | 6039 | 5695 | 7099 | 7613 | 6313 | 7043 | 6158 |
| **Home & Garden** | 9365 | 8759 | 11096 | 10796 | 10093 | 8372 | 9281 | 9523 | 8562 | 8837 |
| **Kids & Toys & Hobbies** | 5381 | 5633 | 6358 | 5594 | 5595 | 4817 | 5502 | 5735 | 4289 | 5904 |
| **Sports & Entertainment** | 6329 | 9265 | 7351 | 7960 | 6929 | 8011 | 10094 | 6789 | 5902 | 5316 |
| **Home Appliances & Improvement** | 7903 | 7920 | 7098 | 9328 | 8930 | 6662 | 8222 | 7463 | 9743 | 7521 |
| **Industrial & Technology Tools** | 10632 | 13504 | 10602 | 11016 | 10307 | 10981 | 11989 | 10579 | 12161 | 9713 |
| **Other** | 14138 | 13943 | 15089 | 14825 | 13176 | 14431 | 15642 | 15191 | 14915 | 16324 |
| **Total** | 76934 | 86832 | 82074 | 85975 | 80118 | 78289 | 87066 | 82351 | 82478 | 81274 |

Table 12: Country × Product Category distribution for Test split

| Country | AU | BR | CL | ES | FR | JP | KR | MX | SA | US |
|---|---|---|---|---|---|---|---|---|---|---|
| **Watches & Luggage Bags** | 2407 | 2407 | 2407 | 2407 | 2407 | 2407 | 2407 | 2407 | 2407 | 2407 |
| **Beauty & Apparel & Shoes** | 1577 | 1577 | 1577 | 1577 | 1577 | 1577 | 1577 | 1577 | 1577 | 1577 |
| **Consumer Electronics** | 2395 | 2395 | 2395 | 2395 | 2395 | 2395 | 2395 | 2395 | 2395 | 2395 |
| **Home & Garden** | 2375 | 2375 | 2375 | 2375 | 2375 | 2375 | 2375 | 2375 | 2375 | 2375 |
| **Kids & Toys & Hobbies** | 1087 | 1087 | 1087 | 1087 | 1087 | 1087 | 1087 | 1087 | 1087 | 1087 |
| **Sports & Entertainment** | 1059 | 1059 | 1059 | 1059 | 1059 | 1059 | 1059 | 1059 | 1059 | 1059 |
| **Home Appliances & Improvement** | 2504 | 2504 | 2504 | 2504 | 2504 | 2504 | 2504 | 2504 | 2504 | 2504 |
| **Industrial & Technology Tools** | 2896 | 2896 | 2896 | 2896 | 2896 | 2896 | 2896 | 2896 | 2896 | 2896 |
| **Other** | 1755 | 1755 | 1755 | 1755 | 1755 | 1755 | 1755 | 1755 | 1755 | 1755 |
| **Total** | 18055 | 18055 | 18055 | 18055 | 18055 | 18055 | 18055 | 18055 | 18055 | 18055 |

## O  STATISTICAL SIGNIFICANCE TESTING

**RFrLS W1&Q1:** We conducted rigorous statistical analysis to validate that the performance improvements of our CountryReward model are not due to random chance.

**Bootstrap Hypothesis Testing:** For each country, we performed 10,000 bootstrap resamples (with replacement) of the test set. For each bootstrap sample, we calculated the accuracy difference between CountryReward and the strongest baseline (Qwen2-VL-7B with FC Head). The p-value was computed empirically as the proportion of bootstrap samples. As shown in the Figure 7, all improvements reported in Table 1 are statistically significant with $p < 0.001$.

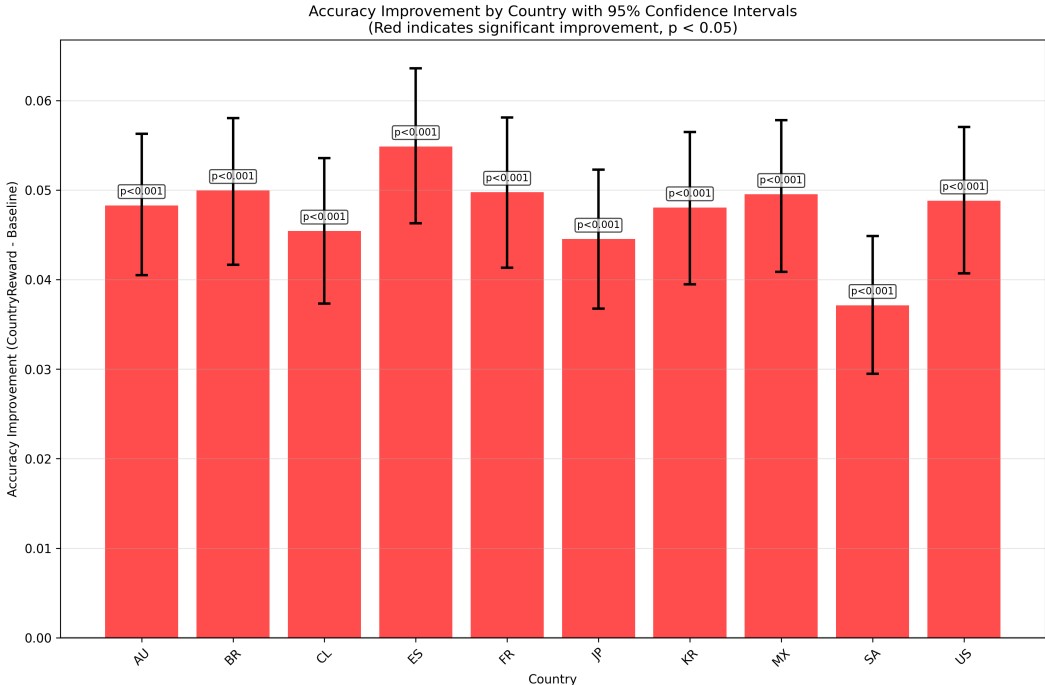

Figure 7: Accuracy Improvement by Country with 95% Confidence Intervals.

**Confidence Intervals:** As shown in Table 13, we report 95% confidence intervals (CIs) for all accuracy estimates using the Clopper-Pearson exact method, which is appropriate for binomial proportions. The table below details the per-country performance of the baseline and CountryReward, including their CIs. The results demonstrate that CountryReward's CIs lie entirely above those of the baseline for all countries, confirming a robust and significant improvement.

Table 13: Model Accuracy Comparison by Country (with 95% Confidence Intervals)

| Country | Sample Size | Baseline Accuracy (95% CI) | CountryReward Accuracy (95% CI) |
|---------|-------------|----------------------------|----------------------------------|
| AU | 18055 | 0.58 (0.58, 0.59) | 0.63 (0.63, 0.64) |
| BR | 18055 | 0.54 (0.53, 0.54) | 0.59 (0.58, 0.59) |
| CL | 18055 | 0.57 (0.56, 0.57) | 0.61 (0.60, 0.62) |
| ES | 18055 | 0.51 (0.50, 0.52) | 0.56 (0.56, 0.57) |
| FR | 18055 | 0.54 (0.54, 0.55) | 0.59 (0.59, 0.60) |
| JP | 18055 | 0.59 (0.58, 0.60) | 0.64 (0.63, 0.64) |
| KR | 18055 | 0.53 (0.52, 0.54) | 0.58 (0.57, 0.59) |
| MX | 18055 | 0.54 (0.53, 0.54) | 0.59 (0.58, 0.59) |
| SA | 18055 | 0.59 (0.58, 0.60) | 0.63 (0.62, 0.64) |
| US | 18055 | 0.57 (0.56, 0.58) | 0.62 (0.61, 0.63) |

