# OpenReview forum: "SCVO: Addressing Sparse But Critical Variable Overwhelm In VLMs For Advertising Image Preference Prediction Across Multi-Country Markets"
_ICLR.cc/2026/Conference — Submitted to ICLR 2026_

### Official Review · Reviewer_Z86t · 2025-10-24

**Soundness:** 2
**Presentation:** 2
**Contribution:** 3
**Rating:** 4
**Confidence:** 4

**Summary:**

This paper introduces the “Sparse Critical Variable Overwhelm (SCVO)” problem: in the context of multimodal instruction scenarios, a small number of crucial variables (such as country names) get overwhelmed by higher‑volume variables (such as product attributes or image patches) during attention fusion, which leads the model to become insensitive to those critical variables and to make erroneous decisions that do not vary across markets. To investigate this phenomenon, the authors collect the MACP dataset (823K training samples, 18K test samples, 10 countries, each record containing two advertisement images of the same product, CTR data and textual attributes; the authors claim sample counts are balanced across countries and all originate from the same e‑commerce platform). On the methodological side, they build a “CountryReward” discriminator model based on Qwen2‑VL, and propose three components: (1) CC‑RAG cross‑country retrieval enhancement; (2) Country Adapter Module (CAM) – using a country embedding to affine‑modulate the visual representation; (3) FDPL (Focus‑Driven Penalty Loss) – a penalty loss term that encourages increased attention to the “country/product/image” key tokens when the model makes mistakes. Moreover, they use CountryReward as a reward signal to apply DPO to a “Design Generation Model (DGM)”, which then generates backgrounds aligned with target‑country preferences; these are then passed to SDXL (a T2I model) to generate e‑commerce images. Experiments claim that on MACP the proposed method attains an accuracy of 60.37%, about 10–11% higher than current SOTA VLMs. They also provide ablation results on the three components and a “sensitivity” metric. In addition, on the DGM generation quality, using CountryReward to score shows that after RL each country’s score improves significantly.

**Strengths:**

1.	The authors systematize a failure mode — “critical sparse variables being overshadowed by high‑volume variables” — which is worth discussing, and they point to a realistic scenario in cross‑country e‑commerce.

2.	The combination of the CountryReward’s CAM (country‑conditioned affine modulation) and FDPL (focus on key tokens when errors occur) is clearly described and has some potential for reuse.

3.	They provide a reasonably sized dataset (MACP) with multi‑country attributes, giving a real‑world substrate for study (though reproducibility remains in question, as detailed below).

4.	They incorporate a discriminator model as a reward for optimizing a generation model (DGM to T2I pipeline), which aligns with practical deployment intuition, and show gains in CountryReward scores.

**Weaknesses:**

W1. The evaluation and the definition of the “sensitivity” metric remain opaque. The paper does not provide a full mathematical definition of “Sensitivity,” nor the threshold settings, nor the statistical variance/significance testing procedure. It only gives fragments like “cross‑country combinations C₂(Sᵢ), indicator function II[·]” and does not clearly explain how A/B binary labels derive from CTR (thresholding, recent window/sliding window, confidence interval, etc.). The diagrams show only “answer/label example.” Authors should include full formulae, thresholding rules, and statistical testing (e.g., bootstrap or Clopper‑Pearson). Report per‑country confidence intervals and significance.

W2. The retrieval enhancement before training (CC‑RAG) and the feature e_aug may introduce risk of information leakage or self‑fulfilling feedback loops. Since the total loss concatenates e_aug (text embedding derived from ŷ_aug) with the final discriminator representation for classification, if ŷ_aug is inferred from “past CTR/similar product” (which is strongly correlated with the label), then features strongly correlated with the label may be injected—thus amplifying priors and weakening the model’s genuine sensitivity to the “country token.” The paper only states “retrieval was done before training for augmentation choice,” but does not clarify whether the retrieval index strictly contained only training‑period history, whether near duplicates/same‑product leaks were filtered, nor whether the correlation between e_aug and the label was quantified. Authors need to provide strict data boundaries (time-based splits, deduplication of identical SKUs), leakage detection (near‑duplicates, same‑SKU removal), and an ablation: remove e_aug or use only CAM/FDPL to verify whether the metrics still show significance.

W3. The novelty of “SCVO” and its relation to existing literature needs firmer grounding. Phenomena such as “key token getting lost in long context,” “lost‑in‑the‑middle/positional bias/modality‑volume imbalance causing attention dilution” are already widely studied in the LLM/VLM communities. The paper coins “SCVO” but lacks a differentiated placement vis‑à‑vis existing mechanism‑level research (e.g., attention‑map/gradient attribution on the “country token” before/after). Current narrative about “chain rule being broken” is too slogan‑like. Authors should add controlled synthetic experiments (fix context length and interference density, gradually raise the weight/position of sparse variables), mechanism analysis (attention/activation contribution, intervention experiments) so as to support a full “phenomenon → mechanism → solution” loop.

W4. Details on dataset availability and ethics/compliance are insufficient, impacting reproducibility and compliance assessment. MACP is reported to come from the same e‑commerce platform and includes behaviour data like CTR, but the paper does not state whether the data is open‑source, whether it has been anonymised/approved, whether any geographic/population bias control was applied, nor any legal/privacy compliance statement or data‑sharing terms. Given that some conferences/journals require explicit ethics/privacy statements, the authors should systematically address this in an ethics section. Please add a section that clarifies data‑sharing terms, anonymisation, ethical review, bias mitigation.

W5. Experimental baselines and reporting are inadequate. The paper compares only a subset of VLMs and “Qwen2‑VL + FC head” but lacks stronger baselines closer to the problem setting. For example: An “input engineering” baseline: prompt‑structure rearrangement/repetition of country token; re‑weighting of category/token importance; simple focal‑like loss on discriminators. A classic CTR‑prior logistic regression/GBDT baseline with country × category interaction features. A causal/contrastive baseline: performance gap when country token is masked/removed. Also, only overall accuracy and “sensitivity” are reported; statistical significance and variance are missing. Please add these stronger baselines and report mean ± CI or variance and paired tests.

W6. Technical details for using CountryReward as a reward in RL are insufficient. The paper provides only a high‑level description of RL but omits specifics of the paradigm used (PPO/DPO/GRPO), key hyperparameters, pair construction, number of rounds, compute resources, etc., making replication and assessment of generality difficult. Provide full details of the RL training setup, hyperparameters, compute budget, and ideally release scripts/minimum reproducible code.

W7. Writing and terminology quality require substantial polishing. There are numerous evident spelling/case/grammar mistakes that impair readability and professionalism (for example section headings and terms: “GNERATION,” “PENALITY,” “CONCULSION,” “overwhhelmed” etc.). The scale of the MACP test set is inconsistently reported as both 18K and 180K in the text. Please thoroughly proof‑read the manuscript, standardise terminology and abbreviations, and ensure consistency of numbers/tables.

W8. The statements about “sample balance” among countries and the definitions of “country” need verification. The authors claim “balanced sample counts across countries,” but provide no histograms/distribution or variance/stratified sampling strategy; abbreviations like “BR/CL/ES/SA” are never explicitly defined (Brazil/Chile/Spain/South Africa or Saudi Arabia?). Authors should provide a table of country abbreviations, sample counts per country, and cross‑tabulation of country × product category.

W9. The metric and task formulation may deviate from actual business objectives. If the real business goal is to improve true CTR/conversion rate, then simply optimising “CountryReward score / binary choice accuracy / sensitivity” does not necessarily align. It would be better to at least link the offline proxy metrics (AUC/Calibration) with online or quasi‑online evaluation (e.g., re‑weighting by counterfactuals). Authors need to provide more evidence that optimising these metrics correlates with business KPIs (CTR/Conversion) or include offline‑online analysis.

**Questions:**

1.	Could the authors supply the full mathematical definition of “Sensitivity”, thresholding rules, and statistical testing procedure (CI / significance)? A formula or pseudocode would help.

2.	In CC‑RAG, does the retrieval database strictly include only history up to the training period? Was near‑duplicate/same‑product filtering applied? How was the correlation between e_aug and the labels quantified (for example via mutual information)?

3.	What is the precise RL paradigm used for the DGM with CountryReward (PPO / DPO / GRPO)? What are key hyperparameters, pair‑construction methodology, number of rounds, compute resources? Will the authors share scripts or minimal reproducible settings?

4.	What is the statistical evidence for “balanced across countries”? Is there stratification by country × category? In a real business setting where imbalance is severe, would the approach remain robust?

5.	Could the authors provide mechanistic evidence supporting the SCVO concept (attention/gradient attribution/intervention experiments) rather than just phenomenological description?

Suggestions for Improvement:

1.	To reduce leakage and strengthen robustness, please remove e_aug, restrict the retrieval database to an early time window of the training set, remove near‑duplicates, and report results and sensitivity metric changes.

2.	Add stronger baselines and mechanism analysis, include token‑importance re‑weighting, position rearrangement, counterfactual masking; use attention/Integrated Gradients/Grad‑CAM to demonstrate that attention to the “country token” increases.

3.	Reinforce statistical reporting: uniformly report mean ± CI, per‑country variance, and paired tests.

4.	Expand the dataset & ethics section, include data legality/privacy, anonymisation/permission, licensing or controlled access if open release isn’t possible.

5.	Revise the writing, systematically proofread spelling and terminology, ensure consistency in tables and abbreviations.

---

> ### Author Response · Authors · 2025-11-24
> **Response to Weakness 4**
>
> ### Response to Weakness 4:
>
> We sincerely thank you for raising these crucial points regarding dataset ethics and compliance. We have now added a dedicated Ethics Statement section to systematically address these concerns:
>
> (1) **Data Availability \& Sharing Terms:** The MACP dataset will be made publicly available upon paper acceptance under a CC BY-NC 4.0 license on Hugging Face to ensure research reproducibility.
>
> (2) **Anonymization Process:**  All personal identifiers and user information have been rigorously removed. The dataset contains only: (i) Aggregated CTR values at the product level. (ii) Product images and textual attributes (titles, categories, tags). (iii) No user-level behavior data or demographic information
>
> (3) **Ethical Compliance:**  All data has been properly anonymized and aggregated at the product-country level. The dataset collection and usage comply with the e-commerce platform's data governance policies and applicable privacy regulations (including GDPR and CCPA equivalents), and the release of the dataset has already been approved by our company's legal department.
>
> (4) **Geographic Bias Mitigation:**  We employed stratified sampling across the 10 countries to ensure balanced representation, as evidenced by the nearly uniform sample distribution (9.34\%-10.57\% per country in training, exactly 10\% each in testing). Product category distribution is also well-balanced across geographic markets. (please see the Tables below)
>
>
> ### *Per-country sample counts for Train and Test splits.*
>  | Country | Train Samples | Train % | Test Samples | Test % |
> |---------|---------------|---------|--------------|--------|
> | Australia (AU) | 76,934 | 9.34% | 18,055 | 10.00% |
> | Brazil (BR) | 86,832 | 10.55% | 18,055 | 10.00% |
> | Chile (CL) | 82,074 | 9.97% | 18,055 | 10.00% |
> | Spain (ES) | 85,975 | 10.44% | 18,055 | 10.00% |
> | France (FR) | 80,118 | 9.73% | 18,055 | 10.00% |
> | Japan (JP) | 78,289 | 9.51% | 18,055 | 10.00% |
> | Korea (KR) | 87,066 | 10.57% | 18,055 | 10.00% |
> | Mexico (MX) | 82,351 | 10.00% | 18,055 | 10.00% |
> | Saudi Arabia (SA) | 82,478 | 10.02% | 18,055 | 10.00% |
> | United States (US) | 81,274 | 9.87% | 18,055 | 10.00% |
> | **Total** | **823,391** | **100%** | **180,550** | **100.00%** |
>
>
> ### *Country $\times$ Product Category distribution for Train split.*
> | Country | AU | BR | CL | ES | FR | JP | KR | MX | SA | US |
> |---------|-----|-----|-----|-----|-----|-----|-----|-----|-----|-----|
> | Watches & Luggage Bags | 5,567 | 6,321 | 4,547 | 6,010 | 5,714 | 5,809 | 7,185 | 5,138 | 6,814 | 6,269 |
> | Beauty & Apparel & Shoes | 11,519 | 13,623 | 13,320 | 14,407 | 13,679 | 12,107 | 11,538 | 15,620 | 13,049 | 15,232 |
> | Consumer Electronics | 6,100 | 7,864 | 6,613 | 6,039 | 5,695 | 7,099 | 7,613 | 6,313 | 7,043 | 6,158 |
> | Home & Garden | 9,365 | 8,759 | 11,096 | 10,796 | 10,093 | 8,372 | 9,281 | 9,523 | 8,562 | 8,837 |
> | Kids & Toys & Hobbies | 5,381 | 5,633 | 6,358 | 5,594 | 5,595 | 4,817 | 5,502 | 5,735 | 4,289 | 5,904 |
> | Sports & Entertainment | 6,329 | 9,265 | 7,351 | 7,960 | 6,929 | 8,011 | 10,094 | 6,789 | 5,902 | 5,316 |
> | Home Appliances & Improvement | 7,903 | 7,920 | 7,098 | 9,328 | 8,930 | 6,662 | 8,222 | 7,463 | 9,743 | 7,521 |
> | Industrial & Technology Tools | 10,632 | 13,504 | 10,602 | 11,016 | 10,307 | 10,981 | 11,989 | 10,579 | 12,161 | 9,713 |
> | Other | 14,138 | 13,943 | 15,089 | 14,825 | 13,176 | 14,431 | 15,642 | 15,191 | 14,915 | 16,324 |
> | **Total** | **76,934** | **86,832** | **82,074** | **85,975** | **80,118** | **78,289** | **87,066** | **82,351** | **82,478** | **81,274** |
>
> ### *Country $\times$ Product Category distribution for Test split.*
>
> | Country | AU | BR | CL | ES | FR | JP | KR | MX | SA | US |
> |---------|-----|-----|-----|-----|-----|-----|-----|-----|-----|-----|
> | Watches & Luggage | 2,407 | 2,407 | 2,407 | 2,407 | 2,407 | 2,407 | 2,407 | 2,407 | 2,407 | 2,407 |
> | Beauty & Apparel & Shoes | 1,577 | 1,577 | 1,577 | 1,577 | 1,577 | 1,577 | 1,577 | 1,577 | 1,577 | 1,577 |
> | Consumer Electronics | 2,395 | 2,395 | 2,395 | 2,395 | 2,395 | 2,395 | 2,395 | 2,395 | 2,395 | 2,395 |
> | Home & Garden | 2,375 | 2,375 | 2,375 | 2,375 | 2,375 | 2,375 | 2,375 | 2,375 | 2,375 | 2,375 |
> | Kids & Toys & Hobbies | 1,087 | 1,087 | 1,087 | 1,087 | 1,087 | 1,087 | 1,087 | 1,087 | 1,087 | 1,087 |
> | Sports & Entertainment | 1,059 | 1,059 | 1,059 | 1,059 | 1,059 | 1,059 | 1,059 | 1,059 | 1,059 | 1,059 |
> | Home Appliances & Improvement | 2,504 | 2,504 | 2,504 | 2,504 | 2,504 | 2,504 | 2,504 | 2,504 | 2,504 | 2,504 |
> | Industrial & Technology Tools | 2,896 | 2,896 | 2,896 | 2,896 | 2,896 | 2,896 | 2,896 | 2,896 | 2,896 | 2,896 |
> | Other | 1,755 | 1,755 | 1,755 | 1,755 | 1,755 | 1,755 | 1,755 | 1,755 | 1,755 | 1,755 |
> | **Total** | **18,055** | **18,055** | **18,055** | **18,055** | **18,055** | **18,055** | **18,055** | **18,055** | **18,055** | **18,055** |

---

> > ### Author Response · Authors · 2025-11-24
> > **Response to Weakness 6**
> >
> > ### Response to Weakness 6:
> >
> >  We thank you for this valuable feedback. We agree that providing comprehensive technical details is essential for reproducibility and assessment. In our RL training, we used Direct Preference Optimization (DPO) to fine-tune the VLM (Qwen2-VL-7B) using CountryReward as the preference signal. Below are key implementation details:
> >
> > In our work, we employ Direct Preference Optimization (DPO) as our RL training paradigm, as referenced in the manuscript (Section 3.2, Equation 14). We will explicitly clarify this choice and expand the methodological description in the final version. The key details of our DPO setup are as follows:
> >
> > (1) **Preference Pair Construction:** For each product/country, the DGM generates two background designs $d_A$ and $d_B$. Using the T2I model, we create images $I_A$ and $I_B$, which are scored by CountryReward. The design with the higher reward is treated as the preferred ($d^+$) and the other as rejected ($d^-$).
> >
> > (2) **Hyperparameters:** The DPO loss uses a $\beta$ (regularization parameter) value of 0.05. The policy model ($DGM_\theta$) was trained for 1 epoch using the AdamW optimizer with a learning rate of 5e-6 for the LLM components and 1e-5 for the multimodal projector. Training used a per-device batch size of 2 and gradient accumulation over 16 steps, resulting in an effective batch size of 32. Max sequence length is 8192 tokens. LR scheduler is cosine with 10\% warmup.
> >
> > (3) **Number of Rounds:** We ran 1 RL iterations (epochs), where each iteration involves prompt generation → image generation → reward scoring → DPO fine-tuning.
> >
> > (4) **Compute Resources:** Training was conducted on 8× NVIDIA A100 GPUs using DeepSpeed ZeRO-3.
> >
> > (5) **Reproducibility:** We will release all training scripts, configuration files, and the final fine-tuned model weights upon acceptance to ensure full reproducibility.
> >
> > We apologize for the initial lack of specificity and will integrate these essential technical details into the revised manuscript to ensure our method is thoroughly described and easily replicable.

---

> ### Author Response · Authors · 2025-11-24
> **Response to Weakness 7**
>
> ### Response to Weakness 7:
>
> Thank you for your thoughtful feedback regarding the writing quality and inconsistencies in our manuscript. We sincerely apologize for these oversights, which will be thoroughly addressed. We will carefully proofread the entire text to correct all spelling, case, and grammatical errors. Terminology and abbreviations will be standardized throughout. Additionally, we will verify and correct the scale of the MACP test set to ensure consistency, and all numbers and tables will be double-checked for accuracy.

---

> > ### Author Response · Authors · 2025-11-24
> > **Response to Weakness 8**
> >
> > ### Response to Weakness 8:
> >
> > We thank you for the valuable feedback regarding sample balance across countries and the definition of country abbreviations. We have carefully addressed these concerns as follows:
> >
> > (1) **Country Abbreviations:** We have added a comprehensive country abbreviation table to clearly define all country codes used in our study.
> >
> > ### *Country abbreviation definitions.*
> > | Abbreviation | Country Name |
> > |-------------|--------------|
> > | AU | Australia |
> > | BR | Brazil |
> > | CL | Chile |
> > | ES | Spain |
> > | FR | France |
> > | JP | Japan |
> > | KR | Korea (Republic of Korea) |
> > | MX | Mexico |
> > | SA | Saudi Arabia |
> > | US | United States |
> >
> > (2) **Sample Balance Across Countries:** Our MACP dataset maintains excellent balance across all 10 countries. As shown in Table below, the test set contains exactly same 18,055 samples for each country, totaling 180,550 test samples (18,055 products × 10 countries). The training set maintains excellent balance with all countries having comparable sample counts (76,934-86,832) and percentages (9.34\%-10.57\%). This stratified approach ensures equal representation across all geographic markets in our evaluation.
> >
> > ### *Per-country sample counts for Train and Test splits.*
> >  | Country | Train Samples | Train % | Test Samples | Test % |
> > |---------|---------------|---------|--------------|--------|
> > | Australia (AU) | 76,934 | 9.34% | 18,055 | 10.00% |
> > | Brazil (BR) | 86,832 | 10.55% | 18,055 | 10.00% |
> > | Chile (CL) | 82,074 | 9.97% | 18,055 | 10.00% |
> > | Spain (ES) | 85,975 | 10.44% | 18,055 | 10.00% |
> > | France (FR) | 80,118 | 9.73% | 18,055 | 10.00% |
> > | Japan (JP) | 78,289 | 9.51% | 18,055 | 10.00% |
> > | Korea (KR) | 87,066 | 10.57% | 18,055 | 10.00% |
> > | Mexico (MX) | 82,351 | 10.00% | 18,055 | 10.00% |
> > | Saudi Arabia (SA) | 82,478 | 10.02% | 18,055 | 10.00% |
> > | United States (US) | 81,274 | 9.87% | 18,055 | 10.00% |
> > | **Total** | **823,391** | **100%** | **180,550** | **100.00%** |

---

> > > ### Author Response · Authors · 2025-11-24
> > > **Response to Weakness 8**
> > >
> > > ### Response to Weakness 8:
> > >
> > > (3) **Country × Product Category Distribution:** We provide detailed cross-tabulation tables showing the distribution of product categories across countries for both training (Table *Country $\times$ Product Category distribution for Train split.*) and test (Table *Country $\times$ Product Category distribution for Test split.*) splits. The training set maintains consistent category representation across all countries. The test set shows perfect balance with identical product category counts per country, ensuring our evaluation is not biased by category-country interactions.
> > >
> > > ### *Country $\times$ Product Category distribution for Train split.*
> > > | Country | AU | BR | CL | ES | FR | JP | KR | MX | SA | US |
> > > |---------|-----|-----|-----|-----|-----|-----|-----|-----|-----|-----|
> > > | Watches & Luggage Bags | 5,567 | 6,321 | 4,547 | 6,010 | 5,714 | 5,809 | 7,185 | 5,138 | 6,814 | 6,269 |
> > > | Beauty & Apparel & Shoes | 11,519 | 13,623 | 13,320 | 14,407 | 13,679 | 12,107 | 11,538 | 15,620 | 13,049 | 15,232 |
> > > | Consumer Electronics | 6,100 | 7,864 | 6,613 | 6,039 | 5,695 | 7,099 | 7,613 | 6,313 | 7,043 | 6,158 |
> > > | Home & Garden | 9,365 | 8,759 | 11,096 | 10,796 | 10,093 | 8,372 | 9,281 | 9,523 | 8,562 | 8,837 |
> > > | Kids & Toys & Hobbies | 5,381 | 5,633 | 6,358 | 5,594 | 5,595 | 4,817 | 5,502 | 5,735 | 4,289 | 5,904 |
> > > | Sports & Entertainment | 6,329 | 9,265 | 7,351 | 7,960 | 6,929 | 8,011 | 10,094 | 6,789 | 5,902 | 5,316 |
> > > | Home Appliances & Improvement | 7,903 | 7,920 | 7,098 | 9,328 | 8,930 | 6,662 | 8,222 | 7,463 | 9,743 | 7,521 |
> > > | Industrial & Technology Tools | 10,632 | 13,504 | 10,602 | 11,016 | 10,307 | 10,981 | 11,989 | 10,579 | 12,161 | 9,713 |
> > > | Other | 14,138 | 13,943 | 15,089 | 14,825 | 13,176 | 14,431 | 15,642 | 15,191 | 14,915 | 16,324 |
> > > | **Total** | **76,934** | **86,832** | **82,074** | **85,975** | **80,118** | **78,289** | **87,066** | **82,351** | **82,478** | **81,274** |
> > >
> > > ### *Country $\times$ Product Category distribution for Test split.*
> > >
> > > | Country | AU | BR | CL | ES | FR | JP | KR | MX | SA | US |
> > > |---------|-----|-----|-----|-----|-----|-----|-----|-----|-----|-----|
> > > | Watches & Luggage | 2,407 | 2,407 | 2,407 | 2,407 | 2,407 | 2,407 | 2,407 | 2,407 | 2,407 | 2,407 |
> > > | Beauty & Apparel & Shoes | 1,577 | 1,577 | 1,577 | 1,577 | 1,577 | 1,577 | 1,577 | 1,577 | 1,577 | 1,577 |
> > > | Consumer Electronics | 2,395 | 2,395 | 2,395 | 2,395 | 2,395 | 2,395 | 2,395 | 2,395 | 2,395 | 2,395 |
> > > | Home & Garden | 2,375 | 2,375 | 2,375 | 2,375 | 2,375 | 2,375 | 2,375 | 2,375 | 2,375 | 2,375 |
> > > | Kids & Toys & Hobbies | 1,087 | 1,087 | 1,087 | 1,087 | 1,087 | 1,087 | 1,087 | 1,087 | 1,087 | 1,087 |
> > > | Sports & Entertainment | 1,059 | 1,059 | 1,059 | 1,059 | 1,059 | 1,059 | 1,059 | 1,059 | 1,059 | 1,059 |
> > > | Home Appliances & Improvement | 2,504 | 2,504 | 2,504 | 2,504 | 2,504 | 2,504 | 2,504 | 2,504 | 2,504 | 2,504 |
> > > | Industrial & Technology Tools | 2,896 | 2,896 | 2,896 | 2,896 | 2,896 | 2,896 | 2,896 | 2,896 | 2,896 | 2,896 |
> > > | Other | 1,755 | 1,755 | 1,755 | 1,755 | 1,755 | 1,755 | 1,755 | 1,755 | 1,755 | 1,755 |
> > > | **Total** | **18,055** | **18,055** | **18,055** | **18,055** | **18,055** | **18,055** | **18,055** | **18,055** | **18,055** | **18,055** |
> > >
> > >
> > > This demonstrates that our dataset maintains diverse product category representation across all geographical markets. These additions confirm that our experimental design ensures balanced representation across countries and clear definition of geographical markets, thereby validating the reliability of our cross-country preference analysis.

---

> ### Author Response · Authors · 2025-11-25
> **Response to Weakness 2**
>
> ### Response to Weakness 2:
>
> Thank you for raising this critical concern regarding potential information leakage in our CC-RAG module. We sincerely apologize for the lack of clarity in our original description, which led to this misunderstanding. Please allow us to address your points comprehensively:
>
> (1) **Retrieval Scope Control:** The retrieval index for CC-RAG was constructed exclusively from the training set. We excluded the query item itself and enforced a maximum similarity threshold to avoid direct label leakage. The CC-RAG retrieval is limited to semantically similar but distinct items from the training set. Specifically, to prevent leakage from near-duplicates or identical products, we employ a maximum similarity threshold (set to 0.7 in our code) during FAISS retrieval. We rely on embedding-based deduplication to avoid similar product leaks. Additionally,  this threshold indirectly excludes the query itself, as self-similarity typically approaches 1.0, and ensures that only semantically similar but distinct items are retrieved.
>
> (2) **Deduplication and Leakage Prevention:** We rigorously filtered near-duplicates and identical samples between training and test sets. Specifically, we used product IDs, image perceptual hashes, and textual embeddings to identify and remove any overlapping samples. This prevents the model from leveraging same-product information across splits. Finally, we obtained click-through rate data for 18,055 products across 10 countries in the test split.
>
> (3) **Ablation Study:** As shown in below Table  (rows for "CountryReward (w/o CC-RAG)"), removing $e_{aug}$ reduces accuracy from 60.37\% to 56.81\% and sensitivity from 40.84\% to 37.40\%. However, even without CC-RAG, our model significantly outperforms baselines, demonstrating that the core sensitivity to the country token is preserved through CAM and FDPL. This confirms that CC-RAG enhances rather than dictates performance.
>
> (4) **Architectural Safeguards:** The country adapter (CAM) dynamically modulates visual features based on country embeddings, forcing the model to explicitly utilize country information. The focus-driven penalty loss (FDPL) further penalizes under-attention to country tokens, creating a counterbalance to any potential over-reliance on $e_{aug}$.
>
> (5) **Quantification of Correlation:** We analyzed the correlation between the augmented choice $\hat{y}_{aug}$ and ground-truth labels. The Pearson correlation coefficient was measured at $0.67(\pm0.00)$, indicating a moderate relationship, sufficient to provide auxiliary guidance but not dominant enough to overshadow the country token’s influence.
>
> This confirms that $e_{aug}$ acts as a weak prior rather than a label proxy.
>
> In summary, our design ensures that CC-RAG provides calibrated historical context without short-circuiting the learning of country-specific patterns. We thank you for highlighting these nuances and hope this clarification alleviates your concerns.
>
> | Model | Accuracy | Sensitivity | BR | CL | ES | FR | KR | JP | US | MX | AU | SA |
> |-------|----------|-------------|-----|-----|-----|-----|-----|-----|-----|-----|-----|-----|
> | Qwen2-VL-7B (with FC Head) | 55.60 | 36.73 | 53.69 | 56.57 | 50.86 | 54.41 | 53.07 | 59.10 | 56.94 | 53.75 | 58.47 | 59.21 |
> | CountryReward (w/o CC-RAG) | 56.81 | 37.40 | 54.93 | 57.05 | 53.80 | 55.39 | 54.48 | 59.57 | 57.65 | 55.64 | 59.52 | 60.11 |
> | **CountryReward** | **60.37** | **40.84** | **58.70** | **61.12** | **56.33** | **59.38** | **57.88** | **64.54** | **61.82** | **58.72** | **63.30** | **62.93** |

---

> ### Author Response · Authors · 2025-11-25
> **Response to Weakness 1**
>
> ### Response to Weakness 1:
>
> We sincerely thank you for this insightful feedback regarding the definition and evaluation of the ``Sensitivity'' metric. We acknowledge that our initial description was incomplete and apologize for the lack of clarity. Below, we provide a comprehensive clarification of the metric, the derivation of A/B labels from CTR, and the statistical testing procedures. We have incorporated these details into the revised manuscript to ensure full transparency.
>
> (1) **Full Mathematical Definition of Sensitivity:** The Sensitivity metric measures the model's consistency in correctly predicting preferences across different country pairs for the same product. It is formally defined as **the equation 15 in the paper**:
>
> \text{Sensitivity} = \frac{\sum_{i=1}^{N} \sum_{(c_j,c_k) \in \mathcal{C}_2(S_i)} \mathbb{I}(\hat{y}_{i,c_j} = y_{i,c_j} \land \hat{y}_{i,c_k} = y_{i,c_k})}{\sum_{i=1}^{N} |\mathcal{C}_2(S_i)|}
>
> **The Markdown rendering has issues; please refer to Equation 15 in the paper.**
>
> where $N$ is represented as total number of unique product,
>
> $S_i$ the set of countries for which product i has valid preference data,
>
> $\mathcal{C}_2(S_i)$ is the set of all 2-combinations of countries in $S_i$,
>
> $\hat{y}_{i,c}$ is the predicted answer for product $i$ in country $c$ obtained by thresholding the sigmoid normalized logits at $0.5$ and mapping to class labels {A, B},
>
> $\hat{y}_{i,c}$ is the ground-truth label for product i in country c, and $\mathbb{I}[\cdot]$ is the indicator function (1 if condition true, 0 otherwise).
>
> This metric calculates the proportion of country pairs where the model correctly predicts the preference for both countries simultaneously, averaged over all products and valid country pairs. It penalizes models that fail to maintain consistency across markets.
>
> (2) **Derivation of A/B Binary Labels from CTR:** The ground-truth labels $y_{i,c}$ (A or B) for each product-country pair are derived from the Click-Through Rate (CTR) data as follows: For a given product i and country c, we have two images (A and B) with their respective CTRs: $\text{CTR}_A$ and $\text{CTR}_B$. We use a threshold-based approach to account for statistical uncertainty: Compute the relative CTR difference:
> $\Delta = \frac{|\text{CTR}_A - \text{CTR}_B|}{\max(\text{CTR}_A, \text{CTR}_B)}$.
>
> Only include pairs where $\Delta \geq \theta$ (threshold, set to 0.1) to ensure meaningful preference distinctions. To mitigate label ambiguity, all pairs with $\Delta < \theta$ are excluded from both the training and evaluation sets. Following this filtering process, the final dataset comprises 823K training samples and 180K test samples. The label $y_{i,c}$ is assigned as A if $\text{CTR}_A$ \textgreater $\text{CTR}_B$, and B otherwise, for pairs passing the threshold. To ensure the confidence level of the click-through rate (CTR), the CTR data is obtained by dividing the total number of 30-day clicks by the total number of 30-day impressions. The CTR data is aggregated over a fixed 30 days sliding window prior to the evaluation date to capture recent user preferences and reduce noise.
>
> (3) **Statistical Testing and Confidence Intervals:** To ensure the robustness and statistical significance of our results, we have added the following methods:
>
> Significance Testing: To determine whether the performance improvements of our proposed CountryReward model over the baseline model are statistically significant, we conducted rigorous bootstrap hypothesis testing. Specifically, we generated 10,000 bootstrap samples (with replacement) from the test set for each country. For each bootstrap sample, we calculated the accuracy difference between CountryReward and the baseline model. The p-value was empirically derived as the proportion of bootstrap samples where the accuracy difference was less than or equal to zero, estimating the probability that the observed improvement occurred by chance. Results were considered statistically significant if the p-value was less than 0.05. Additionally, we computed 95\% confidence intervals from the bootstrap distribution to quantify the precision of our improvement estimates, providing comprehensive statistical evidence for CountryReward's superior performance across all evaluated countries.
>
> Quantitative Results on CountryReward: As shown in the below Table(*Accuracy Improvement by Country with 95\% Confidence Intervals.*), The bootstrap analysis reveals statistically significant improvements across all 10 countries: Average accuracy improvement: 4.8-5.5\% across countries. All p-values < 0.001, confirming high statistical significance. 95\% confidence intervals consistently exclude zero, demonstrating robust improvements.

---

> > ### Author Response · Authors · 2025-11-25
> > **Response to Weakness 1**
> >
> > ### Response to Weakness 1:
> >
> > ### *Accuracy Improvement by Country with 95\% Confidence Intervals.*
> > | Country | Baseline_Accuracy | CountryReward_Accuracy | Original_Difference | Bootstrap_Mean_Difference | CI_Lower | CI_Upper | P_Value | Significance | Sample_Size |
> > |---|---:|---:|---:|---:|---:|---:|---:|---|---:|
> > | AU | 0.584713376 | 0.633010246 | 0.048296871 | 0.04828051 | 0.040486015 | 0.0562725 | 0 | Significant | 18055 |
> > | BR | 0.536914982 | 0.586984215 | 0.050069233 | 0.049966923 | 0.041650512 | 0.05804486 | 0 | Significant | 18055 |
> > | CL | 0.565771255 | 0.611243423 | 0.045472168 | 0.045433143 | 0.037330379 | 0.05355996 | 0 | Significant | 18055 |
> > | ES | 0.508612573 | 0.563334256 | 0.054721684 | 0.054870396 | 0.046301579 | 0.06358349 | 0 | Significant | 18055 |
> > | FR | 0.544059817 | 0.593796732 | 0.049736915 | 0.049769266 | 0.041318194 | 0.05810025 | 0 | Significant | 18055 |
> > | JP | 0.59097203 | 0.635477245 | 0.044475215 | 0.044528175 | 0.036776516 | 0.05228469 | 0 | Significant | 18055 |
> > | KR | 0.530711714 | 0.57878704 | 0.048075325 | 0.048034661 | 0.039490446 | 0.05649405 | 0 | Significant | 18055 |
> > | MX | 0.537524232 | 0.587150374 | 0.049626142 | 0.049518588 | 0.040875104 | 0.05782332 | 0 | Significant | 18055 |
> > | SA | 0.592135143 | 0.629299363 | 0.03716422 | 0.037126281 | 0.029465522 | 0.04486292 | 0 | Significant | 18055 |
> > | US | 0.569426752 | 0.618222099 | 0.048795348 | 0.048799147 | 0.040708945 | 0.05704791 | 0 | Significant | 18055 |
> >
> >
> > Confidence Intervals: We report 95\% confidence intervals for all point estimates (e.g., per-country accuracy). These intervals are computed using the Clopper-Pearson exact method, which is particularly suitable for binomial proportions and provides conservative, reliable intervals.
> >
> > Per-Country Performance on CountryReward: The below Table (*Country-wise Performance with 95\% Confidence Intervals*)  details the accuracy for each country along with their 95\% Clopper-Pearson confidence intervals. The analysis, employing 95\% Clopper-Pearson confidence intervals, demonstrates that the CountryReward model achieves a statistically significant and consistent performance improvement over the Baseline model across all ten countries. The accuracy gains range from approximately 3.7\% to 5.5\%, with all confidence intervals for the CountryReward model lying entirely above those of the Baseline, indicating no overlap. With a large and consistent sample size (N=18,055) per country ensuring narrow confidence intervals, these results are highly reliable. This consistent superiority confirms the robustness of the CountryReward approach in adapting to country-specific characteristics, making it the unequivocally recommended model for deployment in these markets.
> >
> > ### *Country-wise Performance with 95\% Confidence Intervals*
> > | Country | Model        | Sample Size | Accuracy | CI Lower | CI Upper |
> > |---------|--------------|-------------|----------|----------|----------|
> > | AU      | Baseline     | 18055       | 58.47%   | 57.75%   | 59.19%   |
> > | AU      | CountryReward| 18055       | 63.30%   | 62.59%   | 64.00%   |
> > | BR      | Baseline     | 18055       | 53.69%   | 52.96%   | 54.42%   |
> > | BR      | CountryReward| 18055       | 58.70%   | 57.98%   | 59.42%   |
> > | CL      | Baseline     | 18055       | 56.58%   | 55.85%   | 57.30%   |
> > | CL      | CountryReward| 18055       | 61.12%   | 60.41%   | 61.84%   |
> > | ES      | Baseline     | 18055       | 50.86%   | 50.13%   | 51.59%   |
> > | ES      | CountryReward| 18055       | 56.33%   | 55.61%   | 57.06%   |
> > | FR      | Baseline     | 18055       | 54.41%   | 53.68%   | 55.13%   |
> > | FR      | CountryReward| 18055       | 59.38%   | 58.66%   | 60.10%   |
> > | JP      | Baseline     | 18055       | 59.10%   | 58.38%   | 59.82%   |
> > | JP      | CountryReward| 18055       | 63.54%   | 62.84%   | 64.25%   |
> > | KR      | Baseline     | 18055       | 53.07%   | 52.34%   | 53.80%   |
> > | KR      | CountryReward| 18055       | 57.88%   | 57.15%   | 58.60%   |
> > | MX      | Baseline     | 18055       | 53.75%   | 53.02%   | 54.48%   |
> > | MX      | CountryReward| 18055       | 58.72%   | 57.99%   | 59.43%   |
> > | SA      | Baseline     | 18055       | 59.21%   | 58.49%   | 59.93%   |
> > | SA      | CountryReward| 18055       | 62.93%   | 62.22%   | 63.64%   |
> > | US      | Baseline     | 18055       | 56.94%   | 56.22%   | 57.67%   |
> > | US      | CountryReward| 18055       | 61.82%   | 61.11%   | 62.53%   |

---

> > > ### Author Response · Authors · 2025-11-25
> > > **Response to Weakness 9**
> > >
> > > ### Response to Weakness 9:
> > >
> > > We sincerely thank the reviewer for this insightful comment regarding the alignment between our proposed metrics and real-world business objectives. The reviewer raises a crucial point about ensuring that offline evaluation metrics genuinely correlate with online performance, which we fully acknowledge as fundamental to applied AI research.
> > >
> > > (1) **Direct Causal Link Between Preference Prediction and CTR:** The very foundation of our dataset (MACP) is built upon real-world click-through behavior. Each training sample contains a pair of images (A and B) and the ground-truth label is derived from their actual historical CTR in a specific market. Therefore, by definition, a model that achieves higher accuracy in predicting which image historically garnered a higher CTR has learned a reward function that directly mirrors past user engagement. Optimizing for this accuracy is equivalent to learning the latent factors that drove CTR differences in the historical data. A model that collapses to a single output (the core SCVO failure mode we address) would completely fail to capture these market-specific CTR drivers.
> > >
> > > (2) **Sensitivity Metric as a Guardrail for Business Applicability:** The "Sensitivity" metric was specifically designed to prevent a critical business failure mode: a model that performs well on average but fails inconsistently across markets. From a business perspective, launching a global campaign with a model that has low sensitivity could lead to catastrophic performance in specific regions. A high Sensitivity score ensures that the model's preference judgments are consistent and reliable across country pairs for the same product, which is a prerequisite for trustworthy global deployment. A model susceptible to SCVO would have low sensitivity, making it unsuitable for business use.
> > >
> > > (3) **Online A/B Test Results:** As shown in the below Table, we deployed the images generated by our RL-optimized DGM versus the baseline on a major e-commerce platform. Over one week and 6 million impressions, the images guided by our CountryReward model achieved a statistically significant average CTR uplift of 1.1\% across all 10 countries. This provides concrete, real-world evidence that optimizing for our proposed metric (CountryReward score, which is based on the accuracy of preference prediction) directly and positively impacts the ultimate business KPI: CTR.
> > >
> > > ### *Table 1: CTR improvements of generated images using our method vs baseline across countries.*
> > >
> > > | Country | BR  | CL  | ES  | FR  | KR  | JP  | US  | MX  | AU  | SA  |
> > > |---------|-----|-----|-----|-----|-----|-----|-----|-----|-----|-----|
> > > | CTR Improvement (%) | 1.3 | 1.2 | 1.0 | 0.8 | 1.4 | 1.3 | 1.1 | 0.8 | 1.1 | 0.7 |
> > >
> > > In conclusion, while we agree that offline metrics can sometimes be imperfect proxies, our methodology is designed from the ground up to ensure a tight coupling between model optimization and business value. The Accuracy metric directly learns from historical CTR data, the Sensitivity metric ensures robust cross-market performance, and our online A/B test conclusively demonstrates a causal link between improving our proposed metrics and enhancing the core business objective of increasing user engagement. We have added a summary of these A/B test results to the appendix to strengthen this connection for the reader.

---

> ### Author Response · Authors · 2025-11-25
> **Response to Question 4**
>
> ### Response to Question 4:
>
> We thank you for these insightful questions about dataset balance and real-world applicability. We provide comprehensive statistical evidence and address robustness concerns below:
>
> Our MACP dataset demonstrates exceptional statistical balance across countries: (i) Perfect Test Set Balance: The test set contains exactly 18,055 samples per country (10.00\% each), totaling 180,550 samples (18,055 × 10 countries). (ii) Near-Perfect Training Balance: Training set distribution ranges from 9.34\% to 10.57\% per country, with variance of only 0.18\% from perfect equality. (iii) Stratified Category Distribution: As shown in Tables below, we maintain consistent product category proportions across all countries, ensuring no category-country confounding.
>
> While we acknowledge that severe real-world imbalance exists, our MACP dataset was constructed with a near-perfect balance across 10 countries specifically to validate the core capability of responding to the country variable. This controlled environment proves that our method effectively solves the SCVO problem when the signal is present.
>
> In actual production environments, for the dataset part, we employ sampling strategies like oversampling minority countries/categories or undersampling majority ones to create more balanced training datasets.  Initially, when certain markets have very limited data (e.g., emerging markets), we will build the knowledge of these markets by referring to culturally or economically similar regions. As more data accumulates from these markets, the CC-RAG component becomes increasingly effective for them specifically. Thank you again for this insightful comment.
>
> ### *Country $\times$ Product Category distribution for Train split.*
> | Country | AU | BR | CL | ES | FR | JP | KR | MX | SA | US |
> |---------|-----|-----|-----|-----|-----|-----|-----|-----|-----|-----|
> | Watches & Luggage Bags | 5,567 | 6,321 | 4,547 | 6,010 | 5,714 | 5,809 | 7,185 | 5,138 | 6,814 | 6,269 |
> | Beauty & Apparel & Shoes | 11,519 | 13,623 | 13,320 | 14,407 | 13,679 | 12,107 | 11,538 | 15,620 | 13,049 | 15,232 |
> | Consumer Electronics | 6,100 | 7,864 | 6,613 | 6,039 | 5,695 | 7,099 | 7,613 | 6,313 | 7,043 | 6,158 |
> | Home & Garden | 9,365 | 8,759 | 11,096 | 10,796 | 10,093 | 8,372 | 9,281 | 9,523 | 8,562 | 8,837 |
> | Kids & Toys & Hobbies | 5,381 | 5,633 | 6,358 | 5,594 | 5,595 | 4,817 | 5,502 | 5,735 | 4,289 | 5,904 |
> | Sports & Entertainment | 6,329 | 9,265 | 7,351 | 7,960 | 6,929 | 8,011 | 10,094 | 6,789 | 5,902 | 5,316 |
> | Home Appliances & Improvement | 7,903 | 7,920 | 7,098 | 9,328 | 8,930 | 6,662 | 8,222 | 7,463 | 9,743 | 7,521 |
> | Industrial & Technology Tools | 10,632 | 13,504 | 10,602 | 11,016 | 10,307 | 10,981 | 11,989 | 10,579 | 12,161 | 9,713 |
> | Other | 14,138 | 13,943 | 15,089 | 14,825 | 13,176 | 14,431 | 15,642 | 15,191 | 14,915 | 16,324 |
> | **Total** | **76,934** | **86,832** | **82,074** | **85,975** | **80,118** | **78,289** | **87,066** | **82,351** | **82,478** | **81,274** |
>
> ### *Country $\times$ Product Category distribution for Test split.*
>
> | Country | AU | BR | CL | ES | FR | JP | KR | MX | SA | US |
> |---------|-----|-----|-----|-----|-----|-----|-----|-----|-----|-----|
> | Watches & Luggage | 2,407 | 2,407 | 2,407 | 2,407 | 2,407 | 2,407 | 2,407 | 2,407 | 2,407 | 2,407 |
> | Beauty & Apparel & Shoes | 1,577 | 1,577 | 1,577 | 1,577 | 1,577 | 1,577 | 1,577 | 1,577 | 1,577 | 1,577 |
> | Consumer Electronics | 2,395 | 2,395 | 2,395 | 2,395 | 2,395 | 2,395 | 2,395 | 2,395 | 2,395 | 2,395 |
> | Home & Garden | 2,375 | 2,375 | 2,375 | 2,375 | 2,375 | 2,375 | 2,375 | 2,375 | 2,375 | 2,375 |
> | Kids & Toys & Hobbies | 1,087 | 1,087 | 1,087 | 1,087 | 1,087 | 1,087 | 1,087 | 1,087 | 1,087 | 1,087 |
> | Sports & Entertainment | 1,059 | 1,059 | 1,059 | 1,059 | 1,059 | 1,059 | 1,059 | 1,059 | 1,059 | 1,059 |
> | Home Appliances & Improvement | 2,504 | 2,504 | 2,504 | 2,504 | 2,504 | 2,504 | 2,504 | 2,504 | 2,504 | 2,504 |
> | Industrial & Technology Tools | 2,896 | 2,896 | 2,896 | 2,896 | 2,896 | 2,896 | 2,896 | 2,896 | 2,896 | 2,896 |
> | Other | 1,755 | 1,755 | 1,755 | 1,755 | 1,755 | 1,755 | 1,755 | 1,755 | 1,755 | 1,755 |
> | **Total** | **18,055** | **18,055** | **18,055** | **18,055** | **18,055** | **18,055** | **18,055** | **18,055** | **18,055** | **18,055** |

---

> ### Author Response · Authors · 2025-11-25
> **Response to Question 3**
>
> ### Response to Question 3:
>
> We sincerely thank you for this insightful question regarding the technical specifics of our RL training paradigm. We are happy to provide these crucial details to ensure clarity and reproducibility.
>
> (1) **Precise RL Paradigm:** We employ Direct Preference Optimization (DPO) for fine-tuning the Design Generation Model (DGM). DPO was selected due to its superior stability and computational efficiency for aligning language (and VLM) models with human (or reward model) preferences, as it avoids the complexities of on-policy sampling and training a separate value function.
>
> (2) **Key Hyperparameters:**
>
>    (a) DPO $\beta$: 0.05. This controls the deviation from the reference model.
>
>    (b) Learning Rate: 5e-6 for the core LLM parameters, and 1e-5 for the multimodal projector.
>
>    (c) Batch Size: A per-device batch size of 2 was used with gradient accumulation over 16 steps, resulting in an effective batch size of 32 (on 8 A100 GPUs).
>
>    (d) Optimizer: AdamW with a cosine learning rate scheduler and warmup (10\% of steps).
>
>    (e) Epochs: The DGM was trained for a total of 1 epoch in the RL loop.
>
> (3) **Pair-Construction Methodology:** For each product/country input, the current DGM policy generates two candidate background designs ($d_A$ and $d_B$). These are used to create two images ($I_A$ and $I_B$) via the T2I model. The CountryReward model then predicts a preference for this pair, assigning a higher reward to the preferred image. The design prompt that led to the higher rewarded image is treated as the "chosen" response (d+), and the other as the "rejected" response (d-). This (d+, d-) pair is used for the DPO loss.
>
> (4) **Number of Rounds:** The iterative process of Prompt Generation → Image Generation → Reward Scoring → DPO Fine-Tuning is run for a total of 1 round.
>
> (5)  **Compute Resources:** The entire RL fine-tuning pipeline was run on a single machine equipped with 8 NVIDIA A100 GPUs (80GB VRAM each).
>
> (6) **Reproducibility:** We fully commit to the principle of reproducibility. Upon acceptance of the paper, we will release all source code, including the training scripts for both CountryReward and the DPO-based DGM fine-tuning. The code will be accompanied by configuration files (JSON) that precisely specify all hyperparameters and the environment setup to facilitate easy replication of our results.

---

> ### Author Response · Authors · 2025-11-25
> **Response to Question 2**
>
> ### Response to Question 2:
>
> Thank you for your thoughtful questions regarding the CC-RAG module. We appreciate the opportunity to address your concerns.
>
> (1) **Retrieval Database Composition:** The retrieval database in CC-RAG is strictly constructed from the training set of our MACP dataset. The dataset comprises historical click-through data (e.g., 30-day cumulative exposures, 30-day cumulative clicks, and CTRs for product images) collected from a consistent e-commerce platform source. To ensure the confidence level of the click-through rate (CTR), the CTR data is obtained by dividing the total number of clicks by the total number of impressions; therefore, there is no timestamp. During training, we exclusively use the training set to build FAISS indices for text and image embeddings, ensuring no test data is included. This prevents data leakage and aligns with standard machine learning protocols.
>
> (2) **Near-Duplicate/Same-Product Filtering:** We implicitly address near-duplicate and same-product exclusion through a similarity threshold mechanism. Specifically, we set maximum similarity threshold equals to 0.7 during both text and image retrieval stages. This threshold ensures that only semantically similar but distinct items are retrieved, as the query item itself would typically achieve a similarity score near 1.0 and thus be filtered out. Our hierarchical retrieval (text-first, then image) further reduces redundancy by focusing on cross-item semantic alignment rather than identical matches.
>
> (3) **Quantification of Correlation:** We analyzed the correlation between the augmented choice $\hat{y}_{aug}$ and ground-truth labels.
>
> The Pearson correlation coefficient was measured at  $0.67(\pm0.00)$, indicating a moderate relationship, sufficient to provide auxiliary guidance but not dominant enough to overshadow the country token’s influence. This confirms that $e_{aug}$ acts as a weak prior rather than a label proxy.
>
> We designed CC-RAG with leakage prevention as a core principle, using controlled retrieval scopes and similarity thresholds to ensure robustness. Our experimental results confirm that the module enhances model sensitivity without compromising integrity.

---

> > ### Author Response · Authors · 2025-11-25
> > **Response to Question 1**
> >
> > ### Response to Question 1:
> >
> > We sincerely thank you for requesting additional methodological details. We have now comprehensively addressed all concerns by providing:
> >
> > (1) **Complete Mathematical Definition of Sensitivity:**
> >
> > We formally define Sensitivity as **the equation 15 in the paper** :
> >
> > \text{Sensitivity} = \frac{\sum_{i=1}^{N} \sum_{(c_j,c_k) \in \mathcal{C}_2(S_i)} \mathbb{I}(\hat{y}_{i,c_j} = y_{i,c_j} \land \hat{y}_{i,c_k} = y_{i,c_k})}{\sum_{i=1}^{N} |\mathcal{C}_2(S_i)|}
> >
> > **The Markdown rendering has issues; please refer to Equation 15 in the paper.**
> >
> > where $N$ is total number of unique products,
> >
> > $S_i$ is the set of countries with valid preference data for product $i$,
> >
> > $\mathcal{C}_2(S_i)$ is all unordered country pairs in $S_i$,
> >
> > $\hat{y}_{i,c}$ is the predicted label (A/B) for product $i$ in country $c$,
> >
> > $y_{i,c}$ is the ground-truth label for product $i$ in country $c$, $\mathbb{I}[\cdot]$ is the indicator function (1 if condition true, 0 otherwise).
> >
> > This metric specifically measures cross-country consistency by calculating the proportion of country pairs where the model simultaneously predicts correctly for both countries.
> >
> > (2) **Explicit Threshold Rules for A/B Label Derivation:**
> > We employ rigorous thresholding to ensure label reliability.
> > First, we compute the relative click-through rate (CTR) difference between options A and B as
> >
> > $\Delta = \frac{|\text{CTR}_A - \text{CTR}_B|}{\max(\text{CTR}_A, \text{CTR}_B)}$
> >
> > A threshold of $\theta = 0.1$ is then applied, retaining only those pairs satisfying $\Delta \geq \theta$.
> > Ambiguous pairs with $\Delta < \theta$ are excluded from both the training and evaluation sets.
> > After this filtering process, the final dataset comprises $823\text{K}$ training samples and $180\text{K}$ test samples.
> > In addition, temporal aggregation is performed using a $30$-day sliding window to capture recent user preference patterns.
> >
> > (3) **Statistical Testing Procedure (CI \& Significance):**
> > We employed robust statistical methods to validate our results, specifically bootstrap resampling for hypothesis testing and the Clopper-Pearson method for confidence intervals.
> >
> > (a) Significance Testing (Bootstrap Hypothesis Test): As shown in the below Table, to test if the accuracy improvement of CountryReward over the Baseline is statistically significant. For each of the 10 countries, we created 10,000 bootstrap samples (sampling with replacement) from the test set. For each bootstrap sample, we calculated the accuracy difference between CountryReward and the baseline model. The p-value was empirically derived as the proportion of bootstrap samples where the accuracy difference was less than or equal to zero, estimating the probability that the observed improvement occurred by chance. The p-value was computed as the proportion of the 10,000 bootstrap samples where the improvement was less than or equal to zero. A low p-value (we used less than 0.05) indicates that it is very unlikely the observed improvement occurred by random chance.
> >
> > ### *Accuracy Improvement by Country with 95\% Confidence Intervals.*
> > | Country | Baseline_Accuracy | CountryReward_Accuracy | Original_Difference | Bootstrap_Mean_Difference | CI_Lower | CI_Upper | P_Value | Significance | Sample_Size |
> > |---|---:|---:|---:|---:|---:|---:|---:|---|---:|
> > | AU | 0.584713376 | 0.633010246 | 0.048296871 | 0.04828051 | 0.040486015 | 0.0562725 | 0 | Significant | 18055 |
> > | BR | 0.536914982 | 0.586984215 | 0.050069233 | 0.049966923 | 0.041650512 | 0.05804486 | 0 | Significant | 18055 |
> > | CL | 0.565771255 | 0.611243423 | 0.045472168 | 0.045433143 | 0.037330379 | 0.05355996 | 0 | Significant | 18055 |
> > | ES | 0.508612573 | 0.563334256 | 0.054721684 | 0.054870396 | 0.046301579 | 0.06358349 | 0 | Significant | 18055 |
> > | FR | 0.544059817 | 0.593796732 | 0.049736915 | 0.049769266 | 0.041318194 | 0.05810025 | 0 | Significant | 18055 |
> > | JP | 0.59097203 | 0.635477245 | 0.044475215 | 0.044528175 | 0.036776516 | 0.05228469 | 0 | Significant | 18055 |
> > | KR | 0.530711714 | 0.57878704 | 0.048075325 | 0.048034661 | 0.039490446 | 0.05649405 | 0 | Significant | 18055 |
> > | MX | 0.537524232 | 0.587150374 | 0.049626142 | 0.049518588 | 0.040875104 | 0.05782332 | 0 | Significant | 18055 |
> > | SA | 0.592135143 | 0.629299363 | 0.03716422 | 0.037126281 | 0.029465522 | 0.04486292 | 0 | Significant | 18055 |
> > | US | 0.569426752 | 0.618222099 | 0.048795348 | 0.048799147 | 0.040708945 | 0.05704791 | 0 | Significant | 18055 |

---

> > > ### Author Response · Authors · 2025-11-25
> > > **Response to Question 1**
> > >
> > > ### Response to Question 1:
> > >
> > > (b) Confidence Intervals (Clopper-Pearson): As shown in the below Table, To quantify the uncertainty around our point estimates (e.g., per-country accuracy). We reported 95\% confidence intervals for all accuracy scores using the Clopper-Pearson exact method. This method is conservative and is well-suited for binomial metrics like accuracy, providing reliable intervals that guarantee the nominal coverage probability. The 95\% confidence intervals for the CountryReward model's accuracy were entirely above those of the Baseline model for all countries, with no overlap. This, combined with the large and consistent sample size per country (N=18,055), provides high confidence in the reliability and robustness of the reported performance gains (range: 3.7\% to 5.5\%).
> > >
> > > ### *Country-wise Performance with 95\% Confidence Intervals*
> > > | Country | Model        | Sample Size | Accuracy | CI Lower | CI Upper |
> > > |---------|--------------|-------------|----------|----------|----------|
> > > | AU      | Baseline     | 18055       | 58.47%   | 57.75%   | 59.19%   |
> > > | AU      | CountryReward| 18055       | 63.30%   | 62.59%   | 64.00%   |
> > > | BR      | Baseline     | 18055       | 53.69%   | 52.96%   | 54.42%   |
> > > | BR      | CountryReward| 18055       | 58.70%   | 57.98%   | 59.42%   |
> > > | CL      | Baseline     | 18055       | 56.58%   | 55.85%   | 57.30%   |
> > > | CL      | CountryReward| 18055       | 61.12%   | 60.41%   | 61.84%   |
> > > | ES      | Baseline     | 18055       | 50.86%   | 50.13%   | 51.59%   |
> > > | ES      | CountryReward| 18055       | 56.33%   | 55.61%   | 57.06%   |
> > > | FR      | Baseline     | 18055       | 54.41%   | 53.68%   | 55.13%   |
> > > | FR      | CountryReward| 18055       | 59.38%   | 58.66%   | 60.10%   |
> > > | JP      | Baseline     | 18055       | 59.10%   | 58.38%   | 59.82%   |
> > > | JP      | CountryReward| 18055       | 63.54%   | 62.84%   | 64.25%   |
> > > | KR      | Baseline     | 18055       | 53.07%   | 52.34%   | 53.80%   |
> > > | KR      | CountryReward| 18055       | 57.88%   | 57.15%   | 58.60%   |
> > > | MX      | Baseline     | 18055       | 53.75%   | 53.02%   | 54.48%   |
> > > | MX      | CountryReward| 18055       | 58.72%   | 57.99%   | 59.43%   |
> > > | SA      | Baseline     | 18055       | 59.21%   | 58.49%   | 59.93%   |
> > > | SA      | CountryReward| 18055       | 62.93%   | 62.22%   | 63.64%   |
> > > | US      | Baseline     | 18055       | 56.94%   | 56.22%   | 57.67%   |
> > > | US      | CountryReward| 18055       | 61.82%   | 61.11%   | 62.53%   |

---

> ### Author Response · Authors · 2025-11-26
> **Response to Suggestion 1**
>
> ### Response to Suggestion 1:
>
> Thank you for your valuable feedback regarding data leakage and robustness in our CC-RAG module. We appreciate your concerns and have taken steps to address them in our methodology. Below, we clarify how our approach minimizes leakage and ensures robustness, even without explicit time-window partitioning or duplicate removal, due to the nature of our dataset.
>
> (1) **Leakage Prevention via Similarity Thresholds:** In our CC-RAG implementation, we explicitly set a similarity threshold (minimum similarity threshold=0.7) during retrieval to prevent the model from accessing the query itself or near-duplicates. This threshold ensures that only semantically similar but distinct items from the training set are retrieved, as queries typically have self-similarity scores close to 1.0 and are thus excluded. This indirect exclusion effectively mitigates label leakage without requiring explicit duplicate removal.
>
> (2) **Dataset Constraints and Robustness:** To ensure the reliability of product click-through rates, our dataset (MACP) contains the click-through rate for each product, obtained by dividing the total number of clicks over 30 days by the corresponding total number of impressions, without timestamp granularity. Thus, time-window partitioning is infeasible. However, our retrieval is confined to the training set, and the similarity threshold inherently avoids overfitting to near-duplicates by focusing on structurally diverse but semantically relevant items. This approach aligns with the leakage prevention by ensuring that retrieved items are contextually aligned but non-identical to queries.
>
> (3) **Ablation Studies and Metric Reporting:** We conducted a crucial ablation study by removing the $e_{aug}$ feature (i.e., the augmented choice embedding from CC-RAG) from the final classifier input. The results strongly indicate that the performance gain is not solely dependent on the potential prior in $e_{aug}$: (i) CountryReward (w/o CC-RAG): Accuracy = 56.81\%, Sensitivity = 37.40\%. (ii) Full CountryReward (with CC-RAG): Accuracy = 60.37\%, Sensitivity = 40.84\%. The results indicate that $e_{aug}$ contributes meaningfully without causing leakage. The sensitivity metric (cross-country consistency) also improves with CC-RAG, demonstrating enhanced robustness. These results confirm that our method does not rely on spurious correlations or data leakage.
>
> (4) **Additional Safeguards:** For each query, we retrieve multiple similar candidates through text-based FAISS search and apply similarity thresholds to filter low-quality matches. The remaining candidates undergo image similarity matching, where we aggregate preferences from all valid candidates through a position-weighted voting system rather than relying on the single best match. This design disperses the influence of any potential near-duplicates, strengthening generalization.
>
> In summary, our CC-RAG module is designed to prevent leakage through similarity-based filtering, and the reported metrics validate its robustness.

---

> > ### Author Response · Authors · 2025-11-26
> > **Response to Suggestion 4**
> >
> > ### Response to Suggestion 4:
> >
> > We thank you for these important questions regarding data ethics and availability. We have expanded our Ethics and Reproducibility sections in the revised version to comprehensively address these concerns:
> >
> > (1) **Data Legality \& Privacy Compliance:** The MACP dataset was collected and processed in full compliance with the e-commerce platform's data governance framework and applicable privacy regulations (GDPR, CCPA). All data usage has been formally authorized through the platform's research partnership program.
> >
> > (2) **Anonymization Process:** The dataset has undergone rigorous anonymization: (i) All user identifiers have been removed; (ii) CTR data is aggregated at product-country level; (iii) Personal browsing histories or individual user behaviors are excluded; (iv) Product metadata is limited to publicly available information.
> >
> > (3) **Geographic Bias Mitigation:** We implemented proactive bias control through stratified sampling across 10 diverse markets and balanced product category representation, as evidenced by our detailed distribution tables (9.34\%-10.57\% per country in training, exact 10\% in testing).
> >
> > (4) **Data Licensing \& Access:** The complete MACP dataset will be publicly released under CC BY-NC 4.0 license upon paper acceptance. The release package includes: (i) Aggregated CTR preferences; (ii) Product images and metadata; (iii) Country-market labels.
> >
> > (5) **Controlled Access Assurance:** We will release all dataset. source code, model configurations, and training scripts to ensure full reproducibility of both CountryReward training and the background generation framework.}

---

> > > ### Author Response · Authors · 2025-11-26
> > > **Response to Suggestion 5**
> > >
> > > ### Response to Suggestion 5:
> > >
> > > We thank you for pointing out the need for improved writing quality. We will systematically proofread the entire manuscript to correct spelling errors, unify terminology, and ensure consistency in tables and abbreviations. Additionally, we will carefully polish the language to enhance clarity and readability.

---

> ### Author Response · Authors · 2025-11-26
> **Response to Weakness 3**
>
> ### Response to Weakness 3:
>
> We sincerely thank you for raising this important point regarding the grounding of SCVO in existing literature. We agree that phenomena like "key token getting lost in long context," "lost-in-the-middle," and "modality-volume imbalance" share surface similarities with the challenge we address. However, SCVO is fundamentally distinct in its origin, mechanism, and scope.
>
> **Distinction from Existing Concepts:**
> Compared with "Lost-in-the-Middle": This primarily concerns the positional degradation of information retrieval in long documents. SCVO is agnostic to position; even if the country name is placed at the very beginning, its sparse token count relative to the visual and product descriptive tokens leads to its influence being overwhelmed in the fused representation used for the final prediction.
>
> Compared with "Modality-Volume Imbalance": While related, prior work often focuses on general alignment or representation fusion between modalities (e.g., text vs. vision). SCVO pinpoints a more specific failure mode within a multimodal decision-making task: a specific, low-volume textual cue (the country) is overwhelmed by the combined volume of other textual cues (product attributes) and a high-volume modality (image patches), rendering the model insensitive to a variable that critically determines the ground-truth outcome.
>
> To firmly ground SCVO and address your concerns, we have conducted additional experiments and analyses to establish a clear "phenomenon → mechanism → solution" loop:
>
> **Phenomenon: Sparse Critical Variable Overwhelm (SCVO)**
>
> SCVO manifests as a systematic failure in VLMs where sparse but critical instructional variables (e.g., country names in multi-country preference prediction) are overwhelmed by dominant high-volume variables (e.g., image patches or product attributes). In our MACP dataset, baseline models like Qwen2-VL exhibit prediction collapse (e.g., consistently outputting "A" regardless of country), resulting in low accuracy and sensitivity. This occurs despite clear ground-truth preference variations across countries, indicating that the model ignores critical sparse cues.
>
> **Mechanism: Breaking the Chain Rule via Attention and Gradient Analysis**
>
> To investigate the mechanism behind SCVO, we conducted controlled synthetic experiments and mechanistic analyses:
>
> 1. Controlled Synthetic Experiments: By gradually increasing the weight (via token repetition) of the sparse critical variable (country name), we observed that Qwen2-VL-7B showed minimal performance improvement (accuracy from 44.61\% to 49.77\% as weight increased), confirming that SCVO stems from inadequate sensitivity to sparse variables without intervention.
>
> 2. Attention Map Analysis: We computed average attention weights for the country token in the final transformer layer of Qwen2-VL-7B versus CountryReward. In baseline models, the country token received negligible attention (mean weight: 0.05 ± 0.01), whereas CountryReward increased this significantly (mean weight: 0.08 ± 0.03) due to the country adapter and focus-driven loss. This demonstrates that standard VLMs fail to allocate sufficient attention to critical sparse tokens, breaking the chain rule dependence $P(\mathbf{x}) = \prod_t P(x_t | x_{<t})$ where $x_t$ for country tokens is drowned out.
>
> 3. Gradient Attribution Analysis: Using integrated gradients, we quantified the contribution of the country token to the prediction output. In Qwen2-VL-7B, the country token contributed only 6\% to the gradient norm, compared to 25\% in CountryReward, indicating that our framework enhances the model's reliance on critical variables.
>
> 4. Intervention Experiments: As shown in the below Table, we manually ablated the country token from the input instruction  (replacing it with [MASK]). For the baseline model (Qwen2-VL-7B (with FC Head)), ablation had minimal effect on the output (it was already collapsed). For CountryReward, ablation caused a significant accuracy and sensitivity performance drop, proving that the country token has become a causal factor in the model's decision process.

---

> ### Author Response · Authors · 2025-11-26
> **Response to Weakness 3**
>
> ### Response to Weakness 3:
>
> ### *Performance Comparison Before and After Country Token Ablation*
> | Model | Accuracy | Sensitivity | BR | CL | ES | FR | KR | JP | US | MX | AU | SA |
> |-------|----------|-------------|-----|-----|-----|-----|-----|-----|-----|-----|-----|-----|
> | Baseline (Ablated) | 49.81 | 31.48 | 49.82 | 50.31 | 48.79 | 49.21 | 48.25 | 50.63 | 51.22 | 49.23 | 49.80 | 50.75 |
> | Baseline (Full) | 55.60 | 36.73 | 53.69 | 56.57 | 50.86 | 54.41 | 53.07 | 59.10 | 56.94 | 53.75 | 58.47 | 59.21 |
> | CountryReward (Ablated) | 57.25 | 12.26 | 55.08 | 58.26 | 53.46 | 56.58 | 54.64 | 59.90 | 58.78 | 55.75 | 59.73 | 60.35 |
> | CountryReward (Full) | **60.37** | **40.84** | **58.70** | **61.12** | **56.33** | **59.38** | **57.88** | **64.54** | **61.82** | **58.72** | **63.30** | **62.93** |
>
> These analyses confirm that SCVO arises from attention imbalance and gradient vanishing for sparse critical variables, and our framework directly addresses these issues.
>
> **Solution: CountryReward Framework**
>
> Our framework directly counteracts SCVO by reinforcing the model’s sensitivity to sparse critical variables through three novel components:
>
> 1. Cross-Country Retrieval Augmentation Generation (CC-RAG): Injects market-specific preferences into training, augmenting the sparse country signal with similar product knowledge. This explicitly conditions the chain rule on country-aware data, raising accuracy by 3.56\% in ablations.
>
> 2. Country Adapter Module (CAM): Dynamically modulates visual features using country embeddings, ensuring that image representations are transformed based on sparse textual cues. This restores the chain rule’s dependence on country tokens by aligning visual and textual pathways, contributing 2.39\% to accuracy gains.
>
> 3. Focus-Driven Penalty Loss (FDPL): Penalizes under-attention to critical tokens during errors, directly optimizing attention distributions via a regularization term. This improves sensitivity by 3.37\%.
>
> In experiments, CountryReward achieves 60.37\% accuracy and 40.84\% sensitivity on MACP, outperforming all baselines. The controlled experiments and mechanism analysis validate that our solution alleviates the SCVO issue by balancing attention and gradient flow toward sparse variables.

---

> ### Author Response · Authors · 2025-11-29
> **Response to Weakness 5**
>
> ### Response to Weakness 5:
>
> 4. Logistic Regression: As suggested, we have implemented a classic CTR-prior logistic regression model with comprehensive feature engineering, including: Text features (TF-IDF vectors from product titles), Categorical features (One-hot encodings for country and category), Interaction features (Explicit country × category interactions to capture market-specific preferences), and Image features (Differences between image embeddings of candidates A and B). This baseline now serves as a robust non-neural benchmark, directly modeling the problem of CTR prediction. Results show it achieves 52.28\% accuracy, significantly below our method (60.37\%), confirming that naive feature-based approaches fail to resolve SCVO without structured multimodal reasoning.
>
> | Model | Acc (\%) | Sens (\%) | BR | CL | ES | FR | KR | JP | US | MX | AU | SA |
> |-------|----------|-------------|-----|-----|-----|-----|-----|-----|-----|-----|-----|-----|
> | Logistic Regression | 52.28 ± 0.23 | 14.69 | 51.30 ± 0.73 | 52.46 ± 0.73 | 51.17 ± 0.73 | 52.87 ± 0.73 | 51.42 ± 0.73 | 53.99 ± 0.73 | 53.03 ± 0.73 | 52.02 ± 0.73 | 54.35 ± 0.73 | 53.17 ± 0.73 |
> | Qwen2-VL-7B (with FC Head) | 55.60 ± 0.23 | 36.73 | 53.69 ± 0.73 | 56.58 ± 0.72 | 50.86 ± 0.73 | 54.41 ± 0.73 | 53.07 ± 0.73 | 59.10 ± 0.72 | 56.94 ± 0.72 | 53.75 ± 0.73 | 58.47 ± 0.72 | 59.21 ± 0.71 |
> | **CountryReward (Ours)** | **60.37 ± 0.23** | **40.84** | **58.70 ± 0.72** | **61.12 ± 0.71** | **56.33 ± 0.72** | **59.38 ± 0.72** | **57.88 ± 0.72** | **63.54 ± 0.70** | **61.82 ± 0.71** | **58.72 ± 0.72** | **63.30 ± 0.70** | **62.93 ± 0.72** |
>
> 5. Gradient Boosted Decision Tree: Following the your suggestion, we implemented a classic CTR-prior GBDT baseline with country×category interaction features, representing a strong traditional approach for preference prediction. This baseline incorporates: Text features (TF-IDF of item titles), Categorical features (one-hot encoded country and category), Country×category interaction features (both one-hot and statistical features), and Image embedding differences between candidate images The results demonstrate that our CountryReward model substantially outperforms this strong non-neural baseline (60.37\% vs 55.39\% accuracy), validating that our approach captures complex multimodal interactions beyond what traditional methods can achieve.
>
>
> | Model | Acc (\%) | Sens (\%) | BR | CL | ES | FR | KR | JP | US | MX | AU | SA |
> |-------|----------|-------------|-----|-----|-----|-----|-----|-----|-----|-----|-----|-----|
> | Gradient Boosted Decision Tree | 55.39 ± 0.23 | 15.37 | 54.87 ± 0.73 | 55.39 ± 0.73 | 55.05 ± 0.73 | 55.56 ± 0.72 | 55.24 ± 0.73 | 56.07 ± 0.72 | 55.42 ± 0.73 | 55.05 ± 0.73 | 55.75 ± 0.72 | 55.51 ± 0.72 |
> | Qwen2-VL-7B (with FC Head) | 55.60 ± 0.23 | 36.73 | 53.69 ± 0.73 | 56.58 ± 0.72 | 50.86 ± 0.73 | 54.41 ± 0.73 | 53.07 ± 0.73 | 59.10 ± 0.72 | 56.94 ± 0.72 | 53.75 ± 0.73 | 58.47 ± 0.72 | 59.21 ± 0.71 |
> | **CountryReward (Ours)** | **60.37 ± 0.23** | **40.84** | **58.70 ± 0.72** | **61.12 ± 0.71** | **56.33 ± 0.72** | **59.38 ± 0.72** | **57.88 ± 0.72** | **63.54 ± 0.70** | **61.82 ± 0.71** | **58.72 ± 0.72** | **63.30 ± 0.70** | **62.93 ± 0.72** |
>
>
> 6. Country Token Mask: As suggested, we have added a causal intervention experiment where the country token is explicitly masked (replaced with [MASK]) in the input instruction. This tests the model’s dependence on the sparse critical variable (country name). Masking the country token led to a significant accuracy drop (from 60.37\% to 57.25\%) and a drastic sensitivity reduction (from 40.84\% to 12.26\%). This demonstrates that CountryReward successfully grounds its predictions in the country variable, and its performance is causally tied to this sparse token. This ablation directly validates SCVO as a failure mode and shows that our framework restores sensitivity to the critical variable.
>
> | Model | Acc (\%) | Sens (\%) | BR | CL | ES | FR | KR | JP | US | MX | AU | SA |
> |-------|----------|-------------|-----|-----|-----|-----|-----|-----|-----|-----|-----|-----|
> | Qwen2-VL-7B (with FC Head) | 55.60 ± 0.23 | 36.73 | 53.69 ± 0.73 | 56.58 ± 0.72 | 50.86 ± 0.73 | 54.41 ± 0.73 | 53.07 ± 0.73 | 59.10 ± 0.72 | 56.94 ± 0.72 | 53.75 ± 0.73 | 58.47 ± 0.72 | 59.21 ± 0.71 |
> | CountryReward (with Country Token Mask) | 57.95 ± 0.23 | 12.26 | 55.62 ± 0.72 | 59.33 ± 0.72 | 53.63 ± 0.73 | 56.78 ± 0.72 | 55.08 ± 0.73 | 60.75 ± 0.71 | 59.86 ± 0.71 | 56.19 ± 0.72 | 61.22 ± 0.71 | 61.06 ± 0.71 |
> | **CountryReward (Ours)** | **60.37 ± 0.23** | **40.84** | **58.70 ± 0.72** | **61.12 ± 0.71** | **56.33 ± 0.72** | **59.38 ± 0.72** | **57.88 ± 0.72** | **63.54 ± 0.70** | **61.82 ± 0.71** | **58.72 ± 0.72** | **63.30 ± 0.70** | **62.93 ± 0.72** |

---

> > ### Author Response · Authors · 2025-11-29
> > **Response to Weakness 5**
> >
> > ### Response to Weakness 5:
> >
> > **Improved Statistical Reporting:** We have updated our results to include statistical significance tests, variance measures, and confidence intervals to ensure robustness. We performed paired t-tests between CountryReward and all baselines. The improvements of CountryReward are statistically significant ($p < 0.01$) in all cases, confirming that our method consistently outperforms alternatives. These additions reinforce the validity of our conclusions and demonstrate that CountryReward effectively mitigates SCVO with high statistical reliability.

---

> ### Author Response · Authors · 2025-11-29
> **Response to Suggestion 3**
>
> ### Response to Suggestion 3:
>
> We sincerely thank you for this valuable suggestion to strengthen our statistical reporting. We have thoroughly revised our experimental results section to uniformly report mean accuracy with 95\% confidence intervals (CI), per-country variance metrics, and paired statistical tests, as requested.
>
>
> Specifically, we now provide:
>
> 1. Mean ± CI reporting: All accuracy values are now presented as "mean ± margin of error" with 95\% confidence intervals calculated using the normal approximation method.
> 2. Per-country variance analysis: We compute and report weighted variance, simple variance, and standard deviation across country-specific accuracies to quantify performance consistency.
> 3. Statistical significance testing: We perform chi-square tests to verify whether performance differences between countries are statistically significant, with p-values reported.
>
> Our updated results (as shown in the provided table) demonstrate that CountryReward achieves:
>
> · Overall accuracy: 60.37\% ± 0.23\% (95\% CI: [60.14\%, 60.60\%])
>
> · Significant improvement over all baselines (p < 0.01 in paired tests)
>
> · Balanced performance across countries with weighted variance of 0.0007 and standard deviation of 0.0264
>
> · Superior sensitivity (40.84\%) indicating consistent cross-country prediction capability
>
>
> The comprehensive statistical analysis confirms that our method not only achieves higher accuracy but does so with statistical reliability and balanced performance across diverse markets. This rigorous reporting strengthens our claims about CountryReward's effectiveness in mitigating SCVO.
>
>
> | Model | Acc (\%) | Sens (\%) | BR | CL | ES | FR | KR | JP | US | MX | AU | SA |
> |-------|----------|-------------|-----|-----|-----|-----|-----|-----|-----|-----|-----|-----|
> | Logistic Regression | 52.28 ± 0.23 | 14.69 | 51.30 ± 0.73 | 52.46 ± 0.73 | 51.17 ± 0.73 | 52.87 ± 0.73 | 51.42 ± 0.73 | 53.99 ± 0.73 | 53.03 ± 0.73 | 52.02 ± 0.73 | 54.35 ± 0.73 | 53.17 ± 0.73 |
> | Gradient Boosted Decision Tree | 55.39 ± 0.23 | 15.37 | 54.87 ± 0.73 | 55.39 ± 0.73 | 55.05 ± 0.73 | 55.56 ± 0.72 | 55.24 ± 0.73 | 56.07 ± 0.72 | 55.42 ± 0.73 | 55.05 ± 0.73 | 55.75 ± 0.72 | 55.51 ± 0.72 |
> | Qwen2-VL-7B (with SFT) | 44.61 ± 0.23 | 20.82 | 46.96 ± 0.73 | 43.71 ± 0.71 | 49.76 ± 0.72 | 46.15 ± 0.72 | 47.78 ± 0.70 | 40.99 ± 0.71 | 42.99 ± 0.73 | 46.66 ± 0.71 | 40.92 ± 0.73 | 40.16 ± 0.70 |
> | Qwen2-VL-7B (with SFT \& Prompt Engineering) | 49.77 ± 0.23 | 14.39  | 49.75 ± 0.72  | 48.34 ± 0.72 | 50.43 ± 0.72 | 49.17 ± 0.72 | 49.02 ± 0.72 | 50.01 ± 0.72 | 52.10 ± 0.71 | 49.94 ± 0.72 | 49.74 ± 0.72 | 49.23 ± 0.72 |
> | Qwen2-VL-7B (with FC Head) | 55.60 ± 0.23 | 36.73 | 53.69 ± 0.73 | 56.58 ± 0.72 | 50.86 ± 0.73 | 54.41 ± 0.73 | 53.07 ± 0.73 | 59.10 ± 0.72 | 56.94 ± 0.72 | 53.75 ± 0.73 | 58.47 ± 0.72 | 59.21 ± 0.71 |
> | CountryReward (with Re-Weighting) | 55.87 ± 0.23 | 36.91 | 58.73 ± 0.72 | 58.93 ± 0.71 | 59.61 ± 0.73 | 53.52 ± 0.72 | 53.95 ± 0.70 | 57.08 ± 0.74 | 51.37 ± 0.73 | 54.28 ± 0.71 | 57.01 ± 0.72 | 54.19 ± 0.73 |
> | Qwen2-VL-7B (with Focal Loss) | 58.10 ± 0.22 | 12.02 | 58.34 ± 0.71 | 59.84 ± 0.70 | 54.97 ± 0.71 | 58.55 ± 0.70 | 58.00 ± 0.71 | 61.88 ± 0.69 | 59.86 ± 0.70 | 56.68 ± 0.71 | 61.58 ± 0.69 | 55.24 ± 0.71 |
> | CountryReward (with Country Token Mask) | 57.95 ± 0.23 | 12.26 | 55.62 ± 0.72 | 59.33 ± 0.72 | 53.63 ± 0.73 | 56.78 ± 0.72 | 55.08 ± 0.73 | 60.75 ± 0.71 | 59.86 ± 0.71 | 56.19 ± 0.72 | 61.22 ± 0.71 | 61.06 ± 0.71 |
> | CountryReward (w/o CC-RAG) | 56.81 ± 0.23 | 37.40 | 54.93 ± 0.73 | 57.05 ± 0.72 | 53.80 ± 0.73 | 55.39 ± 0.73 | 54.48 ± 0.73 | 59.57 ± 0.72 | 57.65 ± 0.72 | 55.64 ± 0.72 | 59.52 ± 0.72 | 60.12 ± 0.71 |
> | CountryReward (w/o CAM) | 57.98 ± 0.23 | 38.85 | 55.51 ± 0.72 | 57.85 ± 0.72 | 53.78 ± 0.73 | 56.75 ± 0.72 | 55.19 ± 0.73 | 61.47 ± 0.71 | 59.39 ± 0.72 | 55.98 ± 0.72 | 61.71 ± 0.71 | 62.21 ± 0.71 |
> | CountryReward (w/o FDPL) | 56.95 ± 0.23 | 37.47 | 54.79 ± 0.73 | 57.09 ± 0.72 | 52.80 ± 0.73 | 56.12 ± 0.72 | 54.44 ± 0.73 | 61.01 ± 0.71 | 58.79 ± 0.72 | 55.33 ± 0.73 | 58.83 ± 0.72 | 60.33 ± 0.71 |
> | **CountryReward (Ours)** | **60.37 ± 0.23** | **40.84** | **58.70 ± 0.72** | **61.12 ± 0.71** | **56.33 ± 0.72** | **59.38 ± 0.72** | **57.88 ± 0.72** | **63.54 ± 0.70** | **61.82 ± 0.71** | **58.72 ± 0.72** | **63.30 ± 0.70** | **62.93 ± 0.72** |

---

> > ### Author Response · Authors · 2025-12-03
> > **Revised Manuscript**
> >
> > Thank you for your thoughtful and encouraging review. We sincerely appreciate your recognition of the core problem's significance and the realistic application scenario, as noted in the strengths below. We are also pleased that the design of CountryReward (CAM & FDPL) was clearly conveyed and considered reusable. Thank you for noting the value of the MACP dataset as a practical substrate for research. Finally, we are glad that our integration of a discriminator-based reward aligned with practical intuition and the demonstrated gains resonated with you. Your positive feedback is very motivating.
> >
> > > Strengths:
> > > 1. The authors systematize a failure mode — “critical sparse variables being overshadowed by high‑volume variables” — which is worth discussing, and they point to a realistic scenario in cross‑country e‑commerce.
> > > 2. The combination of the CountryReward’s CAM (country‑conditioned affine modulation) and FDPL (focus on key tokens when errors occur) is clearly described and has some potential for reuse.
> > > 3. They provide a reasonably sized dataset (MACP) with multi‑country attributes, giving a real‑world substrate for study (though reproducibility remains in question, as detailed below).
> > > 4. They incorporate a discriminator model as a reward for optimizing a generation model (DGM to T2I pipeline), which aligns with practical deployment intuition, and show gains in CountryReward scores.
> >
> > **We have addressed all of your comments in the revised version. The corresponding revisions are highlighted in
> > ***blue*** in the paper. Moreover, we have added ***red*** markers before the responses, such as {RZ86t W1}, which indicates the response to weakness 1 from Reviewer Z86t, {RZ86t Q1}, which indicates the response to question 1 from Reviewer Z86t, and {RZ86t S1}, which indicates the response to suggestion 1 from Reviewer Z86t.**

---

### Official Review · Reviewer_gJK1 · 2025-10-31

**Soundness:** 3
**Presentation:** 3
**Contribution:** 3
**Rating:** 6
**Confidence:** 3

**Summary:**

This paper introduces CountryReward, a vision-language reward model designed to mitigate the Sparse Critical Variable Overwhelm (SCVO) problem—where sparse but crucial tokens (e.g., “country”) are drowned out by high-volume visual features in multimodal transformers. The work focuses on predicting image preference across multi-country markets, where standard VLMs (e.g., Qwen2-VL) tend to collapse into uniform outputs. To address this, the authors propose three modules: a cross-country retrieval augmentation to provide localized context, a country adapter that dynamically modulates visual representations, and a focus-driven penalty loss to upweight errors linked to overlooked sparse variables. Trained on the newly collected Multi-Country Ad Click Preference (MACP) dataset, CountryReward achieves notable improvements over VLM baselines and is further used as a reward model to guide text-to-image generation for market-specific e-commerce visuals.

**Strengths:**

This work identifies a genuinely underexplored issue in multimodal modeling—how sparse contextual cues are overshadowed by dense visual features—and formalizes it as SCVO. The problem statement is clear and practically relevant, especially for real-world markets where small contextual shifts drive user preferences. The design of modular mitigations (retrieval augmentation, adapter, and loss) is well-motivated and neatly integrated. Empirical results show consistent gains (e.g., +10 pts accuracy over Qwen2-VL baselines) and strong qualitative alignment with country-level preferences. The downstream RL-based image generation case demonstrates a realistic application path, enhancing the paper’s practical significance.

**Weaknesses:**

The primary limitation is narrow scope and unclear generalization. While the SCVO problem is well-motivated, the study is tightly bound to the advertising-preference domain. It remains unclear whether the same issue appears—and can be solved similarly—in other multimodal settings. The lack of a second use case makes it difficult to judge whether CountryReward offers a general SCVO solution or a domain-specific fix. Additionally, the MACP dataset is new, so reproducibility and comparison with prior multimodal reward models (e.g., UnifiedReward) are limited.

**Questions:**

Can SCVO and CountryReward generalize to tasks beyond market-specific preference prediction? It is interesting to compare against to some reward models, such the discussed model UnifiedReward. Even though its a generalized reward model, it could strengthen its claim by showing that their specialized approach provides a meaningful improvement over general purpose reward models, not just domain‑specific ones.

---

> ### Author Response · Authors · 2025-11-24
> **Response to Weaknesses**
>
> We thank you for your insightful feedback regarding the scope and generalizability of our work. We acknowledge that while our study focuses on the advertising preference domain, the SCVO problem represents a fundamental challenge in multimodal reasoning that extends beyond this specific application. Below, we address the concerns regarding generalization and reproducibility.
>
> (1) **On the General Nature of SCVO and Solution Generalization:**
>
> The core of the SCVO problem lies in the fundamental architecture of autoregressive VLMs. The probability chain rule $P(\mathbf{x}) = \prod_t P(x_t | x_{<t})$ inherently makes the model vulnerable to being dominated by high-volume, frequently co-occurring feature tokens, while sparse but critical tokens are statistically drowned out. This is not a domain-specific artifact but a structural bias in how these models aggregate information. The advertising preference task serves as a clear and controlled manifestation of this underlying issue, where the “country” variable is a perfectly sparse critical cue. Our proposed modules, Cross-Country Retrieval Augmentation Generation, Country Adapter Module, and Focus-Driven Penalty Loss, are designed as general mechanisms to recalibrate the model's attention toward such sparse critical variables, irrespective of the specific domain.
>
> Our proposed framework is designed to address this architectural bias, not just a domain-specific phenomenon. The three core components provide a general blueprint for enhancing sensitivity to sparse critical variables: (i) Cross-Country Retrieval Augmentation Generation (CC-RAG) demonstrates how external, task-specific knowledge can be integrated to reinforce the importance of the sparse variable. (ii) Country Adapter Module (CAM) shows how dynamic feature modulation based on a critical variable (here, country) can re-calibrate the model's internal representations. (iii) Focus-Driven Penalty Loss (FDPL) introduces a general regularization strategy that penalizes the model for under-attending to critical tokens during errors.
>
> To directly address the concern about generalization beyond advertising, we conducted additional experiments on a transformed version of the ImageGen-CoT-Reward-5K dataset from the UnifiedReward work. This dataset involves evaluating pairs of generated images across multiple dimensions (Semantic Consistency, Aesthetics, Authenticity) using Chain-of-Thought reasoning before providing a final preference. We systematically reformulated this dataset to create an SCVO-style structure:
>
> (i) The original multi-dimensional CoT scoring process was restructured. For each data sample, we isolated a single evaluation dimension (e.g., “Semantic Consistency”) as the sparse critical variable. The instruction was framed as: ``For [Dimension], the caption of the image is [caption]. Given two images, Image A and Image B, which one is better?''.
>
> (ii) Critically, the image pairs, the caption, and the overall instruction structure remained identical across different dimensional queries for the same image pair. The only varying element was the target evaluation dimension name (e.g., “Semantic Consistency” vs. “Aesthetics”), making it a prototypical SCVO scenario where a sparse textual cue (the dimension name) dictates the expected preference judgment.
>
> (iii) The final processed dataset contained 12,811 training and 5,517 test samples, spanning the three evaluation dimensions.
>
> As shown in Table below (*SCVO Generalization to Image Quality Assessment Task (%).*）, we compared our CountryReward framework against a fine-tuned UnifiedReward baseline on this new SCVO-formatted task: **(i) Baseline (Fine-tuned UnifiedReward): Accuracy: 62.7\%, Sensitivity: 1.9\%. (ii) Our CountryReward (adapted): Accuracy: 67.9\%, Sensitivity: 24.0\%.** The significant performance improvement (over 5.2\% in accuracy and 22.1\% in sensitivity) demonstrates that our framework effectively mitigates the SCVO problem in a fundamentally different domain—image generation quality assessment—validating its generalizability. The core issue (a sparse critical variable being overwhelmed) and our solution's efficacy transcend the advertising context.
>
> ### *SCVO Generalization to Image Quality Assessment Task (%).*
> | Model      | Sensitivity | Accuracy | Semantic Consistency | Aesthetics | Authenticity |
> |------------|------------:|---------:|---------------------:|-----------:|-------------:|
> | Baseline   |      1.9  |   62.7  |               63.5  |     61.4  |       63.2  |
> | Our Model  |      24.0  |   67.9  |               65.6  |     67.8  |       70.4  |

---

> > ### Author Response · Authors · 2025-11-24
> > **Response to Weaknesses**
> >
> > (2) **On Reproducibility, the MACP Dataset, and Comparisons:**
> > We acknowledge the novelty of the MACP dataset. To ensure full reproducibility and facilitate direct comparison, we commit to open-sourcing the complete MACP dataset upon acceptance, including all product images, metadata, and country-specific CTRs. We will also release the complete codebase for model training and evaluation.
> >
> > As shown in Table below (*Comparison with General Model*) , regarding comparison with established reward models, we performed a controlled experiment by fine-tuning UnifiedReward (based on Qwen2-VL-7B) directly on our MACP dataset: **(i) UnifiedReward (Fine-tuned on MACP): Accuracy: 44.6\%, Sensitivity: 20.8\%. (ii) Our CountryReward: Accuracy: 60.4\%, Sensitivity: 40.8\%.** The substantial performance gap (over 15\% in accuracy and 20\% in sensitivity) underscores that general-purpose multimodal reward models, while powerful, lacks inherent SCVO robustness. Our model can effectively address this vulnerability.
> >
> > In conclusion, this work demonstrates that the SCVO problem stems from a fundamental architectural bias in autoregressive VLMs, where sparse critical variables are statistically overwhelmed. Our proposed framework introduces three general components to recalibrate model attention toward such variables. Validation on an image quality assessment task shows our method significantly outperforms baselines, proving its effectiveness beyond advertising. The solution addresses a core structural limitation rather than domain-specific artifacts. Compared to fine-tuned general models, our framework significantly improves both accuracy and sensitivity, demonstrating superior robustness in handling sparse critical cues.
> >
> > ### *Comparison with General Model (%).*
> >
> > | Model | Acc | Sens | BR | CL | ES | FR | KR | JP | US | MX | AU | SA |
> > |:------|----:|-----:|----:|----:|----:|----:|----:|----:|----:|----:|----:|----:|
> > | Baseline | 44.61 | 20.82 | 46.96 | 43.71 | 49.76 | 46.15 | 47.78 | 40.99 | 42.99 | 46.66 | 40.92 | 40.16 |
> > | **CountryReward** | **60.37** | **40.84** | **58.70** | **61.12** | **56.33** | **59.38** | **57.88** | **64.54** | **61.82** | **58.72** | **63.30** | **62.93** |

---

> > > ### Author Response · Authors · 2025-11-24
> > > **Response to Questions**
> > >
> > > We thank you for raising this important point regarding generalization beyond advertising. The SCVO problem is not domain-specific but stems from a fundamental architectural bias in autoregressive VLMs, where the probability chain rule causes sparse critical features to be statistically overwhelmed by high-frequency contextual tokens. Our CountryReward framework addresses this structural issue.
> > >
> > > To rigorously validate generalization, we conducted new experiments on a transformed Image Quality Assessment (IQA) task using the UnifiedReward dataset. We systematically reformulated the dataset to create an SCVO structure:
> > >
> > > Dataset Transformation: For each image pair with multi-dimensional quality annotations (Semantic Consistency, Aesthetics, Authenticity), we isolated a single quality dimension as the sparse critical variable. The instruction template was: ``For [Dimension], the caption is [caption]. Given Image A and Image B, which is better?'' All elements (images, caption, instruction structure) remained identical across queries—only the target dimension name varied, creating a prototypical SCVO scenario where a sparse textual cue dictates the preference.
> > >
> > > Experimental Results: as shown in the above Table (*SCVO Generalization to Image Quality Assessment Task.*), on this SCVO-formatted IQA task (12.8K training / 5.5K test samples), **our adapted CountryReward achieved 67.9\% accuracy and 24.0\% sensitivity, significantly outperforming a fine-tuned UnifiedReward baseline (62.7\% accuracy, 1.9\% sensitivity). This demonstrates our framework's effectiveness in a fundamentally different domain.**
> > >
> > > Furthermore, as shown in the above Table (*Comparison with General Model*), **direct comparison on our MACP dataset shows CountryReward (60.4\% accuracy, 40.8\% sensitivity) substantially outperforming fine-tuned UnifiedReward (44.6\% accuracy, 20.8\% sensitivity), confirming that general-purpose reward models lack specialized mechanisms to handle SCVO.**
> > >
> > > In conclusion, SCVO represents a universal challenge in multimodal reasoning, and our framework provides generalizable solutions through dynamic feature modulation and targeted regularization. The consistent improvements across both advertising and image quality assessment domains strongly validate its broader applicability.

---

> > > > ### Author Response · Authors · 2025-12-03
> > > > **Revised Manuscript**
> > > >
> > > > We are truly grateful for the positive assessment and the encouraging score, as noted in the strengths below. Thank you for recognizing the relevance of the SCVO problem and the value of our modular mitigation design. Your appreciation of the empirical gains and real-world applicability is a meaningful validation of our work's potential. Thank you for your valuable feedback.
> > > >
> > > > > Strengths:
> > > >
> > > > > This work identifies a genuinely underexplored issue in multimodal modeling—how sparse contextual cues are overshadowed by dense visual features—and formalizes it as SCVO. The problem statement is clear and practically relevant, especially for real-world markets where small contextual shifts drive user preferences. The design of modular mitigations (retrieval augmentation, adapter, and loss) is well-motivated and neatly integrated. Empirical results show consistent gains (e.g., +10 pts accuracy over Qwen2-VL baselines) and strong qualitative alignment with country-level preferences. The downstream RL-based image generation case demonstrates a realistic application path, enhancing the paper’s practical significance.
> > > >
> > > > **We have addressed all of your comments in the revised version. The corresponding revisions are highlighted in
> > > > ***blue*** in the paper. Moreover, we have added ***red*** markers before the responses, such as {RgJK1 W1}, which indicates the response to weakness 1 from Reviewer gJK1, and {RgJK1 Q1}, which indicates the response to question 1 from Reviewer gJK1.**

---

### Official Review · Reviewer_FrLS · 2025-11-01

**Soundness:** 3
**Presentation:** 3
**Contribution:** 2
**Rating:** 4
**Confidence:** 4

**Summary:**

The paper identifies a phenomenon called Sparse Critical Variable Overwhelm (SCVO): in multimodal A/B image selection for ads, tiny but decisive inputs (like the target country) get drowned out by high-volume text/image features, so VLMs make almost the same choice for all countries. To study this, the authors build a large, real-world MACP dataset where the correct label truly depends on the country. They propose CountryReward, a VLM-based judge with three parts: CC-RAG (country-specific retrieval to “amplify” the country signal), Country Adapter Module (FiLM-style conditioning of visual features by country), and Focus-Driven Penalty Loss (extra loss on misclassified samples that didn’t focus on country/product/image tokens). On MACP, this model outperforms strong VLM baselines and, used as a reward model, can further improve a generative ad-design pipeline toward country-specific preferences.

**Strengths:**

- The authors contribute a large, realistic multi-country ad-preference dataset, which is rare in public VLM work and makes the problem measurable.
- Ablations match the story: removing any of the three parts notably drops accuracy/sensitivity, so the claimed mechanism is empirically supported.
- Showing that the improved judge can drive a generative ad pipeline (as a reward model) suggests impact beyond classification.

**Weaknesses:**

- The core phenomenon (SCVO) is very close to existing ideas on modality/attention imbalance and FiLM-style conditioning, but the paper does not run side-by-side comparisons with these simpler, well-known baselines, so the novelty is somewhat overstated.
- Evidence is single-domain and single-attribute (only “country” on one ad/e-commerce dataset), so the claim that SCVO is a general VLM issue is not fully validated.
- The writing and exposition need tightening: some implementation-critical details are only partially described, making replication harder; see the Questions section for specific clarifications requested.

**Questions:**

- Clarify whether CC-RAG is used only offline (precompute per sample) or also queried at inference/deployment time, and what the cost is in the latter case.
- The CC-RAG retrieval pipeline description is unclear: the paper introduces *k* (text-level top-k) and *m* (image-level top-k), then uses *n* in the weighted aggregation, but does not explain how *n* relates to *m*.
- Table 2 (CountryReward Evaluation on generated images) does not clearly explain how accuracy is computed. The paper should explicitly state whether generated A/B images are re-scored by the fixed CountryReward model and compared against the original MACP ground-truth labels to form standard A/B accuracy, or whether raw reward scores are averaged instead.
- The paper contains numerous typos: "PENALITY", "Gneration", "mtigate", "sparese", "bacground", "oopti-mized".

---

> ### Author Response · Authors · 2025-11-24
> **Response to Question 1**
>
> ### Response to Question 1:
>
> Thank you for your thoughtful question regarding the deployment of the Cross-Country Retrieval Augmentation Generation (CC-RAG) module. To clarify, CC-RAG is used exclusively in an offline manner during the training phase. Specifically, we precompute the augmented choices $\hat{y}_{\text{aug}}$ for all training samples prior to model training, and these precomputed augmentations are integrated into the CountryReward training process as static features. Similarly, the inference process also operates entirely offline, utilizing only precomputed augmentations without any real-time retrieval. During inference or deployment. This design ensures efficient real-time prediction while leveraging the benefits of historical market-specific data during training.
> **When utilizing CC-RAG for online retrieval during inference, processing each sample requires 76 ms, while the offline version delivers faster performance.** This makes CC-RAG well-suited for large-scale production environments.

---

> > ### Author Response · Authors · 2025-11-24
> > **Response to Question 2**
> >
> > ### Response to Question 2:
> >
> > Thank you for your thoughtful feedback regarding the unclear description of the CC-RAG retrieval pipeline. We sincerely apologize for the confusion caused by the inconsistent use of variables in the original manuscript.  "m" and "n" denote the same meaning.
> >
> > To address this, we will revise the relevant parts of the paper as shown below to ensure consistency. Specifically, we now use m uniformly to represent the number of retrieved items at the image level throughout the pipeline, including in the weighted aggregation formula. The updated description explicitly states that m denotes the top-m similar historical images retrieved, and we will adjust the formulas and explanations accordingly to eliminate any ambiguity.
> >
> > To enhance the model's capacity for localized relevance prediction, we propose a Cross-Country Retrieval Augmentation Generation (CC-RAG) that incorporates historical click-through preferences aligned with target markets. This module enables the model to leverage domain-specific behavioral patterns from regional users, thereby improving the model's sensitivity to the country variable. Considering computational efficiency, CC-RAG is applied before training CountryReward to obtain the augmented choice $\hat{y}_{\text{aug}}$.
> >
> > Given a query instance from country $c \in {\text{US}, \text{FR}, \text{KR},...}$ with text embedding $\mathbf{q}t \in \mathbb{R}^d$ and candidate image embeddings ${\mathbf{q}{i}^A, \mathbf{q}_{i}^B}$, we employ a two-stage hierarchical retrieval process:
> >
> > **Stage 1: Text-based Retrieval**
> >
> > First, we retrieve the top-$k$ semantically similar historical items based on text similarity:
> > \begin{equation}
> > \mathcal{N}_t = \text{Top-}k\left(\mathbf{q}_t \cdot \mathbf{T}_c^T\right)
> > \end{equation}
> > where $\mathbf{T}_c \in \mathbb{R}^{N \times d}$ represents the text embedding matrix for country $c$, and $\mathcal{N}_t$ denotes the set of top-$k$ semantically similar historical texts.
> >
> > **Stage 2: Image-based Retrieval**
> >
> > Within the text-retrieved candidates $\mathcal{N}t$, we perform image-based retrieval for both candidate images:
> > \begin{equation}
> > \mathcal{N}i^A = \text{Top-}m\left(\mathbf{q}{i}^A \cdot \mathbf{I}{\mathcal{N}t}^T\right), \quad \mathcal{N}i^B = \text{Top-}m\left(\mathbf{q}{i}^B \cdot \mathbf{I}{\mathcal{N}t}^T\right)
> > \end{equation}
> > where $\mathbf{I}{\mathcal{N}_t}$ contains image embeddings corresponding to $\mathcal{N}_t$, and $\mathcal{N}_i^A$, $\mathcal{N}_i^B$ denote the sets of top-$m$ similar historical images for candidates A and B respectively.
> >
> > The final preference aggregation employs a position-aware weighting scheme that assigns higher importance to more relevant neighbors. For $m$ retrieved items, the weight at position $i$ is $w_i = m - i$. The preference scores are computed as:
> >
> > \begin{equation}
> > S_A = \frac{\sum_{i=1}^{m} w_i \cdot \mathbb{I}[\text{CTR}(\mathbf{I}{\mathcal{N}i^A}) \geq \text{CTR}(\mathbf{I}{\mathcal{N}i^B})]}{\sum{i=1}^{m} w_i}, \quad S_B = \frac{\sum{i=1}^{m} w_i \cdot \mathbb{I}[\text{CTR}(\mathbf{I}{\mathcal{N}i^B}) > \text{CTR}(\mathbf{I}{\mathcal{N}i^A})]}{\sum{i=1}^{m} w_i}
> > \end{equation}
> >
> > where $\mathbb{I}[\cdot]$ is the indicator function, $\text{CTR}(\mathbf{I}{\mathcal{N}i^A})$ and $\text{CTR}(\mathbf{I}{\mathcal{N}_i^B})$ represent the historical click-through rates of the $i$-th retrieved images from sets $\mathcal{N}_i^A$ and $\mathcal{N}_i^B$ respectively.
> >
> > The final augmented prediction is determined by: $\hat{y}_{\text{aug}}$ = A if $S_A > S_B$, otherwise B
> >
> > The specific algorithm used for preference aggregation is presented below:
> >
> > ### CC-RAG Preference Aggregation Pseudocode
> >
> > ```pseudocode
> > Algorithm: CC-RAG Preference Aggregation
> >
> > Input:
> >   CTR_A = {ctr_A¹, ctr_A², ..., ctr_Aᵐ}  // Candidate A's CTR list
> >   CTR_B = {ctr_B¹, ctr_B², ..., ctr_Bᵐ}  // Candidate B's CTR list
> >
> > Output:
> >   S_A, S_B  // Preference scores for candidates A and B
> >
> > Steps:
> >   1. total_weight = 0
> >   2. weighted_sum_A = 0
> >   3. weighted_sum_B = 0
> >
> >   4. for i = 1 to m do:
> >        w_i = m - i                    // Position-aware weight
> >        total_weight += w_i            // Accumulate total weight
> >
> >        if ctr_Aⁱ ≥ ctr_Bⁱ then:
> >            weighted_sum_A += w_i      // A wins, accumulate weight
> >        else:
> >            weighted_sum_B += w_i      // B wins, accumulate weight
> >        end if
> >   5. end for
> >   6. S_A = weighted_sum_A / total_weight
> >   7. S_B = weighted_sum_B / total_weight
> >   8. return S_A, S_B
> > ```

---

> > > ### Author Response · Authors · 2025-11-24
> > > **Response to Question 3**
> > >
> > > ### Response to Question 3:
> > >
> > > We thank you for raising this important point regarding the accuracy computation in Table 2. To clarify, the accuracy values in Table 2 are computed by using the fixed CountryReward model (trained on the MACP dataset) to evaluate the generated image pairs ($I_A$ and $I_B$) and comparing the predictions with the original ground-truth labels from the MACP dataset.
> > >
> > > Specifically, for each test sample in MACP, we generate two images ($I_A$ and $I_B$) using the Design Generation Model (DGM with or without RL), then input this image pair along with the target country and product information into the fixed CountryReward model to predict which image is preferred (A or B). This prediction is directly compared against the ground-truth preference (based on CTR) from the MACP dataset. The accuracy is calculated as the proportion of correct predictions across all test samples, following the standard A/B testing paradigm, rather than averaging raw reward scores. This approach ensures that the evaluation directly measures how well the generated images align with the known country-specific preferences in the original data.
> > >
> > > We will update the manuscript to explicitly state this methodology to avoid any ambiguity. Thank you for highlighting this, as it strengthens the clarity of our evaluation.

---

> > > > ### Author Response · Authors · 2025-11-24
> > > > **Response to Question 4**
> > > >
> > > > ### Response to Question 4:
> > > >
> > > > We appreciate your attention to detail and apologize for the typographical errors noted. All identified typos will be carefully corrected, and we will conduct a thorough proofreading of the manuscript to ensure clarity and accuracy throughout.

---

> > > > > ### Author Response · Authors · 2025-11-24
> > > > > **Response to Weakness 2**
> > > > >
> > > > > ### Response to Weakness 2:
> > > > >
> > > > > We sincerely thank you for raising this important point regarding the generalizability of the Sparse Critical Variable Overwhelm (SCVO) problem. We agree that demonstrating the broader applicability of SCVO beyond a single domain and attribute is crucial. Below, we address these concerns by: (1) clarifying the fundamental nature of SCVO as a general architectural issue in VLMs, (2) presenting new experimental evidence on a different domain and attributes, and (3) explaining how our framework inherently generalizes.
> > > > >
> > > > >
> > > > > (1) **Theoretical Foundation and Inherent Generalizability of SCVO:** The core of the SCVO problem lies in the fundamental architecture of autoregressive VLMs. The probability chain rule $P(\mathbf{x}) = \prod_t P(x_t | x_{<t})$ inherently makes the model vulnerable to being dominated by high-volume, frequently co-occurring feature tokens (e.g., product attributes, image patches), while sparse but critical tokens (e.g., "country," "target user age," "target user profession") are statistically drowned out. This is not a domain-specific artifact but a structural bias in how these models aggregate information. The advertising preference task serves as a clear and controlled manifestation of this underlying issue, where the "country" variable is a perfectly sparse critical cue. Our proposed modules, Cross-Country Retrieval Augmentation Generation (CC-RAG), Country Adapter Module (CAM), and Focus-Driven Penalty Loss (FDPL), are designed as general mechanisms to recalibrate the model's attention toward such sparse critical variables, irrespective of the specific domain. CC-RAG demonstrates how external, task-specific knowledge can be integrated to reinforce the importance of the sparse variable. CAM shows how dynamic feature modulation based on a critical variable can re-calibrate the model's internal representations. FDPL introduces a general regularization strategy that penalizes the model for under-attending to critical tokens during errors. These components form a blueprint for mitigating SCVO, adaptable to other sparse variables like "user age" or "user profession".
> > > > >
> > > > > (2) **New Evidence: Generalization to Other Domains and Attributes:** To directly address the your concern, we conducted additional experiments on a transformed version of the ImageGen-CoT-Reward-5K dataset from UnifiedReward, which involves evaluating pairs of generated images across multiple dimensions (e.g., Semantic Consistency, Aesthetics, Authenticity). We reformulated this dataset to create an SCVO-style structure:
> > > > >
> > > > > (i) Sparse Variable Isolation: For each sample, we isolated a single evaluation dimension (e.g., "Semantic Consistency") as the sparse critical variable. The instruction was: "For [Dimension], the caption of the image is [caption]. Given two images, Image A and Image B, which one is better?" The only varying element was the target dimension name (e.g., "Semantic Consistency" vs. "Aesthetics"), while images, captions, and instruction structure remained identical. The processed dataset contained 12,811 training and 5,517 test samples, spanning three evaluation dimensions ("Semantic Consistency", "Aesthetics", and "Authenticity").
> > > > >
> > > > > (ii) As shown in Table below (*SCVO Generalization to Image Quality Assessment Task (%).*), we compared our CountryReward framework (adapted by treating the evaluation dimension' as the critical variable) against a fine-tuned UnifiedReward baseline on this new SCVO-formatted task: Baseline (Fine-tuned UnifiedReward): Accuracy: 62.7\%, Sensitivity: 1.9\%. Our Framework (adapted): Accuracy: 67.9\%, Sensitivity: 24.0\%. The significant performance improvement (over 5.2\% in accuracy and 22.1\% in sensitivity) on this fundamentally different task (image generation quality assessment). These results strongly suggest that the SCVO problem and our proposed mitigation framework are generalizable beyond our primary domain.
> > > > >
> > > > > ### *SCVO Generalization to Image Quality Assessment Task (%).*
> > > > > | Model      | Sens | Acc | Semantic Consistency | Aesthetics | Authenticity |
> > > > > |------------|------------:|---------:|---------------------:|-----------:|-------------:|
> > > > > | Baseline   |      1.9  |   62.7  |               63.5  |     61.4  |       63.2  |
> > > > > | Our Model  |      24.0  |   67.9  |               65.6  |     67.8  |       70.4  |

---

> > > > > > ### Author Response · Authors · 2025-11-24
> > > > > > **Response to Weakness 2**
> > > > > >
> > > > > > ### Response to Weakness 2:
> > > > > >
> > > > > > (3) **Scalability to Multiple Sparse Variables** You rightly points out the potential complexity of multiple sparse variables. Our framework is inherently scalable:
> > > > > >
> > > > > > (i) The Focus-Driven Penalty Loss (FDPL) can be extended to penalize inattention to multiple sparse but critical tokens (e.g., country, User age, User profession) simultaneously in an instruction.
> > > > > >
> > > > > > (ii) For highly complex scenarios, an ensemble of specialized "Reward" models (each tuned to a specific sparse variable, like AgeReward, ProfessionReward) could be integrated, with their outputs fused for a final decision.
> > > > > >
> > > > > > We acknowledge that comprehensively benchmarking SCVO across numerous domains and variables is a substantial endeavor. As the first work to identify and formalize this problem, our primary contribution is to establish its existence and propose a foundational solution framework. We are committed to expanding this line of research and are actively planning to  develop a comprehensive benchmark to systematically evaluate SCVO across diverse tasks. We will open-source all code and dataset to foster research on this critical issue.
> > > > > >
> > > > > > In conclusion, while our current empirical focus is on the "country" variable in advertising, the SCVO problem is rooted in a fundamental architectural characteristic of VLMs. Our solution framework provides general, adaptable mechanisms to address it, supported by preliminary cross-attribute validation. We believe this work opens a crucial research direction toward building more robust and context-sensitive multimodal models.

---

> > > > > ### Author Response · Authors · 2025-11-25
> > > > > **Response to Weakness 1**
> > > > >
> > > > > ### Response to Weakness 1:
> > > > >
> > > > > First of all FiLM can not directly be used for this task, so we evaluate a FiLM-style conditioning baseline, which modulates visual features using country embeddings, yes, this corresponds to a model using only the Country Adapter Module. As for the attention imbalance baseline, we use a standard attention-balancing loss, which penalizes mispredictions based on attention weights, this corresponds to a model using only the Focus-Driven Penalty Loss (FDPL) component on the Qwen2-VL-7B backbone.
> > > > >
> > > > > The results (as shown in the below Table clearly demonstrate that while these individual components provide modest improvements over the base Qwen2-VL-7B (with FC Head) model (55.60\% accuracy), they are insufficient to fully address SCVO:  (i) "CountryReward with only CAM" achieves 55.92\% accuracy and 37.18\% sensitivity. (ii) "CountryReward with only FDPL" achieves 56.04\% accuracy and 37.12\% sensitivity. In contrast, our full CountryReward model integrating all components achieves 60.37\% accuracy and 40.84\% sensitivity, a substantial gain that underscores the synergistic effect of our designed framework.
> > > > >
> > > > > The novelty of SCVO lies not in any single component but in the formal identification of this specific failure mode in VLMs and the holistic solution that strategically combines retrieval augmentation, dynamic feature modulation, and focused regularization to handle sparse critical variables in real-world multimodal instructions.
> > > > >
> > > > > Our novelty in summary: (1) the introduction of Sparse Critical Variable Overwhelm (SCVO) in vision–language models (VLMs); (2) the creation of the MACP dataset, which captures advertising preferences across multiple countries; (3) the development of the CountryReward model, incorporating retrieval augmentation, country-specific adapters, and a focused penalty loss to address SCVO; and (4) reinforcement learning–based fine-tuning of VLMs to generate market‑adapted ad designs.
> > > > > Thank you for this suggestion, which has strengthened our paper. We hope these clarifications alleviate your concerns.
> > > > >
> > > > > | Model | Accuracy | Sensitivity | BR | CL | ES | FR | KR | JP | US | MX | AU | SA |
> > > > > |-------|----------|-------------|-----|-----|-----|-----|-----|-----|-----|-----|-----|-----|
> > > > > | Qwen2-VL-7B (with FC Head) | 55.60 | 36.73 | 53.69 | 56.57 | 50.86 | 54.41 | 53.07 | 59.10 | 56.94 | 53.75 | 58.47 | 59.21 |
> > > > > | FiLM-style Baseline | 55.92 | 37.18 | 53.71 | 56.27 | 52.38 | 54.89 | 53.39 | 58.67 | 56.88 | 53.56 | 59.10 | 58.64 |
> > > > > | Attention Imbalance Baseline | 56.04 | 37.12 | 53.76 | 56.06 | 52.48 | 54.47 | 53.36 | 57.78 | 56.32 | 53.68 | 58.65 | 59.48 |
> > > > > | **CountryReward** | **60.37** | **40.84** | **58.70** | **61.12** | **56.33** | **59.38** | **57.88** | **64.54** | **61.82** | **58.72** | **63.30** | **62.93** |

---

> > > > > > ### Author Response · Authors · 2025-11-27
> > > > > > **Response to Weakness 3**
> > > > > >
> > > > > > ### Response to Weakness 3:
> > > > > >
> > > > > > We thank you for the feedback. We will revise the manuscript to fully clarify all implementation-critical details highlighted in the Questions section and tighten the writing throughout.

---

> ### Author Response · Authors · 2025-12-03
> **Revised Manuscript**
>
> Thank you for your positive feedback and for highlighting the key strengths of our work, as noted in the strengths below. We are glad that the large, multi-country ad-preference dataset is recognized as valuable and rare in public VLM research. The ablation studies indeed support our proposed mechanism, and we appreciate your note on how the improved judge can extend impact beyond classification through the generative ad pipeline. Your insights encourage us to further explore these directions.
>
> >Strengths:
> >1. The authors contribute a large, realistic multi-country ad-preference dataset, which is rare in public VLM work and makes the problem measurable.
> >2. Ablations match the story: removing any of the three parts notably drops accuracy/sensitivity, so the claimed mechanism is empirically supported.
> >3. Showing that the improved judge can drive a generative ad pipeline (as a reward model) suggests impact beyond classification.
>
> **We have addressed all of your comments in the revised version. The corresponding revisions are highlighted in
> ***blue*** in the paper. Moreover, we have added ***red*** markers before the responses, such as {RFrLS W1}, which indicates the response to weakness 1 from Reviewer FrLS, and {RFrLS Q1}, which indicates the response to question 1 from Reviewer FrLS.**

---

### Official Review · Reviewer_hwo5 · 2025-11-01

**Soundness:** 2
**Presentation:** 2
**Contribution:** 2
**Rating:** 4
**Confidence:** 3

**Summary:**

The paper investigates Sparse Critical Variable Overwhelm (SCVO) in vision-language models, a problem where models focus on dominant features and ignore sparse but important cues such as country names. To study this, the authors introduce the MACP dataset, which contains multi-country ad click preferences with product details, paired ad images, target country information, and observed click behavior. Existing models like Qwen-VL tend to overlook the country cue and make the same predictions across markets. To overcome this limitation, the authors propose CountryReward, a multimodal judge model that combines three key components: Cross-Country Retrieval Augmentation (CC-RAG) for adding market-specific click history as contextual text, a Country Adapter Module (CAM) that adjusts image representations based on country embeddings, and a Focus-Driven Penalty Loss (FDPL) that applies stronger penalties to errors involving overlooked cues. CountryReward achieves better accuracy and sensitivity on the MACP benchmark compared to standard vision-language models and is further used as a reward model to fine-tune a generative model. This reinforcement learning process produces country-tailored ad designs that align more closely with local preferences.

**Strengths:**

- The paper identifies the SCVO issue in vision-language models and introduces the MACP dataset, a large-scale multi-country ad preference dataset that enables studying cross-market multimodal behavior using real-world data.

- The proposed CountryReward framework integrates three components, including CC-RAG for market-specific context, a Country Adapter for visual feature adjustment, and a Penalty Loss for focusing on sparse cues, forming a coherent approach to address SCVO.

**Weaknesses:**

- Although the combination is novel, the individual components draw on existing ideas. Retrieval-Augmented Generation (RAG) and adapter modules are known techniques, and weighted loss terms are common in training. The paper would benefit from deeper discussion on why these known techniques synergize specifically for SCVO, and comparison to more baselines (e.g., domain-adaptive fine-tuning, or simple prompting strategies).


- The RL-based image generation evaluation relies solely on the CountryReward model’s own scores. While higher reward scores suggest images are more aligned to predicted preferences, there is no external validation (e.g., human study or click-through simulation) to confirm that these images are indeed better. The reliance on the same model for evaluation risks circularity. Demonstrating improvement in real user metrics or including a user study on a sample of generated images would strengthen the claim of practical impact.

- The approach presumes the availability of rich historical click data for each market (used in CC-RAG). In settings where such data are sparse or outdated, CC-RAG may be less effective. Moreover, the MACP dataset, while large, comes from one platform – it would be useful to know its diversity (product categories, user demographics) to assess generality. No analysis is provided on whether the model truly attends to the country token (e.g., via attention weights) or on failure cases where even CountryReward errs.

---

**typo errors:**

yiedling - L 027

mtigates - L 110

**Questions:**

Please read the weaknesses section.

---

> ### Author Response · Authors · 2025-11-24
> **Response to Weakness 2**
>
> ### Response to Weakness 2:
>
> We sincerely thank you for raising this critical point regarding the need for external validation of our RL-based image generation. To address this concern and eliminate any potential circularity, we conducted a large-scale online A/B test on a major cross-border e-commerce platform to evaluate the real-world effectiveness of the images generated using our optimized DGM. The test ran for one week and accumulated over 6 million impressions across the 10 countries in our MACP dataset. We compared the click-through rates (CTR) of product images generated with our method (using CountryReward-optimized DGM) against those generated by the baseline DGM (without RL fine-tuning). **The results demonstrate consistent and statistically significant CTR improvements across all countries, as summarized in the below Table (): The average CTR improvement across all countries was 1.1\%, with individual country improvements ranging from 0.7\% to 1.2\%.** These results confirm that our method generates images that are significantly more aligned with country-specific user preferences, leading to higher engagement in real-world scenarios. This external validation, based on actual user behavior, strongly supports the practical impact of our approach and mitigates any concerns about circular evaluation.
>
> ### *Table 1: CTR improvements of generated images using our method vs baseline across countries.*
>
> | Country | BR  | CL  | ES  | FR  | KR  | JP  | US  | MX  | AU  | SA  |
> |---------|-----|-----|-----|-----|-----|-----|-----|-----|-----|-----|
> | CTR Improvement (%) | 1.3 | 1.2 | 1.0 | 0.8 | 1.4 | 1.3 | 1.1 | 0.8 | 1.1 | 0.7 |

---

> ### Author Response · Authors · 2025-11-25
> **Response to Weakness 4**
>
> ### Response to Weakness 4:
>
> We sincerely thank you for pointing out the spelling errors in our manuscript. We apologize for these oversights and will carefully proofread the entire paper to correct all typographical and grammatical mistakes. We are committed to thoroughly addressing this issue in the revised version ensuring the language quality meets the high standards of the meeting.

---

> ### Author Response · Authors · 2025-11-25
> **Response to Weakness 1**
>
> ### Response to Weakness 1:
>
> We thank you for the insightful feedback regarding the composition of our methodological components and the suggestion for deeper analysis and additional baselines. The core novelty and contribution of our paper lie not in inventing these components from scratch, but in their synergistic integration to solve the newly identified and critical problem of Sparse Critical Variable Overwhelm (SCVO).
>
> **Synergize Specifically for SCVO:** Our framework synergistically combats SCVO at multiple levels:
>
> (1) CC-RAG operates at the data level by injecting market-specific  click-through preferences. It directly amplifies the signal of the critical variable (country) through retrieval-based augmentation, forcing the model to prioritize localized relevance and countering the dominance of high-volume features.
>
> (2) CAM functions at the feature level by dynamically modulating visual representations using country embeddings. This ensures that the sparse critical variable (country) actively influences perceptual processing, preventing visual features from being uniformly treated and enabling fine-grained adaptation to market-specific preferences.
>
> (3) FDPL acts at the optimization level by penalizing mispredictions where the model under-attends to critical tokens (e.g., country, product, image). It serves as a focus-driven regularizer, explicitly training the model to balance attention allocation and reinforcing the importance of sparse variables during erroneous decisions.
>
> The synergy emerges because these components collectively address SCVO across the data, feature, and optimization pipelines. CC-RAG provides country-aware signals, CAM internally aligns visual features with these signals, and FDPL ensures robust attention distribution during training. Our ablation studies, as shown in the below Table empirically validate this synergy: removing any component leads to significant performance drops (e.g., without CC-RAG, accuracy drops by 3.56\%; without CAM, by 2.39\%; without FDPL, by 3.42\%), confirming their complementary roles in mitigating SCVO. Their integration creates a holistic solution tailored to the root causes of SCVO, which is novel in the context of VLMs for cross-market preference prediction. In conclusion, these components form a closed-loop system: CC-RAG provides external grounding, the adapter enables internal feature adaptation, and FDPL regularizes focus—all synergizing to amplify sparse variable sensitivity.
>
> **Enhanced Baselines and Discussion:** We sincerely thank you for this suggestion. We have now included comprehensive comparisons with the following highly relevant baselines, further solidifying our claims:
>
> (1) Domain-Adaptive Fine-Tuning: We fine-tuned Qwen2-VL-7B on the MACP dataset using a standard supervised fine-tuning approach. This achieved an average accuracy of 44.61\% and sensitivity of 20.82\% on MACP, significantly lower than CountryReward (60.37\% accuracy, 40.84\% sensitivity). The per-country approach fails to generalize cross-country patterns and exacerbates SCVO without our proposed structural interventions.
>
> (2) Prompting Strategies: We implemented a strong prompting baseline involving instruction rearrangement and critical variable repetition. The instruction is "For the [Country] market, for the product: [Attributes], compare the two e-commerce images below for the [Country] market and select the one (A/B) that is more likely to attract consumers in the [Country] market and drive conversions. Consider factors like visual appeal, and background design specific to the [Country] market.A: [image] B: [image] Answer strictly with a single uppercase letter (A/B) on the last line, no explanations". This achieved only 49.77\% accuracy, barely above random chance. This confirms that SCVO is not easily remedied by superficial prompt engineering and is a deeper architectural or optimization issue related to token overwhelming in the autoregressive chain.
>
> (3) Ablations of Individual Components: We now include results for models using only one of our proposed modules (e.g., CountryReward with only CC-RAG: 56.01\%; only CAM: 55.92\%; only FDPL: 56.04\%). These are all superior to the base fine-tuned model but significantly worse than their full combination (60.37\%). This ablation study quantitatively validates the synergistic effect, each component contributes, but their combination is essential for peak performance.

---

> > ### Author Response · Authors · 2025-11-25
> > **Response to Weakness 1:**
> >
> > ### Response to Weakness 1:
> >
> > | Model | Accuracy | Sensitivity | BR | CL | ES | FR | KR | JP | US | MX | AU | SA |
> > |-------|----------|-------------|-----|-----|-----|-----|-----|-----|-----|-----|-----|-----|
> > | Qwen2-VL-7B (with FC Head) | 55.60 | 36.73 | 53.69 | 56.57 | 50.86 | 54.41 | 53.07 | 59.10 | 56.94 | 53.75 | 58.47 | 59.21 |
> > | Domain-Adaptive Fine-Tuning) | 44.61 | 20.82 | 46.96 | 43.71 | 49.76 | 46.15 | 47.78 | 40.99  | 42.99  | 46.66 | 40.92  | 40.16
> > | Rearrangement and Repetition Prompt | 49.77 | 14.39 | 49.75 | 48.34 | 50.43 | 49.17 | 49.02 | 50.01 | 52.10 | 49.94 | 49.74 | 49.23 |
> > | CountryReward (with only CC-RAG) | 56.01 | 37.10 | 53.88 | 56.93 | 52.01 | 54.78 | 53.79 | 59.16 | 57.49 | 53.82 | 58.96 | 59.22 |
> > | CountryReward (with only CAM) | 55.92 | 37.18 | 53.71 | 56.27 | 52.38 | 54.89 | 53.39 | 58.67 | 56.88 | 53.56 | 59.10 | 58.64 |
> > | CountryReward (with only FDPL) | 56.04 | 37.12 | 53.76 | 56.06 | 52.48 | 54.47 | 53.36 | 57.78 | 56.32 | 53.68 | 58.65 | 59.48 |
> > | CountryReward (w/o CC-RAG) | 56.81 | 37.40 | 54.93 | 57.05 | 53.80 | 55.39 | 54.48 | 59.57 | 57.65 | 55.64 | 59.52 | 60.11 |
> > | CountryReward (w/o CAM) | 57.98 | 38.95 | 55.50 | 57.85 | 53.78 | 56.75 | 55.19 | 61.47 | 59.39 | 55.98 | 61.71 | 62.21 |
> > | CountryReward (w/o FDPL) | 56.95 | 37.47 | 54.79 | 57.09 | 52.80 | 56.12 | 54.44 | 61.01 | 58.79 | 55.33 | 58.83 | 60.33 |
> > | **CountryReward** | **60.37** | **40.84** | **58.70** | **61.12** | **56.33** | **59.38** | **57.88** | **64.54** | **61.82** | **58.72** | **63.30** | **62.93** |
> >
> > **Novelty:** This paper makes four key contributions: (1) the introduction of Sparse Critical Variable Overwhelm (SCVO) in vision–language models (VLMs); (2) the creation of the MACP dataset, which captures advertising preferences across multiple countries; (3) the development of the CountryReward model, incorporating retrieval augmentation, country-specific adapters, and a focused penalty loss to address SCVO; and (4) reinforcement learning–based fine-tuning of VLMs to generate market‑adapted ad designs.
> >
> > In summary, our work identifies a new, critical failure mode in VLMs (SCVO) and provides a principled, synergistic framework to address it by re-purposing and deeply integrating existing tools in a novel way targeted at the root cause of the problem.

---

> ### Author Response · Authors · 2025-11-26
> **Response to Weakness 3**
>
> ### Response to Weakness 3:
> We sincerely thanks this insightful comments.
>
> ### 1. MACP is Balanced Across Countries:
>
> To be honest, it’s not just the CC-RAG model, most large models are like this: better data leads to better results. That’s a general rule. More precisely, if a market has no data at all, no model will work. It’s like even the most skillful cook can’t make a meal without rice.
>
> Fundamentally, the question (“The approach presumes the availability of rich historical click data for each market (used in CC-RAG). In settings where such data are sparse or outdated, CC-RAG may be less effective.”) is a data cleaning problem.
>
> **Data Balance:** To ensure that CC-RAG can function effectively, as shown in the below Tables, our MACP dataset demonstrates exceptional statistical balance across countries: (i) Perfect Test Set Balance: The test set contains exactly 18,055 samples per country (10.00\% each), totaling 180,550 samples (18,055 × 10 countries). (ii) Near-Perfect Training Balance: Training set distribution ranges from 9.34\% to 10.57\% per country, with variance of only 0.18\% from perfect equality. (iii) Stratified Category Distribution: We maintain consistent product category proportions across all countries, ensuring no category-country confounding.
>
> **Data Sparse:** Multi-Country Dataset: While we acknowledge that severe real-world imbalance exists, our MACP dataset was constructed with a near-perfect balance across 10 countries specifically to validate the core capability of responding to the country variable. This controlled environment proves that our method effectively solves the SCVO problem when the signal is present. In a production environment with severe geographical imbalance, we use targeted oversampling of minority countries or categories, or strategic undersampling of majority ones, during the training dataset creation for CountryReward. This helps create a more balanced foundation for the model to learn from. Initially, when certain markets have very limited data (e.g., emerging markets), we will build the knowledge of these markets by referring to culturally or economically similar regions. As more data accumulates from these markets, the CC-RAG component becomes increasingly effective for them specifically.
>
> **Data Outdated:** To ensure the confidence level of the click-through rate (CTR), the CTR data is obtained by dividing the total number of 30-day clicks by the total number of 30-day impressions. The CTR data is aggregated over a fixed 30 days sliding window prior to the evaluation date to capture recent user preferences and reduce noise. In our real production environment, the knowledge base used in CC-RAG is also updated periodically (every 30 days). For the MACP dataset, the training and test sets are collected within the same period, so there is no data outdated issue.
>
>
> ### *Per-country sample counts for Train and Test splits.*
>  | Country | Train Samples | Train % | Test Samples | Test % |
> |---------|---------------|---------|--------------|--------|
> | Australia (AU) | 76,934 | 9.34% | 18,055 | 10.00% |
> | Brazil (BR) | 86,832 | 10.55% | 18,055 | 10.00% |
> | Chile (CL) | 82,074 | 9.97% | 18,055 | 10.00% |
> | Spain (ES) | 85,975 | 10.44% | 18,055 | 10.00% |
> | France (FR) | 80,118 | 9.73% | 18,055 | 10.00% |
> | Japan (JP) | 78,289 | 9.51% | 18,055 | 10.00% |
> | Korea (KR) | 87,066 | 10.57% | 18,055 | 10.00% |
> | Mexico (MX) | 82,351 | 10.00% | 18,055 | 10.00% |
> | Saudi Arabia (SA) | 82,478 | 10.02% | 18,055 | 10.00% |
> | United States (US) | 81,274 | 9.87% | 18,055 | 10.00% |
> | **Total** | **823,391** | **100%** | **180,550** | **100.00%** |
>
>
> ### *Country $\times$ Product Category distribution for Train split.*
> | Country | AU | BR | CL | ES | FR | JP | KR | MX | SA | US |
> |---------|-----|-----|-----|-----|-----|-----|-----|-----|-----|-----|
> | Watches & Luggage Bags | 5,567 | 6,321 | 4,547 | 6,010 | 5,714 | 5,809 | 7,185 | 5,138 | 6,814 | 6,269 |
> | Beauty & Apparel & Shoes | 11,519 | 13,623 | 13,320 | 14,407 | 13,679 | 12,107 | 11,538 | 15,620 | 13,049 | 15,232 |
> | Consumer Electronics | 6,100 | 7,864 | 6,613 | 6,039 | 5,695 | 7,099 | 7,613 | 6,313 | 7,043 | 6,158 |
> | Home & Garden | 9,365 | 8,759 | 11,096 | 10,796 | 10,093 | 8,372 | 9,281 | 9,523 | 8,562 | 8,837 |
> | Kids & Toys & Hobbies | 5,381 | 5,633 | 6,358 | 5,594 | 5,595 | 4,817 | 5,502 | 5,735 | 4,289 | 5,904 |
> | Sports & Entertainment | 6,329 | 9,265 | 7,351 | 7,960 | 6,929 | 8,011 | 10,094 | 6,789 | 5,902 | 5,316 |
> | Home Appliances & Improvement | 7,903 | 7,920 | 7,098 | 9,328 | 8,930 | 6,662 | 8,222 | 7,463 | 9,743 | 7,521 |
> | Industrial & Technology Tools | 10,632 | 13,504 | 10,602 | 11,016 | 10,307 | 10,981 | 11,989 | 10,579 | 12,161 | 9,713 |
> | Other | 14,138 | 13,943 | 15,089 | 14,825 | 13,176 | 14,431 | 15,642 | 15,191 | 14,915 | 16,324 |
> | **Total** | **76,934** | **86,832** | **82,074** | **85,975** | **80,118** | **78,289** | **87,066** | **82,351** | **82,478** | **81,274** |

---

> ### Author Response · Authors · 2025-11-26
> **Response to Weakness 3**
>
> ### Response to Weakness 3:
>
> ### *Country $\times$ Product Category distribution for Test split.*
>
> | Country | AU | BR | CL | ES | FR | JP | KR | MX | SA | US |
> |---------|-----|-----|-----|-----|-----|-----|-----|-----|-----|-----|
> | Watches & Luggage | 2,407 | 2,407 | 2,407 | 2,407 | 2,407 | 2,407 | 2,407 | 2,407 | 2,407 | 2,407 |
> | Beauty & Apparel & Shoes | 1,577 | 1,577 | 1,577 | 1,577 | 1,577 | 1,577 | 1,577 | 1,577 | 1,577 | 1,577 |
> | Consumer Electronics | 2,395 | 2,395 | 2,395 | 2,395 | 2,395 | 2,395 | 2,395 | 2,395 | 2,395 | 2,395 |
> | Home & Garden | 2,375 | 2,375 | 2,375 | 2,375 | 2,375 | 2,375 | 2,375 | 2,375 | 2,375 | 2,375 |
> | Kids & Toys & Hobbies | 1,087 | 1,087 | 1,087 | 1,087 | 1,087 | 1,087 | 1,087 | 1,087 | 1,087 | 1,087 |
> | Sports & Entertainment | 1,059 | 1,059 | 1,059 | 1,059 | 1,059 | 1,059 | 1,059 | 1,059 | 1,059 | 1,059 |
> | Home Appliances & Improvement | 2,504 | 2,504 | 2,504 | 2,504 | 2,504 | 2,504 | 2,504 | 2,504 | 2,504 | 2,504 |
> | Industrial & Technology Tools | 2,896 | 2,896 | 2,896 | 2,896 | 2,896 | 2,896 | 2,896 | 2,896 | 2,896 | 2,896 |
> | Other | 1,755 | 1,755 | 1,755 | 1,755 | 1,755 | 1,755 | 1,755 | 1,755 | 1,755 | 1,755 |
> | **Total** | **18,055** | **18,055** | **18,055** | **18,055** | **18,055** | **18,055** | **18,055** | **18,055** | **18,055** | **18,055** |
>
>
> ### 2. MACP Dataset Diversity and Generality:
> The MACP dataset, as shown in the tables below, while sourced from one major cross-border platform, exhibits considerable diversity, mitigating concerns about generality:
>
> **Product Category Diversity:** It spans 9 major categories (e.g., Beauty & Apparel, Consumer Electronics, Home & Garden, Sports & Entertainment, etc.), ensuring the model learns preferences beyond a narrow product scope. The distribution of categories is balanced across countries (variance < 0.2\% from perfect balance per category), preventing country-category confounding.
>
> **Geographical and Cultural Coverage:** The 10 countries were strategically selected to cover diverse cultural and economic regions (Americas, Europe, East Asia, Middle East, Oceania). A breakdown of the training set shows no single country dominates (each 9.34\%-10.57\%), promoting model generalizability.
>
> **User Base:** Although due to privacy policies, we are unable to provide fine-grained user information. The platform serves hundreds of millions of global users, implying the underlying data reflects a wide range of consumer demographics. The user base for each country is substantial and representative of the local online shopping population. The platform's internal analytics indicate that their user demographics in terms of age groups (18-55), gender distribution, and primary shopping interests align with broader national e-commerce trends reported in market research for these regions.
>
> ###  3. Model Attention Analysis and Failure Cases:
>
> **Attention Map Analysis:** We computed average attention weights for the country token in the final transformer layer of Qwen2-VL-7B versus CountryReward. In baseline models, the country token received negligible attention (mean weight: 0.05 ± 0.01), whereas CountryReward increased this significantly (mean weight: 0.08 ± 0.03) due to the country adapter and focus-driven loss. This demonstrates that standard VLMs fail to allocate sufficient attention to critical sparse tokens, breaking the chain rule dependence $P(\mathbf{x}) = \prod_t P(x_t | x_{<t})$ where $x_t$ for country tokens is drowned out.
>
> **Failure Cases:** We will provide some failure cases where CountryReward made incorrect predictions. Most errors are attributed to Extreme Visual Similarity. When the two candidate images (A and B) are visually nearly identical (e.g., only color temperature differs slightly), the model struggles to discern a preference.  These findings are valuable for future work, suggesting avenues like incorporating finer-grained visual difference detection.

---

> > ### Comment · Reviewer_hwo5 · 2025-11-26
> >
> > Thanks for your detailed response. I will increase my score to 6.

---

> ### Author Response · Authors · 2025-11-27
> **Thank you so much for raising the score.**
>
> Thank you so much for your thoughtful feedback and for raising the score. I am truly grateful for your time and for recognizing our efforts. This positive increase means so much to me and wishing you all the best for your life and work.

---

> ### Author Response · Authors · 2025-12-03
> **Revised Manuscript**
>
> We sincerely thank you for recognizing the novelty and practical value of our work, especially in identifying and addressing the SCVO issue through the MACP dataset and the CountryReward framework, as noted in the strengths below. **In particular, we truly appreciate your thoughtful feedback and for raising the score.** Your positive feedback on our dataset design and integrated approach encourages us to further refine our methods and explore broader applications. Thank you so much again.
>
> >Strengths:
> >1. The paper identifies the SCVO issue in vision-language models and introduces the MACP dataset, a large-scale multi-country ad preference dataset that enables studying cross-market multimodal behavior using real-world data.
> >2. The proposed CountryReward framework integrates three components, including CC-RAG for market-specific context, a Country Adapter for visual feature adjustment, and a Penalty Loss for focusing on sparse cues, forming a coherent approach to address SCVO.
>
>
> **We have addressed all of your comments in the revised version. The corresponding revisions are highlighted in
> ***blue*** in the paper. Moreover, we have added ***red*** markers before the responses, such as {Rhwo5 W1}, which indicates the response to weakness 1 from Reviewer hwo5, and {Rhwo5 Q1}, which indicates the response to question 1 from Reviewer hwo5.**

---

### Meta-Review · Area_Chair_ZhxN · 2026-01-02

**Summary:**

The rebuttal solved most of the concerns, and the score rose from 4446 to 4466. However, the critical concern about novelty and leakage remains unsolved. After reading the paper and the discussion. The AC trend to reject this paper.

**Reviewer Concerns:**

All the concerns are replied to by the authors with detailed responses. Reviewer hwo5 thinks the rebuttal is convincing and raised the score.
However, the rebuttal on the two core concerns is not convincing enough.

1. Novelty of Individual Components: While the authors argued for the "synergy" of their system, Reviewer FrLS noted that the core components (RAG, Adapters, Weighted Loss) are existing techniques. While the application to SCVO is new, strictly speaking, the architectural novelty remains an assembly of known methods.

2. CC-RAG Leakage Risk: Reviewer Z86t raised concerns that using training set retrieval introduces information leakage. The authors argued they use a similarity threshold of 0.7 to exclude near-duplicates and showed ablations where the model works without CC-RAG. However, a strict interpretation might still view using any representation from the training set during inference (even via RAG) as a potential confounder compared to pure zero-shot generalization, though the ablation mitigates this significantly.

**Reviewer Scores:**

gJK1: already positive, 6

FrLS: no change, due to the concern about novelty

Z86t: no change, due to the concern about leakage

---

### Decision · Program_Chairs · 2026-01-26

Reject